

# Emergent supersymmetry on the edges

Jin-Beom Bae[1] and Sungjay Lee[2]

**1** Mathematical Institute, University of Oxford, Andrew Wiles Building,
Radcliffe Observatory Quarter, Woodstock Road, Oxford, OX2 6GG, U.K.
**2** Korea Institute for Advanced Study, 85 Hoegiro ,
Dongdaemun-Gu, Seoul 02455, Korea

## Abstract

The WZW models describe the dynamics of the edge modes of Chern-Simons theories in three dimensions. We explore the WZW models which can be mapped to supersymmetric theories via the generalized Jordan-Wigner transformation. Some of such models have supersymmetric Ramond vacua, but the others break the supersymmetry spontaneously. We also make a comment on recent proposals that the Read-Rezayi states at filling fraction $\nu = 1/2,\ 2/3$ are able to support supersymmetry.



# 1 Introduction and Conclusion

Ever since its discovery, space-time supersymmetry [1] has been widely utilized in high-energy physics and cosmology. Despite its theoretical attraction, the presence of supersymmetry in nature has not been observed in high-energy experiments yet. Instead, the series of recent works suggested interesting ideas of realizing space-time supersymmetry in the condensed matter systems [2, 3, 4, 5, 6, 7].

Of particular interest are the integer and fractional quantum Hall systems. The fractional quantum Hall states are shown to have topological order[8, 9] which is beyond the Landau theory of symmetry breaking. Those topologically ordered states are known to possess gapless edge excitations with fractional charges.

The plateaus of fractional quantum Hall effect are characterized by filling fraction $\nu$, the parameter associated with Hall conductivity $\sigma = \frac{e^2}{2\pi\hbar}\nu$. The theoretical approach to the observed filling fractions was pioneered by Laughlin [10], and ideas such as hierarchy states [11, 12] and the composite fermion [13] were proposed to explain the fractional quantum Hall states with Abelian anyonic statistics. Soon afterwards, it was shown in [14, 15] that the Pfaffian wave function provides theoretical understanding of the non-Abelian fractional quantum Hall effect.

The Chern-Simons theories in the 2+1 space-time dimensions provide low-energy effective theories capturing the response of the quantum Hall ground state to low-energy fluctuations. Especially, the non-Abelian braiding statistics of the anyon can be understood essentially from the computation of Wilson lines of Chern-Simons theories on $S^3$ [16]. The quantization of the Chern-Simons theory on a manifold with boundaries leads to a chiral rational conformal field theory(RCFT) [16, 17] that describes the edge excitations of quantum Hall states. A prominent example is a correspondence between Laughlin states of $\nu = 1/k$ and RCFT with $u(1)_k$ affine algebra. For more details, see the discussion in [18, 19].

The goal of this paper is to explore the emergent supersymmetry on the edges of the 2+1-dimensional gapped phases. By the superymmetry on the edges, we mean that a chiral RCFT has a chiral primary $G(z)$ of conformal weight $h = 3/2$ satisfying the operator product expansion(OPE) below,

$$
\begin{aligned}
T(z)G(0) &\sim \frac{3}{z^2}G(0) + \frac{1}{z}\partial G(0), \\
G(z)G(0) &\sim \frac{2c}{3z^3} + \frac{1}{z}T(0),
\end{aligned}
\tag{1}
$$

where $T(z)$ is the stress-energy tensor with the central charge $c$. Then the primary $G(z)$ plays a role of the supersymmetry current.

In the present work, we mainly pay attention to the non-chiral Wess-Zumino-Witten(WZW) models that has a chiral primary of $h = 3/2$ obeying (1). Once such a model exists, its chiral sector that can arise on the edges is then able to support a supersymmetry current. The above problem is also equivalent to another problem searching for non-chiral WZW models associated with supersymmetric theories via the so-called the Jordan-Wigner transformation [20, 21] where the chiral primary of $h = 3/2$ of the WZW model becomes the superymmetry current.

To understand the equivalence, let us consider an well-known example, the tricritical Ising model. The tricritical Ising model is the Virasoro minimal model with central charge $c = 7/10$, and has six chiral primaries of $h = 0$, 1/10, 3/5, 3/2, 3/80 and 7/16 whose characters are denoted by $\chi_h(\tau)$. The model is a bosonic CFT by itself and thus has no supersymmetry current, which is clearly understood from its torus partition function

$$
Z_{\mathcal{B}} = \left|\chi_{h=0}(\tau)\right|^2 + \left|\chi_{h=\frac{1}{10}}(\tau)\right|^2 + \left|\chi_{h=\frac{3}{5}}(\tau)\right|^2 + \left|\chi_{h=\frac{3}{2}}(\tau)\right|^2 + \left|\chi_{h=\frac{3}{80}}(\tau)\right|^2 + \left|\chi_{h=\frac{7}{16}}(\tau)\right|^2.
$$

It is known that the model has a primary operator $\epsilon''(z,\bar{z})$ of $(h,\bar{h}) = (3/2, 3/2)$ that satisfies the OPE

$$\epsilon''(z,\bar{z})\epsilon''(0) \sim \left(\frac{7}{15}\right)^2 \frac{1}{(z\bar{z})^3} + \frac{7}{15}\left(\frac{T(0)}{z\bar{z}^3} + \frac{\bar{T}(0)}{z^3\bar{z}}\right) + \frac{T(0)\bar{T}(0)}{z\bar{z}}. \tag{2}$$

Note that the above OPE involves only the identity operator and its descendants, consistent with the fusion algebra of the tricritical Ising model. The OPE (2) implies that the chiral part of $\epsilon''(z,\bar{z})$, denoted by $\epsilon''(z)$, would satisfy (1) with $c = 7/10$, namely the chiral part of the model has the supersymmetry current. On the other hand, the generalized Jordan-Wigner transformation maps the (non-chiral) tricritical Ising model to the $\mathcal{N} = 1$ supesymmetric minimal model with $c = 7/10$ [21] where the chiral primary $\epsilon''(z)$ becomes the supersymmetry current. From the partition function of the $\mathcal{N} = 1$ supersymmetric minimal model with $c = 7/10$ in the Neveu-Schwarz(NS) sector,

$$\begin{aligned}
Z^{\text{NS}} &\equiv \text{tr}_{\mathcal{H}_{NS}}\left[q^{L_0 - c/24}\bar{q}^{\bar{L}_0 - c/24}\right] \\
&= \left|\chi_{h=0}(\tau) + \chi_{h=\frac{3}{2}}(\tau)\right|^2 + \left|\chi_{h=\frac{1}{10}}(\tau) + \chi_{h=\frac{3}{5}}(\tau)\right|^2 \\
&= \chi_{h=0}(\tau)\bar{\chi}_{\bar{h}=\frac{3}{2}}(\bar{\tau}) + \chi_{h=\frac{3}{2}}(\tau)\bar{\chi}_{\bar{h}=0}(\bar{\tau}) + \cdots,
\end{aligned} \tag{3}$$

where $\chi_h(\tau)$ are the characters of the tricritical Ising model, one can see that the chiral primary $\epsilon''(z)$ of $h = 3/2$ indeed appears as the descendant of the vacuum and becomes the holomorphic supersymmtry current.

A fermionic CFT refers to a conformal field theory which has primaries of half-integer spin. We need to choose a spin structure to define such a fermionic CFT. On a two-torus, there exist four different spin structures denoted by (NS,NS), (R,NS), (NS,R) and (R,R) where the former in the parenthesis specifies either the Neveu-Schwarz(NS) or the Ramond(R) boundary condition along the temporal circle while the latter along the spatial circle. We use a shorthand notation NS, $\widetilde{\text{NS}}$, R and $\widetilde{\text{R}}$ for those spin structures in what follows.

Of course, the WZW models themselves cannot be a supersymmetric theory due to their bosonic nature. Instead, we will apply the fermionization, a.k.a. the generalized Jordan-Wigner transformation, to convert them to candidates of interest. A modern understanding of the fermionization is to couple a bosonic theory having non-anomalous $\mathbb{Z}_2$ symmetry with the low-energy limit of the topological phase of the Kitaev Majorana chain followed by the $\mathbb{Z}_2$ quotient [22, 20]. This idea has been applied in recent works [23, 21, 24, 25, 26, 27]. The fermionization also provides a theoretical ground to the well-known fact that the tricritical Ising model is supersymmetric [28], as explained above. The target of our analysis is to search for supersymmetric RCFTs with $c \geq 1$ and we fermionize the WZW models that has no more than 60 primaries to achieve the goal.

Although there are many known constructions for the unitary supersymmetric RCFTs such as the supersymmetric minimal models [28, 29, 30], $\mathcal{N} = 1$ extremal supersymmetric conformal field theory(SCFT) [31], $\mathcal{N} = 1$ "Beauty and the Beast" SCFT [32], the full classification is still far-fetched. The classification program of fermionic RCFTs with a few number of primaries has been studied only recently. For instance, see [33, 34]. Instead of explicit construction or full classification, we rather focus on the essential features of the supersymmetric RCFTs for the exploration. First, the NS-sector has to contain spin-3/2 currents as the vacuum descendants that play a role of the supersymmetry currents. Second, the torus partition function of $\widetilde{\text{R}}$-sector becomes an index and thus take a constant value. Finally, the R-sector primaries satisfy the supersymmetric unitarity constraint $h^R \geq \frac{c}{24}$. Whenever a fermionic RCFT fulfills the above necessary conditions, we regard it as an unitary supersymmetric RCFT. We also check if the candidate partition functions allow the super Virasoro character decomposition. For later convenience, we refer to the above four conditions as SUSY conditions.

Table 1: List of WZW models that allows fermionization to supersymmetric models with supersymmetric vacua.

| Type | $G_k$ |
|---|---|
| A-type | $SU(12)_1, SU(4)_3, SU(6)_2, SU(2)_6$ |
| A$'$-type | $SU(4)_4/\mathbb{Z}_2$ |
| C-type | $Sp(4)_3, Sp(6)_2, Sp(12)_1$ |
| D-type | $SO(8)_3, SO(12)_2, SO(24)_1$ |
| $SO(N)_3$ | $SO(3)_3(=SU(2)_6), SO(4)_3(=(SU(2)_6)^2),$ $SO(5)_3(=Sp(4)_3), SO(6)_3=SU(4)_3, SO(7)_3, \cdots$ |

Table 2: List of WZW models whose $\mathbb{Z}_2$ orbifold allows fermionization to supersymmetric models. However their Ramond vacua do not saturate the supersymmetric unitarity bound.

| Type | $G_k$ |
|---|---|
| E-type | $(E_7)_2, (E_8)_2$ |
| Orbifold | $SO(16)_2/\mathbb{Z}_2, SU(16)_1/\mathbb{Z}_2, SU(8)_2/\mathbb{Z}_2$ |

The main results of this paper are summarized in table 1 and 2. The WZW models listed in table 1 can be fermionized to satisfy the aforementioned SUSY conditions. Moreover, their Ramond vacua saturate the supersymmetric unitarity bound $h^R \geq \frac{c}{24}$. This implies that supersymmetry in the Ramond sector remains unbroken. We also remark that those WZW models in table 1 are shown in the recent work [35] to have $\mathcal{N} = 1$ supersymmetric vertex operator algebra.

On the one hand, table 2 presents the WZW models that can be mapped to fermionic theories satisfying the SUSY condition via the Jordan-Wigner transformation. However, those model have none of the R-sector primaries saturating the unitarity bound $h^R \geq \frac{c}{24}$. In other words, supersymmetry is spontaneously broken in the Ramond sector.

We also investigate the emergence of supersymmetry in certain chiral RCFTs proposed to describe the Read-Rezayi states at filling fraction $\nu = \frac{k}{kM+2}$ for $k = 2, 3, 4, \cdots$ and nonnegative $M$ [15]. We observe that the fermionized RCFTs for $(k = 2, M = 1)$ and $(k = 4, M = 1)$ can be identified with the $\mathcal{N} = 2$ unitary supersymmetric minimal models, consistent with the recent work of [36, 37].

This paper is organized as follows. Section 2 is for brief reviews on the modern perspective on the fermionization, the WZW models and their center symmetry, and the Chern-Simons theories with emphasis on one-form global symmetries and their gauging. We also summarize SUSY conditions that has to be obeyed by any supersymmetric RCFTs. In Sections 3 and 4, we present detailed analyses that show the emergence of supersymmetry for the WZW models in table 1 and 2. We examine in Section 5 if the RCFTs for the Read-Rezayi states at $\nu = 1/2, 2/3$ preserve supersymmetry. In appendix A, we argue that supersymmetry can emerge only for the $\mathbb{Z}_6$ and $\mathbb{Z}_8$ parafermion CFTs via the Jordan-Wigner transformation.

## 2 Preliminaries

### 2.1 $\mathbb{Z}_2$ Orbifold and Fermionization

We briefly review on the generalized Jordan-Wigner transformation that maps a given bosonic theory to a fermionic theory and vice versa in two dimensions [22, 20].

Let us start with a bosonic theory with non-anomalous $\mathbb{Z}_2$ symmetry, denoted by $\mathcal{B}$. Gauging the discrete $\mathbb{Z}_2$ symmetry leads to an orbifold $\tilde{\mathcal{B}} = \mathcal{B}/\mathbb{Z}_2$. The partition functions of $\mathcal{B}$ and $\tilde{\mathcal{B}}$ on a genus $g$ Riemann surface $\Sigma_g$ satisfy the relation below

$$Z_{\tilde{\mathcal{B}}}[T] = \frac{1}{2^g} \sum_{s \in H^1(\Sigma_g, \mathbb{Z}_2)} Z_{\mathcal{B}}[s] \exp\left[i\pi \int s \cup T\right], \tag{4}$$

where the sum $s$ is over all possible discrete gauge fields for the $\mathbb{Z}_2$ symmetry. It is known that the orbifold $\tilde{\mathcal{B}}$ has an emergent $\mathbb{Z}_2$ quantum symmetry, and $T \in H^1(\Sigma_g, \mathbb{Z}_2)$ is the background gauge field for the quantum symmetry. Here the cup product '$\cup$' is a product on cohomology classes which becomes the wedge product of differential forms in the case of de Rham cohomology.

One can also utilize the $\mathbb{Z}_2$ symmetry to obtain a fermionic theory $\mathcal{F}$ 'dual' to $\mathcal{B}$. To do so, one needs a non-trivial two-dimensional invertible spin topological theory, known as the Kitaev Majorana chain. Note that the theory of the Majorana chain has the $\mathbb{Z}_2$ global symmetry. The fermionization can be described as coupling $\mathcal{B}$ to the Kitaev Majorana chain, and then taking the diagonal $\mathbb{Z}_2$ orbifold. The partition functions of $\mathcal{F}$ on a genus $g$ Riemann surface with spin structure $\rho$ is related to that of $\mathcal{B}$ as follows,

$$Z_{\mathcal{F}}[T + \rho] = \frac{1}{2^g} \sum_{s \in H^1(\Sigma_g, \mathbb{Z}_2)} Z_{\mathcal{B}}[s] \exp\left[i\pi\left(\mathrm{Arf}[s + \rho] + \mathrm{Arf}[\rho] + \int s \cup T\right)\right], \tag{5}$$

where $T \in H^1(\Sigma_g, \mathbb{Z}_2)$ is the background gauge field for the fermion parity $(-1)^F$. Here $\mathrm{Arf}[\rho]$ is the so-called Arf invariant which accounts for the contribution from the Kitaev Majorana chain. The Arf invariant is a mod 2 index and becomes 1 when $\rho$ is even and 0 otherwise. For instance, when $\Sigma_g = T^2$, a choice of periodic (R) or anti-periodic (NS) boundary condition around each of the two cycles specifies a spin structure $\rho$. The Arf invariant is then given by

$$\mathrm{Arf}[\rho] = \begin{cases} 1 & \text{(R,R)} \\ 0 & \text{(NS,R), (R,NS), (NS,NS)} \end{cases}. \tag{6}$$

On the other hand, we can make use of the quantum $\mathbb{Z}_2$ symmetry to obtain a fermionic theory $\tilde{\mathcal{F}}$ from $\tilde{\mathcal{B}}$:

$$Z_{\tilde{\mathcal{F}}}[T + \rho] = \frac{1}{2^g} \sum_{s \in H^1(\Sigma_g, \mathbb{Z}_2)} Z_{\tilde{\mathcal{B}}}[s] \exp\left[i\pi\left(\mathrm{Arf}[s + \rho] + \mathrm{Arf}[\rho] + \int s \cup T\right)\right]. \tag{7}$$

Note that the map (4) further relates $Z_{\tilde{\mathcal{F}}}$ to $Z_{\mathcal{B}}$,

$$Z_{\tilde{\mathcal{F}}}[T + \rho] = \frac{1}{2^{2g}} \sum_{s,t \in H^1(\Sigma_g, \mathbb{Z}_2)} Z_{\mathcal{B}}[t] \exp\left[i\pi\left(\mathrm{Arf}[s + \rho] + \mathrm{Arf}[\rho] + \int s \cup (T + t)\right)\right]$$
$$= \frac{1}{2^g} \sum_{t \in H^1(\Sigma_g, \mathbb{Z}_2)} Z_{\mathcal{B}}[t] \exp\left[i\pi\mathrm{Arf}[(T + t + \rho)]\right], \tag{8}$$

where we applied an identity below for the last equality,

$$\frac{1}{2^g} \sum_{s \in H^1(\Sigma_g, \mathbb{Z}_2)} \exp\left[i\pi\left(\mathrm{Arf}[s + \rho] + \mathrm{Arf}[\rho] + \int s \cup t\right)\right] = \exp\left[i\pi\mathrm{Arf}[t + \rho]\right]. \tag{9}$$

Using another identity for the Arf invarint,

$$\exp\left[i\pi\mathrm{Arf}[s + t + \rho]\right] = \exp\left[i\pi\left(\mathrm{Arf}[s + \rho] + \mathrm{Arf}[t + \rho] + \mathrm{Arf}[\rho] + \int s \cup t\right)\right],$$



Table 3: A bosonic theory $\mathcal{B}$ on $S^1$ has either the untwisted or the twisted Hilbert space, depending on whether a nontrivial $\mathbb{Z}_2$ holonomy along the circle is turned off or not. Each Hilbert space can be further decomposed into the $\mathbb{Z}_2$ even and odd sectors. On the other hand, a fermionic theory $\mathcal{F}$ on a circle has either the Neveu-Schwarz or the Ramond sector, depending on the periodic condition along $S^1$. One can divide each sector into the $(-1)^F$ even and odd sectors where $F$ denotes the fermion number operator.

| $\mathcal{B}$ | untwisted | twisted |
|---|---|---|
| even | $\mathcal{H}_u^e$ | $\mathcal{H}_t^e$ |
| odd | $\mathcal{H}_u^o$ | $\mathcal{H}_t^o$ |

| $\tilde{\mathcal{B}}$ | untwisted | twisted |
|---|---|---|
| even | $\mathcal{H}_u^e$ | $\mathcal{H}_u^o$ |
| odd | $\mathcal{H}_t^e$ | $\mathcal{H}_t^o$ |

| $\mathcal{F}$ | NS sector | R sector |
|---|---|---|
| even | $\mathcal{H}_u^e$ | $\mathcal{H}_u^o$ |
| odd | $\mathcal{H}_t^o$ | $\mathcal{H}_t^e$ |

| $\tilde{\mathcal{F}}$ | NS sector | R sector |
|---|---|---|
| even | $\mathcal{H}_u^e$ | $\mathcal{H}_t^e$ |
| odd | $\mathcal{H}_t^o$ | $\mathcal{H}_u^o$ |

one can show that the fermionic theory $\tilde{\mathcal{F}}$ can be obtained by coupling $\mathcal{F}$ to the Kitaev Majorana chain

$$Z_{\tilde{\mathcal{F}}}[\rho] = Z_{\mathcal{F}}[\rho]\exp\left[i\pi\mathrm{Arf}[\rho]\right]. \tag{10}$$

For proofs of identities, see [38, 20]. The equation (10) implies that $Z_{\mathcal{F}}$ is the same as $Z_{\tilde{\mathcal{F}}}$ in any choice of the spin structure except the symmetric $\rho$ where the sign would be flipped. One can learn from the maps (4) and (5) how the Hilbert spaces of $\mathcal{B}$, $\widetilde{\mathcal{B}}$, $\mathcal{F}$, and $\widetilde{\mathcal{F}}$ are shuffled to each other, summarized in table 3

In the present work, bosonic CFTs $\mathcal{B}$ mainly refers to the WZW models on simple Lie groups. Most of WZW models have a natural $\mathbb{Z}_2$ symmetry, which is a part of the center symmetry. We utilize the $\mathbb{Z}_2$ symmetry to construct either the orbifold $\tilde{\mathcal{B}}$ or the fermionic partner $\mathcal{F}$ in what follows. In Appendix A, we consider the $\mathbb{Z}_k$ parafermion theories as an excursion beyond the WZW models.

## 2.2 Supersymmetry Conditions

We define a superconformal theory as a conformal theory having a conserved current $G(z)$ of weight $h = 3/2$ satisfying the OPEs below,

$$\begin{aligned} T(z)G(0) &\sim \frac{3}{z^2}G(0) + \frac{1}{z}\partial G(0), \\ G(z)G(0) &\sim \frac{2c}{3z^3} + \frac{1}{z}T(0), \end{aligned} \tag{11}$$

where $T(z)$ denotes the stress-energy tensor and $c$ is the central charge. The conserved current $G(z)$ is then called supersymmetry current. To search for a bosonic CFT which can be fermionized to a superconformal theory, one needs to find such an $h = 3/2$ supersymmetry current. Once we find such a bosonic theory, as explained in introduction, its chiral part has the supersymmetry current. It is rather nontrivial to show if a candidate primary of $h = 3/2$ obeys the supersymmetry current OPE (11) for an interacting theory. Instead, we discuss necessary conditions for a given bosonic RCFT to be mapped to a supersymmetric theory via the generalized Jordan-Wigner transformation (5):

1. We first require a bosonic theory $\mathcal{B}$ to have an $h = 3/2$ primary operator. This primary will play a role as a supersymmetry current after the fermionization. We also note that the NS vacuum character of the supersymemtric theory dual to $\mathcal{B}$ can be obtained by combining the vacuum character with the character for the primary of $h = 3/2$.

2. For a supersymmetric theory $\mathcal{F}$, the torus partition function in the Ramond-Ramond sector, denoted by $Z_{\mathcal{F}}^{\tilde{R}}$, becomes an index and thus constant. As a consequence, the difference between torus partition functions of $\mathcal{B}$ and $\tilde{\mathcal{B}}$ also takes the constant value,

$$Z_{\mathcal{B}}(\tau, \bar{\tau}) - Z_{\tilde{\mathcal{B}}}(\tau, \bar{\tau}) = Z_{\mathcal{F}}^{\tilde{R}}(\tau, \bar{\tau}) = const. \tag{12}$$

When the index $Z_{\mathcal{F}}^{\tilde{R}}(\tau, \bar{\tau})$ does not vanish, the corresponding theory has supersymemtric vacua. On the other hand, the index vanishes when the supersymmetry is spontaneously broken unless the the supersymmetric unitarity bound below is saturated. The $Z_{\mathcal{F}}^{\tilde{R}}$ also vanishes whenever a given theory has a free fermion regardless of whether the superymmetry is preserved. In the present work, we focus on a model having no free fermion.

3. The supersymmetric unitarity bound $h^R \geq c/24$ has to be obeyed. This is because, in the Ramond sector where $G(z) = \sum_{r \in 1/2 + \mathbb{Z}} G_r / z^{r+3/2}$, each Laurent mode $G_r$ satisfies the anti-commutation relations below,

$$\{G_r, G_s\} = 2L_{r+s} + \frac{1}{2}\left(r^2 - \frac{c}{24}\right)\delta_{r+s}. \tag{13}$$

4. Finally, the fermionic partition function in each spin structure should allow the super Virasoro character decomposition.

As an illustration, let us choose the $U(1)_{12}$ WZW model for $\mathcal{B}$. This model is the theory of a compact boson on $S^1$ with radius $2\sqrt{3}$ and has twelve primaries of conformal weights $h = n^2/24$ for $n = 0, \pm 1, .., \pm 5, 6$. Note that the $U(1)_{12}$ WZW model has a primary of $h = 3/2$ which could potentially play a role as a supersymmetry current. The model has a $\mathbb{Z}_2$ symmetry which acts on each primary as follows,

$$\mathbb{Z}_2 : \left|h = \frac{n^2}{24}\right\rangle \longrightarrow (-1)^n \cdot \left|h = \frac{n^2}{24}\right\rangle. \tag{14}$$

The torus partition functions of $\mathcal{B} = U(1)_{12}$ and the orbifold $\tilde{\mathcal{B}} = \mathcal{B}/\mathbb{Z}_2$ are given by

$$Z_{\mathcal{B}} = \left|\chi_0\right|^2 + \left|\chi_6\right|^2 + \sum_{n=1}^{5}\left(\left|\chi_n\right|^2 + \left|\chi_{-n}\right|^2\right), \tag{15}$$

$$Z_{\tilde{\mathcal{B}}} = \left|\chi_0\right|^2 + \left|\chi_6\right|^2 + \sum_{n=2,3,4}\left(\left|\chi_n\right|^2 + \left|\chi_{-n}\right|^2\right) + \left(\chi_1\bar{\chi}_5 + \chi_{-1}\bar{\chi}_{-5} + \chi_5\bar{\chi}_1 + \chi_{-5}\bar{\chi}_{-1}\right),$$

which implies that their difference is constant,

$$Z_{\mathcal{B}} - Z_{\tilde{\mathcal{B}}} = \left|\chi_1 - \chi_5\right|^2 + \left|\chi_{-1} - \chi_{-5}\right|^2 = 1^2 + 1^2. \tag{16}$$

We see that the $U(1)_{12}$ WZW model satisfies the aforementioned first and second necessary conditions, and thus it is likely that the corresponding fermionic theory is supersymmetric. To see this, we apply (5) to obtain the torus partition function of the fermionic theory $\mathcal{F}$ for each spin structure given by,

$$\begin{aligned}
Z_{\mathcal{F}}^{\text{NS}} &= \left|\chi_0 + \chi_6\right|^2 + \left|\chi_2 + \chi_4\right|^2 + \left|\chi_{-2} + \chi_{-4}\right|^2, \\
Z_{\mathcal{F}}^{\widetilde{\text{NS}}} &= \left|\chi_0 - \chi_6\right|^2 + \left|\chi_2 - \chi_4\right|^2 + \left|\chi_{-2} - \chi_{-4}\right|^2, \\
Z_{\mathcal{F}}^{\text{R}} &= \left|\chi_1 + \chi_5\right|^2 + \left|\chi_{-1} + \chi_{-5}\right|^2 + 2\left|\chi_3\right|^2 + 2\left|\chi_{-3}\right|^2, \\
Z_{\mathcal{F}}^{\tilde{R}} &= \left|\chi_1 - \chi_5\right|^2 + \left|\chi_{-1} - \chi_{-5}\right|^2 = 1^2 + 1^2.
\end{aligned} \tag{17}$$

It is straightforward to show that (17) perfectly agrees with the partition function of the $\mathcal{N}=1$ supersymmetric minimal model with $c=1$, which confirms our expectation that $\mathcal{F}$ preserves the supersymmetry. Indeed we can express (17) in terms of the $\mathcal{N}=1$ super-Virasoro characters of $c=1$.

We make a remark that $U(1)_{12}$ and its fermionic partner in the NS sector can arise as the theories of gapless edges of three-dimensional $U(1)$ Chern-Simons theories with level $k=12$ and $k=3$, which will be discussed in more detail later in this section.

## 2.3 Wess-Zumino-Witten Models

Wess-Zumino-Witten(WZW) models are non-linear sigma models with the group manifold $G$ as target space. The current algebra of WZW models is known to be described by the affine Lie algebra $\hat{G}$. In this subsection, we briefly review the WZW models and present the prime consequences.

Let us take the bosonic field $g(z,\bar{z})$ which is valued in the unitary representation of the semi-simple group $G$. The action of the WZW model has a form of

$$S = \frac{|k|}{4\pi} \int_{S^2} d^2 z \mathrm{Tr}\left[ g^{-1}\partial_\mu g g^{-1}\partial^\mu g \right] + \frac{k}{12\pi i} \int_B d^3 y \epsilon_{\alpha\beta\gamma} \mathrm{Tr}\left[ \tilde{g}^{-1}\partial^\alpha \tilde{g} \tilde{g}^{-1}\partial^\beta \tilde{g} \tilde{g}^{-1}\partial^\gamma \tilde{g} \right], \quad (18)$$

where $B$ denote the three-dimensional manifold whose boundary is the two-sphere $S^2$ and $\tilde{g}$ is an extension of the bosonic field $g(z,\bar{z})$ to the three-dimensional manifold $B$. For the path integral to be well-defined, the level $k$ should be quantized. Having constructed an action (18), one can show that the holomorphic current $J(z) = -k\partial g g^{-1}$ and the anti-holomorphic current $\bar{J}(\bar{z}) = k g^{-1}\bar{\partial} g$ are conserved separately.

The primary states of the WZW models are associated with the affine weights of $\hat{G}$, which are labeled by non-negative integers referred to as affine Dynkin labels $\hat{\mu} = (\mu_0, \mu_1, \cdots, \mu_r)$. Here, $r$ denote the rank of $\hat{G}$. More precisely, the primary fields are in correspondence with the weights in the integrable representation $P_+^k$ defined as

$$P_+^k = \left\{ \hat{\mu} \middle| \mu_j \geq 0, \quad 0 \leq \sum_{j=1}^r a_j^\vee \mu_j \leq k \right\}. \quad (19)$$

The comarks $a_j^\vee$ are combined with the Dynkin labels $\mu_j$ to produce the level $k$ as follows.

$$k = \mu_0 + \sum_{j=1}^r a_j^\vee \mu_j. \quad (20)$$

Because the physically relevant fields ought to be in the $P_+^k$, the level $k$ is given by a positive integer. WZW models possess a finite number of primary fields for a given $k$, therefore one can consider them as rational conformal field theory(RCFT).

The central charge of a given WZW model and the conformal weights of primary states can be computed via the Sugawara construction. Their explicit forms are given by

$$c = \frac{k \dim(\hat{G})}{k + h^\vee}, \qquad h_{\hat{\mu}} = \frac{(\hat{\mu}, \hat{\mu} + 2\hat{\rho})}{2(k + h^\vee)}, \quad (21)$$

where $h^\vee$ and $\hat{\rho}$ are the dual Coxeter number and affine Weyl vector, respectively.

The torus partition function of a given WZW model takes the form of

$$Z(\tau, \bar{\tau}) = \sum_{\hat{\mu}, \hat{\nu} \in P_+^k} \chi_{\hat{\mu}}(\tau) \mathcal{M}_{\hat{\mu}\hat{\nu}} \bar{\chi}_{\hat{\nu}}(\bar{\tau}). \quad (22)$$

Table 4: Outer automorphism of the affine Lie algebras are listed in this table.

| $G$ | $O(\hat{G})$ | Action of $O(\hat{G})$ |
|---|---|---|
| $A_N$ | $\mathbb{Z}_{N+1}$ | $A(\lambda_0, \lambda_1, \cdots, \lambda_{N-1}, \lambda_N) = (\lambda_N, \lambda_0, \cdots, \lambda_{N-2}, \lambda_{N-1})$ |
| $B_N$ | $\mathbb{Z}_2$ | $A(\lambda_0, \lambda_1, \cdots, \lambda_{N-1}, \lambda_N) = (\lambda_1, \lambda_0, \cdots, \lambda_{N-1}, \lambda_N)$ |
| $C_N$ | $\mathbb{Z}_2$ | $A(\lambda_0, \lambda_1, \cdots, \lambda_{N-1}, \lambda_N) = (\lambda_N, \lambda_{N-1}, \cdots, \lambda_1, \lambda_0)$ |
| $D_{2k}$ | $\mathbb{Z}_2 \times \mathbb{Z}_2$ | $A(\lambda_0, \lambda_1, \cdots, \lambda_{2k-1}, \lambda_{2k}) = (\lambda_1, \lambda_0, \lambda_2, \cdots, \lambda_{2k}, \lambda_{2k-1})$ <br> $\tilde{A}(\lambda_0, \lambda_1, \cdots, \lambda_{2k-1}, \lambda_{2k}) = (\lambda_{2k}, \lambda_{2k-1}, \lambda_{2k-2}, \cdots, \lambda_1, \lambda_0)$ |
| $D_{2k+1}$ | $\mathbb{Z}_4$ | $A(\lambda_0, \lambda_1, \cdots, \lambda_{2k}, \lambda_{2k+1}) = (\lambda_{2k}, \lambda_{2k+1}, \lambda_{2k-1}, \cdots, \lambda_1, \lambda_0)$ |
| $E_6$ | $\mathbb{Z}_3$ | $A(\lambda_0, \lambda_1, \cdots, \lambda_5, \lambda_6) = (\lambda_1, \lambda_5, \lambda_4, \lambda_3, \lambda_6, \lambda_0, \lambda_2)$ |
| $E_7$ | $\mathbb{Z}_2$ | $A(\lambda_0, \lambda_1, \cdots, \lambda_6, \lambda_7) = (\lambda_6, \lambda_5, \lambda_4, \lambda_3, \lambda_2, \lambda_1, \lambda_0, \lambda_7)$ |

When $\mathcal{M}_{\hat{\mu}\hat{\nu}} = \delta_{\hat{\mu}\hat{\nu}}$, we call $Z(\tau, \bar{\tau})$ as a diagonal partition function, otherwise we refer it as the non-diagonal partition function in what follows. The character $\chi_{\hat{\mu}}(\tau)$ of irreducible representation with the highest weight state $\hat{\mu}$ can be computed by the Weyl-Kac formula. An explicit expression for the character is given by

$$\chi_{\hat{\mu}}(\tau) = \frac{\sum_{w \in W} \epsilon(w) e^{w(\hat{\rho} + \hat{\mu}) - \hat{\rho}}}{\sum_{w \in W} \epsilon(w) e^{w(\hat{\rho}) - \hat{\rho}}}, \tag{23}$$

where $\epsilon(w)$ denote the sign of an element $w$ of the Weyl group $W$. The computation of the $q$-series of characters is done in SageMath [39].

The modular properties of characters are known as

$$\chi_{\hat{\mu}}(\tau + 1) = \sum_{\hat{\nu} \in P_+^k} T_{\hat{\mu}\hat{\nu}} \chi_{\hat{\nu}}(\tau), \quad \chi_{\hat{\mu}}(-1/\tau) = \sum_{\hat{\nu} \in P_+^k} S_{\hat{\mu}\hat{\nu}} \chi_{\hat{\nu}}(\tau), \tag{24}$$

where the modular matrices have the expressions of

$$\begin{aligned} T_{\hat{\mu}\hat{\nu}} &= \delta_{\hat{\mu}\hat{\nu}} e^{2\pi i (h_{\hat{\mu}} - \frac{c}{24})}, \\ S_{\hat{\mu}\hat{\nu}} &= N \sum_{w \in W} \epsilon(w) \exp\left[ -\frac{2\pi i}{k + h^\vee} (w(\hat{\mu} + \hat{\rho}), \hat{\nu} + \hat{\rho}) \right]. \end{aligned} \tag{25}$$

The normalization constant $N$ can be fixed by the unitary constraint of the $S$-matrix. The data of WZW models, including the list of primary and the numerical value of $S$-matrix elements, are accessible via the program kac [40].

Let us move our attention to the outer automorphism of the WZW models. We wish to construct the orbifold theory of the WZW models by employing (4). In this paper, we mostly focus on the non-anomalous $\mathbb{Z}_2$ symmetry that arise from the outer automorphism of the affine Lie algebra. More precisely, we focus on the $\mathbb{Z}_2$ subgroup of the outer automorphism. The outer automorphism $O(\hat{G})$ is defined as the quotient of symmetry groups of Lie algebra $G$ and affine algebra $\hat{G}$ which will be denoted as $D(\hat{G})$ and $D(G)$ respectively. Here, the symmetry group means the set of transformations preserving Cartan matrices. The symmetry transformation of $O(\hat{G})$ and the action of $O(\hat{G})$ to an arbitrary weight $\hat{\lambda} = (\lambda_0, \lambda_1, \cdots, \lambda_N)$ are presented in figure 1 and table 4.

It has been known that the outer automorphism $O(\hat{G})$ is isomorphic to the center of Lie group $G$. To see this, let us define a subgroup $B(G)$ of $G$ with elements:

$$b = e^{-2\pi i A\hat{\omega}_0 \cdot H}, \tag{26}$$

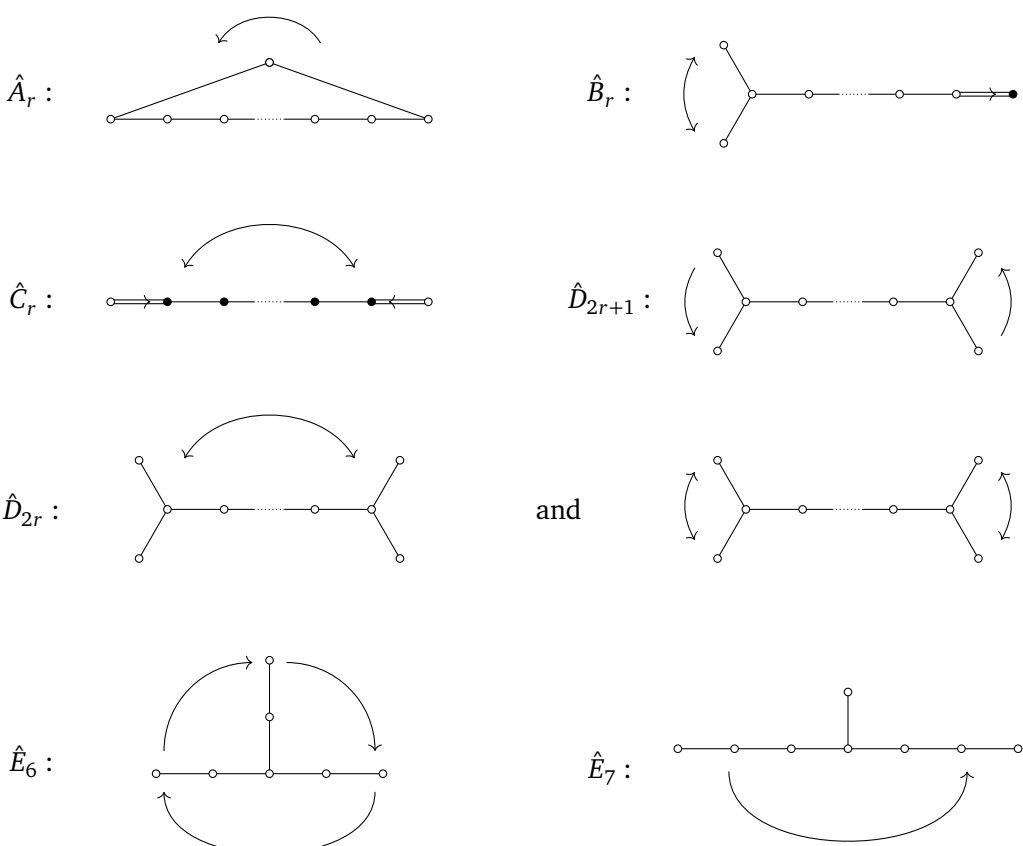

Figure 1: Affine Dynkin diagrams and their outer automorphism are presented. The white node and black node denote long root and short root respectively.

where the fundamental weight $A\hat{\omega}_0$ can be obtained by the action of the outer automorphism $A$ on the zeroth fundamental weight $\hat{\omega}_0 = (1, 0, \cdots, 0)$ (see table 4). One can show that the group element $b$ of $B(G)$ indeed commutes with the arbitrary group element of $G$[41]. The action of $b$ to the arbitrary state $|\Lambda\rangle$ is given by

$$b|\Lambda\rangle = e^{-2\pi i(A\hat{\omega}_0, \Lambda)}|\Lambda\rangle. \tag{27}$$

Let us now take the $\mathbb{Z}_2$ subgroup of the outer automorphism $O(\hat{G})$. Note that not every $O(\hat{G})$ of the affine Lie algebra admit the $\mathbb{Z}_2$ subgroup. Specifically, the outer automorphism of $\widehat{A_N}$ with $N$ odd and $\widehat{E_6}$ cannot have the $\mathbb{Z}_2$ subgroup. Except such cases, one can utilize the $\mathbb{Z}_2$ subgroup of center symmetry to construct the orbifold partition function.

As an illustrative example, let us consider the $SU(2)_6$ WZW model. This model involves seven primaries. Denoting the $SU(2)$ fundamental weight by $\omega_1$, the $SU(2)$ representations of the seven primaries are $\Lambda_j = j\omega_1$ for $j = 0, 1, .., 6$. Since the outer automorphism $O(\hat{G})$ sends $\hat{\omega}_0$ to $\hat{\omega}_1$, the phase factor of (27) becomes $(-1)^j$. In general, one can read the $\mathbb{Z}_2$ action in a similar manner for the $G_k$ WZW model as long as the outer automorphism of $\hat{G}$ has a $\mathbb{Z}_2$ subgroup.

### 2.4   Chern-Simons Theories : Bulk Descriptions

We discuss here the connection between two and three dimensional theories, often referred to as the bulk-boundary correspondence, established for two-dimensional rational CFTs and three-dimensional Chern-Simons theories.[1]  We mainly consider the WZW models as two-dimensional theories in what follows.

The Chern-Simons theory provides a macroscopic approach to understand the physics of the both integer and fractional quantum Hall states. When a given system in such a topological phase is defined on a manifold with boundaries, a certain two-dimensional chiral RCFT describes the modes that live at the edges. To be more precise, all the states of the current algebra and its representations can be recovered by quantizing the three-dimensional Chern-Simons theory on the manifold with boundaries.

It was further shown recently in [42] that each character of the current algebra can be computed by direct evaluation of the Chern-Simon path-integral in the presence of a Wilson line. This implies that one can describe the chiral primary on the boundary as the non-trivial Wilson lines allowed in the bulk. A global symmetry which acts on the edge modes can then be elevated to a one-form global symmetry under which the Wilson lines transform. We can also show that the conformal weight of a chiral primary is the same with the spin of the corresponding Wilson line.

Sometimes a global symmetry of a boundary CFT has a 't Hooft anomaly which prevents us from the gauging the symmetry in a consistent manner. For instance, let us consider an orbifold $\tilde{\mathcal{B}}$ from gauging an anomalous discrete symmetry. The modular invariance of the partition function $Z_{\tilde{\mathcal{B}}}$ could determine the spectrum in the twisted sector. However the anomalous behavior of the discrete symmetry is then reflected on the fact that such twisted sector is incompatible with the action of the symmetry.

We will explain below what the consequence of the above two-dimensional 't Hooft anomaly could be in the bulk, and discuss the conditions for non-anomalous symmetry necessary to define either an orbifold $\tilde{\mathcal{B}}$ or a fermionic theory $\mathcal{F}$ on the boundary and their bulk descriptions as well.

$U(1)_k$ **Chern-Simons theory**   Let us start with the U(1) Chern-Simons theory with level $k$,

$$S = \frac{k}{4\pi} \int A \wedge dA \,. \tag{28}$$

For even $k$ one can define the theory on any manifold. The Chern-Simons theory corresponds to the theory of a free compact chiral boson $\phi$ of radius $R = \sqrt{k}$. This is not the case for odd $k$ where the theory is defined only on spin manifolds and depends on the choice of spin structure.

When the Chern-Simons theories are applied to the fractional quantum Hall systems, one can describe the world-lines of quasi-particles as the Wilson lines.

For even $k$, we have $k$ distinct Wilson lines labelled by the $U(1)$ charge $n$,

$$W_n(C) = \exp\Big[ in \int_C A \Big], \tag{29}$$

for $n = 0, \pm 1, \pm 2, .., \pm(k-2)/2, k/2$. This is because the Wilson line $W_k$ has a trivial vacuum expectation value, and thus the lines with $n$ and $n+k$ are identical. One can also show that

---

[1]We focus on the gauge interactions but bear in mind the presence of the gravitational interactions in the correspondence.

$W_n(C)$ has spin

$$s = \frac{n^2}{2k}. \tag{30}$$

The spin of $W_n(C)$ is defined modulo one, which is reflected on the fact that the trivial Wilson line $W_k(C)$ carries integer spin $k/2$. Note that the Wilson line $W_n$ corresponds to the two-dimensional chiral primary of weight $h = n^2/(2k)$ via the bulk-boundary correspondence.

On the other hand, (30) says the Wilson line $W_k(C)$ carries half-integer spin for odd $k$. This implies that, although $W_n(C)$ and $W_{n+k}(C)$ induce the same holonomy, their spins differ by $1/2$. For odd $k$, the Wilson line labelled by $n$ can thus be identified with that by $n + 2k$, namely, there are $2k$ distinct Wilson lines $W_n(C)$ for $n = 0, \pm1, \pm2, .., \pm(k-1), k$.

The $U(1)_k$ Chern-Simons theory has a $\mathbb{Z}_k$ global one-form symmetry which acts on the $U(1)$ gauge field as follows

$$A \longrightarrow A + \frac{1}{k}\epsilon, \tag{31}$$

where the transformation parameter $\epsilon$ is closed and has integral periods

$$\int \epsilon \in 2\pi\mathbb{Z}. \tag{32}$$

It is useful to describe the symmetry transformation as an operator $U_g$ associated with a group element of the symmetry. In the case of the Chern-Simons theory, $U_g$ for the $\mathbb{Z}_k$ one-form symmetry is now associated with a one-cycle, and is given by the Wilson line

$$U_{g=e^{2\pi in/k}}(C) = W_n(C). \tag{33}$$

The Wilson lines $W_n(C')$ also provide operators charged under the one-form symmetry. More precisely, the line $W_n(C)$ rotates under (31) as follows

$$W_m(C) \longrightarrow e^{im\int_C \epsilon/k}W_m(C) = e^{\frac{2\pi inm}{k}}W_m(C), \tag{34}$$

where $\int_C \epsilon = 2\pi n$ with an integer $n$. One can also express the above transformation rule in terms of operators as

$$U_{g=e^{2\pi in/k}}(C)W_m(C') = e^{\frac{2\pi inm}{k}}W_m(C'), \tag{35}$$

where $C$ is a circle around the loop $C'$. Note also from the operator relation (35) that the generators $U_{g=e^{2\pi in/k}}(C)$ themselves are charged under the $\mathbb{Z}_k$ symmetry. As explained in [43], this implies that the $\mathbb{Z}_k$ symmetry is anomalous.

Let us now discuss how to gauge the one-form global symmetry. To this end, we restrict our attention on a non-anomalous subgroup of the $\mathbb{Z}_k$ symmetry. The generators $U_g$ of a non-anomalous symmetry are required to be neutral under the symmetry. In general, one can show from (35) that the Wilson line $W_n(C)$ satisfies

$$W_n(C)W_n(C') = e^{2\pi i(2s)}W_n(C') , \tag{36}$$

where $s = n^2/2k$ is the spin of the line and the phase can be understood as the statistical phase. Therefore, the symmetry generators $U_g(C)$ have to be chosen as the Wilson lines of either half-integer spin or integer spin. For $k = 4p$ ($p \in \mathbb{Z}$), we see the $U(1)_{4p}$ Chern-Simons

theory has the non-anomalous $\mathbb{Z}_2$ one-form symmetry, which is generated by the Wilson line $W_{n=2p}(C)$ of spin $p/2$

$$U_{g=-1}(C) = W_{n=2p}(C) = \exp\left[i(2p)\int_C A\right],$$
$$\left(U_{g=-1}(C)\right)^2 = 1. \tag{37}$$

The $\mathbb{Z}_2$ one-form global symmetry acts on various Wilson lines as

$$U_{g=-1}(C)W_n(C') = \left(-1\right)^n W_n(C'), \tag{38}$$

which is nothing but the three-dimensional uplifting of the $\mathbb{Z}_2$ action (14). Summing over all possible insertions of $U_{g=-1}(C)$ accounts for gauging the $\mathbb{Z}_2$ symmetry. A typical insertion is

$$\langle U_{g=-1}(C)\rangle = \int \mathcal{D}A \, e^{i\frac{4p}{4\pi}\int A\wedge dA + i(2p)\int A\wedge\delta(C)}$$
$$= \int \mathcal{D}B \, e^{i\frac{p}{4\pi}\int B\wedge dB + \text{(background terms)}}, \tag{39}$$

where $dB = 2dA + 2\pi\delta(C)$. We should stress that the field redefinition $B$ is allowed because its first Chern number takes an integer value. As a consequence, the $\mathbb{Z}_2$ gauging results in the $U(1)$ Chern-Simons theory with level $k = p$.

When $p$ is odd, the $U(1)_p$ Chern-Simons theory is a spin topological field theory and depends on the choice of the spin structure. From the $\mathbb{Z}_2$ gauging approach, this fact can be explained by the vacuum expectation value of the Wilson line $U_{g=-1}(C)$ that carries $h = p/2$ half-integer spin.

As a prominent example, one can take the bosonic theory $\mathcal{B}$ as the $U(1)_{12}$ Chern-Simons theory. As discussed above, the corresponding fermionic theory $\mathcal{F}$ is the $\mathcal{N} = 1$ supersymmetric minimal model. Once the $\mathbb{Z}_2$ one-form symmetry is gauged, we end up with the $U(1)_3$ Chern-Simons theory. The $U(1)_3$ Chern-Simons theory has 6 inequivalent lines labeled by

$$n = 0, \pm 1, \pm 2, 3, \tag{40}$$

whose spins are 0, 1/6, 1/6, 2/3, 2/3 and 3/2. Interestingly, these lines correspond to the primaries involved in the NS partition function (17) of the $\mathcal{N} = 1$ SUSY minimal model. This is not a coincident but the $U(1)_p$ with odd $p$ is closely related to a fermionic model associated with the $U(1)_{4p}$ Chern-Simons model in general.

**K-matrices**  One obvious generalization of the Abelian quantum Hall state is a theory of $N$ $U(1)$ gauge fields characterized by a $K$ matrix,

$$S = \frac{K_{ij}}{4\pi}\int A^i \wedge dA^j . \tag{41}$$

The gauge invariance of the action requires that all elements of the matrix $K$ have to be integers. When the diagonal elements of $K$ are even, the theory becomes independent of the spin structure. Otherwise, the theory is a spin topological field theory.

The Wilson line labeled by an $N$-vector $n_i$ can be expressed as

$$W_{\vec{n}}(C) = \exp\left[in_i\int_C A^i\right], \tag{42}$$

and carries a spin $s = (K^{ij}n_i n_j)/2$ where $K^{ij}$ is the inverse matrix of $K_{ij}$. However, not all Wilson lines are independent. This is because certain Wilson lines become trivial, and any Wilson line dressed by trivial Wilson lines can be identified with itself. For instance, when the diagonal elements of $K_{ij}$ are all even integers, there exist $N$ trivial Wilson lines,

$$W_{\vec{K}_j}(C) \equiv \exp\left[iK_{ij}\int_C A^i\right], \quad j = 1, 2, \cdots, N, \tag{43}$$

each of which induces a trivial holonomy and has an integer spin $s = K_{jj}/2$. Then each line $W_{\vec{n}}(C)$ can be identified with the Wilson lines $W_{\vec{n}'}(C)$ with $n_i' = n_i + K_{ij}p_j$ where $p_j$ are arbitrary integers. If a certain diagonal element of $K_{ij}$ is an odd integer, some of $W_{\vec{K}_j}$ have half-integral spins and cannot be ignored. Then, some of $p_j$ are constrained by even integers accordingly.

When the $K$-matrix is chosen as the Cartan matrix of the Lie algebra of a simple Lie group $G$, the Abelian Chern-Simons theory (41) can describe anyons of the non-Abelian Chern-Simons theory with the gauge group $G$ and level one $k = 1$. A typical example is the "equivalence" bewteen $U(1)_2$ and $SU(2)_1$ Chern-Simons theories. In fact, it can be viewed as the bulk realization of the Frenkel-Kac construction where the level-one WZW model on $G$ can be described as free bosons on the root-lattice of $\mathfrak{g}$ [44, 45].

**Non-Abelian Chern-Simons theory**    The non-Abelian Chern-Simons theories in turn give rise to the non-Abelian anyons. A prominent example is the $SU(2)$ Chern-Simons theory with $k = 2$ which describes the bosonic Moore-Read states[46]. It is thus plausible that the non-Abelian Chern-Simons theories provide low-energy effective theories for the non-Abelian quantum Hall states, although the full description needs more sophisticated elaborations [36, 19]. When a non-Abelian Chern-Simons theory with gauge group $G$ and level $k$ is placed on a manifold with boundaries, the modes on edges can be described by the WZW model on $G$ with level $k$.

Although it is fun to review various features of the non-Abelian Chern-Simons theories to some extent, we mainly focus on the discussion about some aspects of discrete symmetries in what follows.

As a demonstration, let us consider the $SU(2)$ Chern-Simons theory with level $k$. The Wilson lines $W_j(C)$ are now characterized by the $SU(2)$ representation $j$, and their spins $s = j(j+1)/(k+2)$. This model has the one-form $\mathbb{Z}_2$ center symmetry, generated by the Wilson line $U_{g=-1}(C)$ associated with the $SU(2)$ representation of $j = k/2$,

$$U_{g=-1}(C) \equiv W_{j=k/2}(C). \tag{44}$$

The above generator has spin $s = k/4$, and it implies that

$$U_{g=-1}(C)U_{g=-1}(C') = (-1)^k U_{g=-1}(C'), \tag{45}$$

where the loop $C$ is around the loop $C'$. The $\mathbb{Z}_2$ one-form symmetry is thus free from the 't Hooft anomaly only when $k$ is even. Otherwise, $U_{g=-1}(C)$ is charged under $\mathbb{Z}_2$ and generate an anomalous $\mathbb{Z}_2$ symmetry. On the other hand, a charged operator $W_j(C)$ rotates under $\mathbb{Z}_2$ as

$$U_{g=-1}(C)W_j(C') = (-1)^{2j} W_j(C'). \tag{46}$$

The transformation rule (46) can be also understood from the expectation value of the two linked loops in the $j$ and $j'$ representations given by

$$S_{jj'} = \sqrt{\frac{2}{k+2}} \sin\left[\frac{\pi}{k+2}(2j+1)(2j'+1)\right]. \tag{47}$$

The matrix $S$ coincides with the modular S-matrix of the corresponding $SU(2)$ WZW model with level $k$ [16]. From (47), we can see that

$$S_{\frac{k}{2}j} = (-1)^{2j} S_{0j}, \tag{48}$$

which is consistent with (46).

In general, the generator of non-anomalous $\mathbb{Z}_2$ center symmetry of a WZW model on $G$ is given by Wilson line of a representation $\mathcal{R}$ whose spin takes either an integer or a half-integer value.[2] One can then readily determine the $\mathbb{Z}_2$ action by computing the ratio of two modular $S$-matrix elements,

$$U_{g=-1}(C)W_{\mathcal{R}'}(C') = \frac{S_{\mathcal{R}\mathcal{R}'}}{S_{0\mathcal{R}'}} W_{\mathcal{R}'}(C'), \tag{49}$$

which is in perfect agreement with the action of automorphism group (27).

From the boundary point of view, a symmetry operator $U_g(C)$ of spin $s$ is related to a Verlinde line $\mathcal{L}_h$ associated with a primary of conformal weight $h = s$. The Verlinde line is a topological defect line that preserves the left and right chiral algebra [48]. It acts on a primary state $|\phi_k\rangle$ by

$$\mathcal{L}_{h'}|\phi_k\rangle = \frac{S_{i,k}}{S_{0,k}}|\phi_k\rangle. \tag{50}$$

Here, $S_{i,k}$ denote the $(i,k)$ entry of $S$-matrix where the conformal weight of $i$-th primary is given by $h = h'$. The vacuum representation is labeled by $i = 0$.

## 3 Models with Unbroken SUSY

We propose that the WZW models listed in the table 1 can be fermionized to supersymmetric theories having supersymmetric Ramond vacua. There are four different types of such WZW models. We choose one illustrative example of each type, and demonstrate explicitly that they satisfy the SUSY conditions. We put emphasis that the $SU(12)_1$, $Sp(12)_1$ and $SO(24)_1$ WZW models can be described as theories of compact bosons on the root lattices of the Lie groups $G$.

All of those models except the $SU(4)_4$ WZW model are recently shown to have $\mathcal{N} = 1$ supersymmetric vertex operator algebras [35], which supports our proposal.

**A-type** We define the A-type models as the WZW models on the special unitary groups having the primary of $h = 3/2$. The A-type involves $SU(12)_1$, $SU(6)_2$, $SU(4)_3$ and $SU(2)_6$ WZW models, as listed in table 5. Note that the models in A-type obeying the constraint $kN = 12$.

As an illustration, let us choose the $SU(12)$ WZW model with level one. We start with the diagonal modular invariant partition function of the model

$$Z_{\mathcal{B}}(\tau, \bar{\tau}) = \sum_{i=0}^{11} \left| \chi_i(\tau) \right|^2. \tag{51}$$

Here $\chi_i$ ($i = 0, 1, .., 11$) denote the characters for the primaries whose $SU(12)$ representations and conformal weights $h_i$ are summarized in table 6. The $\mathbb{Z}_{12}$ center symmetry of the model is known to be anomalous, but has the non-anomalous $\mathbb{Z}_2$ subgroup. The $\mathbb{Z}_2$ center symmetry

---

[2]The possible anomalies of $\mathbb{Z}_2$ symmetry in bosonic CFTs are carefully discussed in [47]. In particular, $\mathbb{Z}_2$ symmetry becomes anomalous when the generator has spin either $\frac{1}{4}$, or $\frac{3}{4}$ modulo 1.

Table 5: List of A-type WZW models satisfying the supersymmetry conditions after the fermionization. The four theories in this table have the non-anomalous $\mathbb{Z}_2$ symmetry, a subgroup of the center symmetry. The difference between $Z_{\mathcal{B}}$ and $Z_{\tilde{\mathcal{B}}}$ is given by a constant for each model. It suggests that their fermionic theories in the correspondence have supersymmetric Ramond vacua.

| $G_k$ | $c$ | $Z_{\tilde{\mathcal{B}}}$ | $G_k$ | $c$ | $Z_{\tilde{\mathcal{B}}}$ | $G_k$ | $c$ | $Z_{\tilde{\mathcal{B}}}$ | $G_k$ | $c$ | $Z_{\tilde{\mathcal{B}}}$ |
|---|---|---|---|---|---|---|---|---|---|---|---|
| $SU(12)_1$ | $11$ | $Z_{\mathcal{B}} - 288$ | $SU(4)_3$ | $\frac{45}{7}$ | $Z_{\mathcal{B}} - 32$ | $SU(6)_2$ | $\frac{35}{4}$ | $Z_{\mathcal{B}} - 72$ | $SU(2)_6$ | $\frac{9}{4}$ | $Z_{\mathcal{B}} - 4$ |

Table 6: $SU(12)_1$: primaries labeled by $i$ are characterized by weights $h$ and $SU(12)$ Dynkin labels. Their characters are denoted by $\chi_i(\tau)$. We highlight the primary related to the $\mathbb{Z}_2$ subgroup of center symmetry $\mathbb{Z}_{12}$.

| $i$ | $h$ | Rep | $\mathbb{Z}_2$ | $i$ | $h$ | Rep | $\mathbb{Z}_2$ |
|---|---|---|---|---|---|---|---|
| 0 | $0$ | $(1;0,0,0,0,0,0,0,0,0,0,0)$ | $+$ | 6 | $\frac{3}{2}$ | $(0;0,0,0,0,0,1,0,0,0,0,0)$ | $+$ |
| 1 | $\frac{11}{24}$ | $(0;1,0,0,0,0,0,0,0,0,0,0)$ | $-$ | 7 | $\frac{35}{24}$ | $(0;0,0,0,0,0,0,1,0,0,0,0)$ | $-$ |
| 2 | $\frac{5}{6}$ | $(0;0,1,0,0,0,0,0,0,0,0,0)$ | $+$ | 8 | $\frac{4}{3}$ | $(0;0,0,0,0,0,0,0,1,0,0,0)$ | $+$ |
| 3 | $\frac{9}{8}$ | $(0;0,0,1,0,0,0,0,0,0,0,0)$ | $-$ | 9 | $\frac{9}{8}$ | $(0;0,0,0,0,0,0,0,0,1,0,0)$ | $-$ |
| 4 | $\frac{4}{3}$ | $(0;0,0,0,1,0,0,0,0,0,0,0)$ | $+$ | 10 | $\frac{5}{6}$ | $(0;0,0,0,0,0,0,0,0,0,1,0)$ | $+$ |
| 5 | $\frac{35}{24}$ | $(0;0,0,0,0,1,0,0,0,0,0,0)$ | $-$ | 11 | $\frac{11}{24}$ | $(0;0,0,0,0,0,0,0,0,0,0,1)$ | $-$ |

is generated by the Verlinde line defect $\mathcal{L}_{h=3/2}$ associated with the primary of $h = 3/2$. This implies that the $\mathbb{Z}_2$ action on each primary of $h_i$ is given by

$$\mathbb{Z}_2 : \; \left|h_i\right\rangle \longrightarrow \mathcal{L}_{h=3/2}\left|h_i\right\rangle = \frac{S_{3/2, h_i}}{S_{0, h_i}}\left|h_i\right\rangle = (-1)^i \left|h_i\right\rangle, \tag{52}$$

which is consistent with (27). Here $S_{h_i, h_j}$ denotes $(i, j)$-entry of the modular $S$-matrix where the conformal weights of $i$ and $j$-th primaries are given by $h_i$ and $h_j$. We should stress here that the primary of $h = 3/2$ is even under $\mathbb{Z}_2$, and thus the $\mathbb{Z}_2$ center symmetry is non-anomalous.

The partition function of $\tilde{\mathcal{B}} = \mathcal{B}/\mathbb{Z}_2$ is given by summing over the discrete $\mathbb{Z}_2$ gauge backgrounds (4),

$$Z_{\tilde{\mathcal{B}}}(\tau, \bar{\tau}) = \frac{1}{2}\Big[Z_{(1,1)}(\tau, \bar{\tau}) + Z_{(g,1)}(\tau, \bar{\tau}) + Z_{(1,g)}(\tau, \bar{\tau}) + Z_{(g,g)}(\tau, \bar{\tau})\Big], \tag{53}$$

where

$$
\begin{aligned}
Z_{(1,1)}(\tau, \bar{\tau}) &\equiv Z_{\mathcal{B}}(\tau, \bar{\tau}) = \mathrm{Tr}_{\mathcal{H}_u}\Big[q^{L_0 - c/24}\bar{q}^{\bar{L}_0 - c/24}\Big], \\
Z_{(g,1)}(\tau, \bar{\tau}) &\equiv \mathrm{Tr}_{\mathcal{H}_u}\Big[\mathcal{L}_{\frac{3}{2}}q^{L_0 - c/24}\bar{q}^{\bar{L}_0 - c/24}\Big], \\
Z_{(1,g)}(\tau, \bar{\tau}) &\equiv \mathrm{Tr}_{\mathcal{H}_t}\Big[q^{L_0 - c/24}\bar{q}^{\bar{L}_0 - c/24}\Big], \\
Z_{(g,g)}(\tau, \bar{\tau}) &\equiv \mathrm{Tr}_{\mathcal{H}_t}\Big[\mathcal{L}_{\frac{3}{2}}q^{L_0 - c/24}\bar{q}^{\bar{L}_0 - c/24}\Big].
\end{aligned}
\tag{54}
$$

Here, $\mathcal{H}_u$ ($\mathcal{H}_t$) stands for the Hilbert space of $\mathcal{B}$ on $S^1$ in the untwisted (twisted) sector and $g$ is a group element of $\mathbb{Z}_2$. It is straightforward from the $\mathbb{Z}_2$ action (52) to determine $Z_{(g,1)}(\tau, \bar{\tau})$,

$$Z_{(g,1)}(\tau, \bar{\tau}) = \sum_{i \text{ even}} \left|\chi_i(\tau)\right|^2 - \sum_{i \text{ odd}} \left|\chi_i(\tau)\right|^2, \tag{55}$$

and the rests of (54) can be also readily obtained by performing modular transformations on $Z_{(g,1)}(\tau, \bar{\tau})$. To be more precise,

$$
\begin{aligned}
Z_{(1,g)}(\tau, \bar{\tau}) &= Z_{(g,1)}(-1/\tau, -1/\bar{\tau}) \\
&= \left\{ \left( \chi_0 \bar{\chi}_6 + \chi_1 \bar{\chi}_5 + \chi_2 \bar{\chi}_4 + \chi_7 \bar{\chi}_{11} + \chi_8 \bar{\chi}_{10} \right) + (\text{c.c}) \right\} + |\chi_3|^2 + |\chi_9|^2,
\end{aligned}
\tag{56}
$$

and

$$
\begin{aligned}
Z_{(g,g)}(\tau, \bar{\tau}) &= Z_{(1,g)}(\tau + 1, \bar{\tau} + 1) \\
&= \left\{ \left( -\chi_0 \bar{\chi}_6 + \chi_1 \bar{\chi}_5 - \chi_2 \bar{\chi}_4 + \chi_7 \bar{\chi}_{11} - \chi_8 \bar{\chi}_{10} \right) + (\text{c.c}) \right\} + |\chi_3|^2 + |\chi_9|^2,
\end{aligned}
\tag{57}
$$

where we used the modular $S$- and $T$-matrix (25) for the last equality of each equation. The orbifold partition function thus becomes

$$
Z_{\tilde{\mathcal{B}}}(\tau, \bar{\tau}) = \sum_{i \text{ even}} |\chi_i(\tau)|^2 + \left\{ \left( \chi_1 \bar{\chi}_5 + \chi_7 \bar{\chi}_{11} \right) + (\text{c.c}) \right\} + |\chi_3|^2 + |\chi_9|^2.
\tag{58}
$$

Since $\chi_1 - \chi_5 = 12$ and $\chi_{11} - \chi_7 = 12$, we finally see that the difference between $Z_{\mathcal{B}}$ and $Z_{\tilde{\mathcal{B}}}$ is constant

$$
Z_{\tilde{\mathcal{B}}}(\tau, \bar{\tau}) = Z_{\mathcal{B}}(\tau, \bar{\tau}) - 288.
\tag{59}
$$

It strongly suggests that the generalized Jordan-Wigner transformation could map $\mathcal{B}$ to a supersymmetric theory $\mathcal{F}$ with supersymmetric vacua.

To see this, let us apply (5) to construct the partition functions of the corresponding fermionic theory $\mathcal{F}$. One can first express the NS partition function as

$$
Z_{\mathcal{F}}^{\text{NS}}(\tau, \bar{\tau}) = \frac{1}{2} \left[ Z_{(1,1)}(\tau, \bar{\tau}) + Z_{(g,1)}(\tau, \bar{\tau}) + Z_{(1,g)}(\tau, \bar{\tau}) - Z_{(g,g)}(\tau, \bar{\tau}) \right].
\tag{60}
$$

Plugging (51), (55), (56), and (57) into (60), one obtains

$$
\begin{aligned}
Z_{\mathcal{F}}^{\text{NS}}(\tau, \bar{\tau}) &= |\chi_0 + \chi_6|^2 + |\chi_2 + \chi_4|^2 + |\chi_8 + \chi_{10}|^2 \\
&= |f_0^{\text{NS}}(\tau)|^2 + 2|f_1^{\text{NS}}(\tau)|^2,
\end{aligned}
\tag{61}
$$

where the NS-sector characters are given as

$$
\begin{aligned}
f_0^{\text{NS}}(\tau) &= \chi_0(\tau) + \chi_6(\tau) = q^{-\frac{11}{24}} \left( 1 + 143q + 924q^{\frac{3}{2}} + \cdots \right), \\
f_1^{\text{NS}}(\tau) &= \chi_2(\tau) + \chi_4(\tau) = \chi_8(\tau) + \chi_{10}(\tau) \\
&= q^{\frac{5}{6} - \frac{11}{24}} \left( 66 + 495q^{\frac{1}{2}} + 2718q + \cdots \right).
\end{aligned}
\tag{62}
$$

The above two characters (62) has appeared in [34] as the solutions of the second-order modular linear differential equation for $\Gamma_\theta$. Similarly, one can obtain the partition functions in other sectors as follows,

$$
\begin{aligned}
Z_{\mathcal{F}}^{\widetilde{\text{NS}}}(\tau, \bar{\tau}) &= \frac{1}{2} \left[ Z_{(1,1)}(\tau, \bar{\tau}) + Z_{(g,1)}(\tau, \bar{\tau}) - Z_{(1,g)}(\tau, \bar{\tau}) + Z_{(g,g)}(\tau, \bar{\tau}) \right] \\
&= |\chi_0 - \chi_6|^2 + |\chi_2 - \chi_4|^2 + |\chi_8 - \chi_{10}|^2, \\
Z_{\mathcal{F}}^{\text{R}}(\tau, \bar{\tau}) &= \frac{1}{2} \left[ Z_{(1,1)}(\tau, \bar{\tau}) - Z_{(g,1)}(\tau, \bar{\tau}) + Z_{(1,g)}(\tau, \bar{\tau}) + Z_{(g,g)}(\tau, \bar{\tau}) \right] \\
&= |\chi_1 + \chi_5|^2 + |\chi_7 + \chi_{11}|^2 + |\sqrt{2}\chi_3|^2 + |\sqrt{2}\chi_9|^2, \\
Z_{\mathcal{F}}^{\widetilde{\text{R}}}(\tau, \bar{\tau}) &= \frac{1}{2} \left[ Z_{(1,1)}(\tau, \bar{\tau}) - Z_{(g,1)}(\tau, \bar{\tau}) - Z_{(1,g)}(\tau, \bar{\tau}) - Z_{(g,g)}(\tau, \bar{\tau}) \right] \\
&= |\chi_1 - \chi_5|^2 + |\chi_7 - \chi_{11}|^2 = 12^2 + 12^2.
\end{aligned}
\tag{63}
$$

Table 7: $SU(2)_6$: primaries labeled by $i$ are characterized by weights $h$ and $SU(2)$ Dynkin labels. Their characters are denoted by $\chi_i(\tau)$. We highlight the primary related to the $\mathbb{Z}_2$ center symmetry. The partition functions for $\mathcal{F}$ in the NS and R sectors are also presented.

| $i$ | $h$ | Rep | $\mathbb{Z}_2$ | $i$ | $h$ | Rep | $\mathbb{Z}_2$ | $i$ | $h$ | Rep | $\mathbb{Z}_2$ | $i$ | $h$ | Rep | $\mathbb{Z}_2$ |
|---|---|---|---|---|---|---|---|---|---|---|---|---|---|---|---|
| 0 | 0 | (6;0) | + | 2 | $\frac{3}{32}$ | (5;1) | - | 4 | $\frac{1}{4}$ | (4;2) | + | 6 | $\frac{15}{32}$ | (3;3) | - |
| 1 | $\frac{3}{2}$ | (0;6) | + | 3 | $\frac{35}{32}$ | (1;5) | - | 5 | $\frac{3}{4}$ | (2;4) | + | | | | |

$$Z_{\mathcal{F}}^{\text{NS}} = \left|\chi_0 + \chi_1\right|^2 + \left|\chi_4 + \chi_5\right|^2, \quad Z_{\mathcal{F}}^{\text{R}} = \left|\chi_2 + \chi_3\right|^2 + \left|\sqrt{2}\chi_6\right|^2.$$

Table 8: $SU(4)_3$: primaries labeled by $i$ are characterized by weights $h$ and $SU(4)$ Dynkin labels. Their characters are denoted by $\chi_i(\tau)$. We highlight the primary related to the $\mathbb{Z}_2$ subgroup of center symmetry $\mathbb{Z}_4$. The partition functions for $\mathcal{F}$ in the NS and R sectors are also presented.

| $i$ | $h$ | Rep | $\mathbb{Z}_2$ | $i$ | $h$ | Rep | $\mathbb{Z}_2$ | $i$ | $h$ | Rep | $\mathbb{Z}_2$ | $i$ | $h$ | Rep | $\mathbb{Z}_2$ |
|---|---|---|---|---|---|---|---|---|---|---|---|---|---|---|---|
| 0 | 0 | (3;0,0,0) | + | 5 | $\frac{71}{56}$ | (0;1,2,0) | - | 10 | $\frac{9}{14}$ | (1;0,0,2) | + | 15 | $\frac{55}{56}$ | (0;1,0,2) | - |
| 1 | $\frac{9}{8}$ | (0;0,0,3) | - | 6 | $\frac{9}{14}$ | (1;2,0,0) | + | 11 | $\frac{71}{56}$ | (0;0,2,1) | - | 16 | $\frac{4}{7}$ | (1;1,0,1) | + |
| 2 | $\frac{3}{2}$ | (0;0,3,0) | + | 7 | $\frac{15}{56}$ | (2;0,0,1) | - | 12 | $\frac{6}{7}$ | (1;0,2,0) | + | 17 | $\frac{39}{56}$ | (1;0,1,1) | - |
| 3 | $\frac{9}{8}$ | (0;3,0,0) | - | 8 | $\frac{8}{7}$ | (0;2,1,0) | + | 13 | $\frac{55}{56}$ | (0;2,0,1) | - | 18 | $\frac{15}{14}$ | (0;1,1,1) | + |
| 4 | $\frac{8}{7}$ | (0;0,1,2) | + | 9 | $\frac{15}{56}$ | (2;1,0,0) | - | 14 | $\frac{5}{14}$ | (2;0,1,0) | + | 19 | $\frac{39}{56}$ | (1;1,1,0) | - |

$$Z_{\mathcal{F}}^{\text{NS}} = \left|\chi_0 + \chi_2\right|^2 + \left|\chi_4 + \chi_{10}\right|^2 + \left|\chi_6 + \chi_8\right|^2 + \left|\chi_{12} + \chi_{14}\right|^2 + \left|\chi_{16} + \chi_{18}\right|^2,$$

$$Z_{\mathcal{F}}^{\text{R}} = \left|\chi_5 + \chi_9\right|^2 + \left|\chi_7 + \chi_{11}\right|^2 + \left|\sqrt{2}\chi_1\right|^2 + \left|\sqrt{2}\chi_3\right|^2 + \left|\sqrt{2}\chi_{13}\right|^2 + \left|\sqrt{2}\chi_{15}\right|^2 + \left|\sqrt{2}\chi_{17}\right|^2 + \left|\sqrt{2}\chi_{19}\right|^2.$$

Based on the facts that the Virasoro primary of $h = 3/2$ appears as the NS vacuum descendant and $Z_{\mathcal{F}}^{\widetilde{\text{R}}}$ becomes constant, it is likely that the fermionic theory constructed from the $SU(12)_1$ WZW model is supersymmetric.

We notice that the $SU(12)_1$ WZW model can describe the Abelian quantum Hall state at filling fraction $\nu = \frac{11}{12}$ discussed in [49]. To see this, we consider an action below

$$S = \frac{1}{4\pi} K_{ij} \int A^i \wedge dA^j + \frac{1}{2\pi} \int q_i B \wedge dA^i, \tag{64}$$

where $B$ is the $U(1)$ electromagnetic background field. Here the K-matrix $K_{ij}$ is the Cartan matrix of $su(12)$ and $\vec{q} = (1, 0, 0, \cdots, 0)$. The filling fraction is then determined by

$$\nu = (K^{-1})^{ij} q_i q_j = \frac{11}{12}. \tag{65}$$

To make the discussion concise, we skip analyzing the other examples in the A-type models in details, but rather provide a few essential facts in tables 7, 8 and 9 from which one can easily construct the partition function of $\mathcal{F}$ in each spin structure.

**A′-type**  Among the WZW models on $SU(N)$, we pay special attention to the $SU(4)_4$ WZW model. This model contains three $h = 3/2$ primaries of the Dynkin labels $(0; 0, 0, 4)$, $(0; 4, 0, 0)$ and $(0; 1, 2, 1)$. The Verlinde line associated with the representation $(0; 1, 2, 1)$ does not generate a discrete symmetry. Moreover, the lines for the representations $(0; 0, 0, 4)$ and $(0; 4, 0, 0)$ generate the $\mathbb{Z}_4$ center symmetry rather than the $\mathbb{Z}_2$ symmetry. It is the line defect $\mathcal{L}_{h=1}$ that realizes the non-anomalous $\mathbb{Z}_2$ symmetry of our interest, which is a subgroup of the center symmetry.

Let us then gauge the above $\mathbb{Z}_2$ symmetry. Although the orbifold theory $\widetilde{\mathcal{B}} = SU(4)_4/\mathbb{Z}_2$ has the modular invariant partition function disobeying (12), it has not only a primary of

Table 9: $SU(6)_2$: primaries labeled by $i$ are characterized by weights $h$ and $SU(6)$ Dynkin labels. Their characters are denoted by $\chi_i(\tau)$. We highlight the primary related to the $\mathbb{Z}_2$ subgroup of center symmetry $\mathbb{Z}_6$. The partition functions for $\mathcal{F}$ in the NS and R sectors are also presented.

| $i$ | $h$ | Rep | $\mathbb{Z}_2$ | $i$ | $h$ | Rep | $\mathbb{Z}_2$ | $i$ | $h$ | Rep | $\mathbb{Z}_2$ |
|---|---|---|---|---|---|---|---|---|---|---|---|
| 0 | 0 | (2;0,0,0,0,0) | + | 7 | $\frac{33}{32}$ | (0;0,0,0,1,1) | - | 14 | $\frac{7}{12}$ | (1;0,1,0,0,0) | + |
| 1 | $\frac{5}{6}$ | (0;0,0,0,0,2) | + | 8 | $\frac{131}{96}$ | (0;0,0,1,1,0) | - | 15 | $\frac{3}{4}$ | (0;1,0,0,0,1) | + |
| 2 | $\frac{4}{3}$ | (0;0,0,0,2,0) | + | 9 | $\frac{131}{96}$ | (0;0,1,1,0,0) | - | 16 | $\frac{7}{12}$ | (1;0,0,0,1,0) | + |
| 3 | $\frac{3}{2}$ | (0;0,0,2,0,0) | + | 10 | $\frac{33}{32}$ | (0;1,1,0,0,0) | - | 17 | $\frac{13}{12}$ | (0;0,0,1,0,1) | + |
| 4 | $\frac{4}{3}$ | (0;0,2,0,0,0) | + | 11 | $\frac{35}{96}$ | (1;1,0,0,0,0) | - | 18 | $\frac{95}{96}$ | (0;1,0,0,1,0) | - |
| 5 | $\frac{5}{6}$ | (0;2,0,0,0,0) | + | 12 | $\frac{5}{4}$ | (0;0,1,0,1,0) | + | 19 | $\frac{21}{32}$ | (1;0,0,1,0,0) | - |
| 6 | $\frac{35}{96}$ | (1;0,0,0,0,1) | - | 13 | $\frac{13}{12}$ | (0;1,0,1,0,0) | + | 20 | $\frac{95}{96}$ | (0;0,1,0,0,1) | - |

$$Z_{\mathcal{F}}^{\mathrm{NS}} = \left|\chi_0 + \chi_3\right|^2 + \left|\chi_1 + \chi_2\right|^2 + \left|\chi_4 + \chi_5\right|^2 + \left|\chi_{12} + \chi_{15}\right|^2 + \left|\chi_{13} + \chi_{14}\right|^2 + \left|\chi_{16} + \chi_{17}\right|^2,$$

$$Z_{\mathcal{F}}^{\mathrm{R}} = \left|\chi_6 + \chi_8\right|^2 + \left|\chi_9 + \chi_{11}\right|^2 + \left|\chi_{18} + \chi_{20}\right|^2 + \left|\sqrt{2}\chi_7\right|^2 + \left|\sqrt{2}\chi_{19}\right|^2.$$

Table 10: $SU(4)_4/\mathbb{Z}_2$: primaries labeled by $i$ are characterized by weights $h$. Their characters $\tilde{\chi}_i$ are expressed by the characters of $SU(4)_4$ WZW model. We highlight the primary related to the $\widetilde{\mathbb{Z}}_2$ symmetry.

| $i$ | $h$ | $\tilde{\chi}_i$ | $\widetilde{\mathbb{Z}}_2$ | $i$ | $h$ | $\tilde{\chi}_i$ | $\widetilde{\mathbb{Z}}_2$ | $i$ | $h$ | $\tilde{\chi}_i$ | $\widetilde{\mathbb{Z}}_2$ |
|---|---|---|---|---|---|---|---|---|---|---|---|
| 0 | 0 | $\chi_{(4;0,0,0)} + \chi_{(0;0,4,0)}$ | + | 4 | $\frac{5}{16}$ | $\chi_{(3;0,1,0)} + \chi_{(1;0,3,0)}$ | - | 8 | $\frac{3}{4}$ | $\chi_{(2;0,2,0)}$ | + |
| 1 | $\frac{3}{2}$ | $\chi_{(0;0,0,4)} + \chi_{(0;4,0,0)}$ | + | 5 | $\frac{21}{16}$ | $\chi_{(0;1,0,3)} + \chi_{(0;3,0,1)}$ | - | 9 | $\frac{5}{4}$ | $\chi_{(0;2,0,2)}$ | + |
| 2 | $\frac{9}{16}$ | $\chi_{(2;0,0,2)} + \chi_{(0;2,2,0)}$ | - | 6 | 1 | $\chi_{(1;0,1,2)} + \chi_{(1;2,1,0)}$ | + | 10 | $\frac{15}{16}$ | $\chi_{(1;1,1,1)}$ | - |
| 3 | $\frac{25}{16}$ | $\chi_{(0;0,2,2)} + \chi_{(2;2,0,0)}$ | - | 7 | $\frac{3}{2}$ | $\chi_{(0;1,2,1)} + \chi_{(2;1,0,1)}$ | + | | | | |

$h = 3/2$ again but also a new quantum $\widetilde{\mathbb{Z}}_2$ symmetry generated by the Verlinde line $\tilde{\mathcal{L}}_{h=3/2}$ associated with the primary of $h = 3/2$. To be more precise, we present the orbifold partition function below,

$$Z_{\widetilde{\mathcal{B}}}(\tau, \bar\tau) = \sum_{i=0}^{7} \left|\tilde{\chi}_i\right|^2 + 2\left|\tilde{\chi}_8\right|^2 + 2\left|\tilde{\chi}_9\right|^2 + 2\left|\tilde{\chi}_{10}\right|^2, \tag{66}$$

where the conformal characters $\tilde{\chi}(\tau)$ for $\widetilde{\mathcal{B}}$ are defined in table 10. In addition, table 10 summarize the $\widetilde{\mathbb{Z}}_2$ action on the primaries of orbifold theory $\widetilde{\mathcal{B}}$.

Since $\widetilde{\mathbb{Z}}_2$ is non-anomalous, we can further gauge the symmetry which leads to the orbifold partition function,

$$Z_{\widetilde{\mathcal{B}}/\widetilde{\mathbb{Z}}_2}(\tau, \bar\tau) = \sum_{i=0,1,6,7} \left|\tilde{\chi}_i\right|^2 + 2 \sum_{j=8,9,10} \left|\tilde{\chi}_j\right|^2 + \left(\left(\tilde{\chi}_2\bar{\tilde{\chi}}_3 + \tilde{\chi}_4\bar{\tilde{\chi}}_5\right) + (\text{c.c})\right), \tag{67}$$

that finally differs to $Z_{\widetilde{\mathcal{B}}}(\tau, \bar\tau)$ by the constant

$$Z_{\widetilde{\mathcal{B}}}(\tau, \bar\tau) - Z_{\widetilde{\mathcal{B}}/\widetilde{\mathbb{Z}}_2}(\tau, \bar\tau) = 36. \tag{68}$$

It is then natural to expect that the orbifold $SU(4)_4/\mathbb{Z}_2$ can be fermionized to a supersymmetric theory.

Table 11: List of C-type WZW models satisfying the supersymmetry conditions after the fermionization. Each of three models in this table has a $\mathbb{Z}_2$ center symmetry. Based on the fact that $(Z_{\mathcal{B}} - Z_{\widetilde{\mathcal{B}}})$ is constant, their fermionic models are expected to have supersymmetric Ramond vacua.

| $G_k$ | $c$ | $Z_{\widetilde{\mathcal{B}}}$ | $G_k$ | $c$ | $Z_{\widetilde{\mathcal{B}}}$ | $G_k$ | $c$ | $Z_{\widetilde{\mathcal{B}}}$ |
|---|---|---|---|---|---|---|---|---|
| $Sp(4)_3$ | 5 | $Z_{\mathcal{B}} - 16$ | $Sp(6)_2$ | 7 | $Z_{\mathcal{B}} - 36$ | $Sp(12)_1$ | $\frac{39}{4}$ | $Z_{\mathcal{B}} - 144$ |

Table 12: $Sp(6)_2$: primaries labeled by $i$ are characterized by weights $h$ and $Sp(6)$ Dynkin labels. Their characters are denoted by $\chi_i(\tau)$. We highlight the primary related to the center symmetry $\mathbb{Z}_2$.

| $i$ | $h$ | Rep | $\mathbb{Z}_2$ | $i$ | $h$ | Rep | $\mathbb{Z}_2$ | $i$ | $h$ | Rep | $\mathbb{Z}_2$ | $i$ | $h$ | Rep | $\mathbb{Z}_2$ |
|---|---|---|---|---|---|---|---|---|---|---|---|---|---|---|---|
| 0 | 0 | (2;0,0,0) | + | 3 | $\frac{1}{2}$ | (1;0,1,0) | + | 6 | $\frac{7}{24}$ | (1;1,0,0) | - | 9 | $\frac{7}{8}$ | (0;1,1,0) | - |
| 1 | $\frac{3}{2}$ | (0;0,0,2) | + | 4 | 1 | (0;1,0,1) | + | 7 | $\frac{7}{6}$ | (0;0,2,0) | + | | | | |
| 2 | $\frac{5}{8}$ | (1;0,0,1) | - | 5 | $\frac{31}{24}$ | (0;0,1,1) | - | 8 | $\frac{2}{3}$ | (0;2,0,0) | + | | | | |

Indeed (5) gives the fermionic partition functions below

$$
\begin{aligned}
Z_{\mathcal{F}}^{\text{NS}} &= \left|\tilde{\chi}_0 + \tilde{\chi}_1\right|^2 + \left|\tilde{\chi}_6 + \tilde{\chi}_7\right|^2 + 2\left|\tilde{\chi}_8 + \tilde{\chi}_9\right|^2, \\
Z_{\mathcal{F}}^{\widetilde{\text{NS}}} &= \left|\tilde{\chi}_0 - \tilde{\chi}_1\right|^2 + \left|\tilde{\chi}_6 - \tilde{\chi}_7\right|^2 + 2\left|\tilde{\chi}_8 - \tilde{\chi}_9\right|^2, \\
Z_{\mathcal{F}}^{\text{R}} &= \left|\tilde{\chi}_2 + \tilde{\chi}_3\right|^2 + \left|\tilde{\chi}_4 + \tilde{\chi}_5\right|^2 + \left|\sqrt{2}\tilde{\chi}_{10}\right|^2, \\
Z_{\mathcal{F}}^{\widetilde{\text{R}}} &= \left|\tilde{\chi}_2 - \tilde{\chi}_3\right|^2 + \left|\tilde{\chi}_4 - \tilde{\chi}_5\right|^2 = 6^2 + 0^2.
\end{aligned}
\tag{69}
$$

We can see that the NS vacuum has the primary of $h = 3/2$ as a descendant, which is able to play a potential role as the supersymmetry current. Based on the fact that $Z_{\mathcal{F}}^{\widetilde{R}}(\tau, \bar{\tau})$ becomes constant, we also see that there are non-trivial cancellations between bosonic and fermionic contributions beyond the vacuum. Since the model under study has no free fermion, it could be explained by the existence of the supersymmetry.

**C-type**   As presented in table 11, the C-type includes the WZW models on the symplectic group $Sp(2N)$ that obey the supersymmetric conditions after fermionization. They are $Sp(4)_3$, $Sp(6)_2$, and $Sp(12)_1$ WZW models, all of which have a primary of $h = 3/2$. The rank and the level of the each C-type model are also constrained by a relation $kN = 6$.

We choose the $Sp(6)_2$ WZW model to discuss the fermionization in details. The modular invariant of partition function of this model is

$$
Z_{\mathcal{B}}(\tau, \bar{\tau}) = \sum_{i=0}^{9} \left|\chi_i(\tau)\right|^2,
\tag{70}
$$

where $\chi_i$ are the conformal characters for the primaries of $h_i$. See table 12 where the group representation and the conformal weight $h_i$ of each primary are summarized.

The model has the $\mathbb{Z}_2$ center symmetry, generated by the Verlinde line defect $\mathcal{L}_{h=3/2}$ associated with the primary of $h = 3/2$. One find the transformation rules for ten primaries of the model under $\mathbb{Z}_2$ in table 12.

Let us consider the orbifold $\widetilde{\mathcal{B}} = \mathcal{B}/\mathbb{Z}_2$. Based on the $\mathbb{Z}_2$ actions and modular matrices, one

Table 13: $Sp(4)_3$: primaries labeled by $i$ are characterized by weights $h$ and $Sp(4)$ Dynkin labels. Their characters are denoted by $\chi_i(\tau)$. We highlight the primary related to the center symmetry $\mathbb{Z}_2$.

| $i$ | $h$ | Rep | $\mathbb{Z}_2$ | $i$ | $h$ | Rep | $\mathbb{Z}_2$ | $i$ | $h$ | Rep | $\mathbb{Z}_2$ | $i$ | $h$ | Rep | $\mathbb{Z}_2$ |
|---|---|---|---|---|---|---|---|---|---|---|---|---|---|---|---|
| 0 | 0 | (3;0,0) | + | 3 | $\frac{5}{6}$ | (1;0,2) | + | 6 | $\frac{5}{8}$ | (1;1,1) | - | 9 | $\frac{7}{8}$ | (0;3,0) | - |
| 1 | $\frac{3}{2}$ | (0;0,3) | + | 4 | $\frac{5}{24}$ | (2;1,0) | - | 7 | $\frac{1}{2}$ | (1;2,0) | + | | | | |
| 2 | $\frac{1}{3}$ | (2;0,1) | + | 5 | $\frac{29}{24}$ | (0;1,2) | - | 8 | 1 | (0;2,1) | + | | | | |

$$Z_{\mathcal{F}}^{\text{NS}} = \left|\chi_0 + \chi_1\right|^2 + \left|\chi_2 + \chi_3\right|^2 + \left|\chi_7 + \chi_8\right|^2, \quad Z_{\mathcal{F}}^{\text{R}} = \left|\chi_4 + \chi_5\right|^2 + \left|\sqrt{2}\chi_6\right|^2 + \left|\sqrt{2}\chi_9\right|^2.$$

Table 14: $Sp(12)_1$: primaries labeled by $i$ are characterized by weights $h$ and $Sp(12)$ Dynkin labels. Their characters are denoted by $\chi_i(\tau)$. We highlight the primary related to the center symmetry $\mathbb{Z}_2$. The partition functions for $\mathcal{F}$ in the NS and R sectors are also presented.

| $i$ | $h$ | Rep | $\mathbb{Z}_2$ | $i$ | $h$ | Rep | $\mathbb{Z}_2$ | $i$ | $h$ | Rep | $\mathbb{Z}_2$ |
|---|---|---|---|---|---|---|---|---|---|---|---|
| 0 | 0 | (1;0,0,0,0,0,0) | + | 3 | $\frac{13}{32}$ | (0;1,0,0,0,0,0) | - | 6 | $\frac{33}{32}$ | (0;0,0,1,0,0,0) | - |
| 1 | $\frac{3}{2}$ | (0;0,0,0,0,0,1) | + | 4 | $\frac{5}{4}$ | (0;0,0,0,1,0,0) | + | | | | |
| 2 | $\frac{45}{32}$ | (0;0,0,0,0,1,0) | - | 5 | $\frac{3}{4}$ | (0;0,1,0,0,0,0) | + | | | | |

$$Z_{\mathcal{F}}^{\text{NS}} = \left|\chi_0 + \chi_1\right|^2 + \left|\chi_4 + \chi_5\right|^2, \quad Z_{\mathcal{F}}^{\text{R}} = \left|\chi_2 + \chi_3\right|^2 + \left|\sqrt{2}\chi_6\right|^2.$$

can compute various twisted partition functions

$$
\begin{aligned}
Z_{(g,1)} &= \sum_{i=0,1,3,4,7,8} \left|\chi_i\right|^2 - \sum_{i=2,5,6,9} \left|\chi_i\right|^2, \\
Z_{(1,g)} &= \left|\chi_2\right|^2 + \left|\chi_9\right|^2 + \left\{(\chi_0\bar{\chi}_1 + \chi_3\bar{\chi}_4 + \chi_5\bar{\chi}_6 + \chi_7\bar{\chi}_8) + (\text{c.c})\right\}, \\
Z_{(g,g)} &= \left|\chi_2\right|^2 + \left|\chi_9\right|^2 + \left\{(-\chi_0\bar{\chi}_1 - \chi_3\bar{\chi}_4 + \chi_5\bar{\chi}_6 - \chi_7\bar{\chi}_8) + (\text{c.c})\right\},
\end{aligned}
\tag{71}
$$

which leads to the orbifold partition function below

$$
Z_{\widetilde{\mathcal{B}}} = \sum_{i=0}^{4} \left|\chi_i\right|^2 + \sum_{i=7}^{9} \left|\chi_i\right|^2 + \left(\chi_5\bar{\chi}_6 + \text{c.c}\right).
\tag{72}
$$

Since $\chi_5 - \chi_6 = -6$, we can see that $\mathbb{Z}_{\mathcal{B}}$ and $\mathbb{Z}_{\widetilde{\mathcal{B}}}$ differ by a constant 36. It is then natural to expect that a fermionic model dual to the model of our interests respects the supersymmetry.

Applying the fermionization with use of (71), we indeed verify that

$$
\begin{aligned}
Z_{\mathcal{F}}^{\text{NS}} &= \left|\chi_0 + \chi_1\right|^2 + \left|\chi_3 + \chi_4\right|^2 + \left|\chi_7 + \chi_8\right|^2, \\
Z_{\mathcal{F}}^{\widetilde{\text{NS}}} &= \left|\chi_0 - \chi_1\right|^2 + \left|\chi_3 - \chi_4\right|^2 + \left|\chi_7 - \chi_8\right|^2, \\
Z_{\mathcal{F}}^{\text{R}} &= \left|\chi_5 + \chi_6\right|^2 + \left|\sqrt{2}\chi_2\right|^2 + \left|\sqrt{2}\chi_9\right|^2, \quad Z_{\mathcal{F}}^{\widetilde{\text{R}}} = \left|\chi_5 - \chi_6\right|^2 = 6^2.
\end{aligned}
\tag{73}
$$

As proposed, the model has the $h = 3/2$ Virasoro primary as a vacuum descendant, and the $\mathbb{Z}_{\mathcal{F}}^{\widetilde{\text{R}}}$ becomes an index, namely constant.

We do not repeat the same exercise for other C-type WZW models listed in table 11. Instead, we provide a summary of how fermionization computes the partition functions of $\mathcal{F}$ in the four spin structures in table 13 and 14.

Table 15: List of D-type WZW models satisfying the supersymmetry conditions after the fermionization. Each has a non-anomalous $\mathbb{Z}_2$ symmetry, a subgroup of center symmetry. Since $Z_\mathcal{B}$ and $Z_{\tilde{\mathcal{B}}}$ differ by constant, we expect that their fermionic theories have supersymmetric vacua in the Ramond sector.

| $G_k$ | $c$ | $Z_{\tilde{\mathcal{B}}}$ | $G_k$ | $c$ | $Z_{\tilde{\mathcal{B}}}$ | $G_k$ | $c$ | $Z_{\tilde{\mathcal{B}}}$ |
|---|---|---|---|---|---|---|---|---|
| $SO(8)_3$ | $\frac{28}{3}$ | $Z_\mathcal{B} - 128$ | $SO(12)_2$ | 11 | $Z_\mathcal{B} - 144$ | $SO(24)_1$ | 12 | $Z_\mathcal{B} - 576$ |

Table 16: $SO(12)_2$: primaries labeled by $i$ are characterized by weights $h$ and $SO(12)$ Dynkin labels. Their characters are denoted by $\chi_i(\tau)$. Primaries $i = 1, 2, 3$ are the generator of $\mathbb{Z}_2^A, \mathbb{Z}_2^B, \mathbb{Z}_2^C$, respectively. We highlight the primary related to the $\mathbb{Z}_2$ subgroup of the center symmetry $\mathbb{Z}_2 \times \mathbb{Z}_2$.

| $i$ | $h$ | Rep | $\mathbb{Z}_2^A$ | $\mathbb{Z}_2^B$ | $\mathbb{Z}_2^C$ | $i$ | $h$ | Rep | $\mathbb{Z}_2^A$ | $\mathbb{Z}_2^B$ | $\mathbb{Z}_2^C$ |
|---|---|---|---|---|---|---|---|---|---|---|---|
| 0 | 0 | (2;0,0,0,0,0) | + | + | + | 7 | $\frac{19}{16}$ | (0;1,0,0,0,1) | + | - | - |
| 1 | $\frac{3}{2}$ | (0;0,0,0,0,2) | + | + | + | 8 | $\frac{35}{24}$ | (0;0,0,0,1,1) | - | + | - |
| 2 | 1 | (0;2,0,0,0,0) | + | + | + | 9 | $\frac{11}{24}$ | (1;1,0,0,0,0) | - | + | - |
| 3 | $\frac{3}{2}$ | (0;0,0,0,2,0) | + | + | + | 10 | $\frac{4}{3}$ | (0;0,0,1,0,0) | + | + | + |
| 4 | $\frac{11}{16}$ | (1;0,0,0,0,1) | - | - | + | 11 | $\frac{5}{6}$ | (0;1,0,0,0,0) | + | + | + |
| 5 | $\frac{19}{16}$ | (0;1,0,0,0,1,0) | - | - | + | 12 | $\frac{9}{8}$ | (0;0,0,1,0,0) | - | + | - |
| 6 | $\frac{11}{16}$ | (1;0,0,0,1,0) | + | - | - | | | | | | |

**D-type** From table 15, the D-type models refer to the $SO(2N)_k$ WZW models containing $h = \frac{3}{2}$ primary. One can see that $SO(24)_1$, $SO(12)_2$ and $SO(8)_3$ WZW models are the member of D-type where the rank and the level are constrained by the condition $kN = 12$. We choose the $SO(12)_2$ WZW model as an interesting example of the D-type models, and verify that it has all the features proposed above.

The model of our interests is a rational CFT with $c = 11$ and has 13 primaries whose $SO(12)$ representations and conformal weights can be found in table 16. Note that there are three candidates generating $\mathbb{Z}_2$ symmetry. They are Verlinde defects associated with the primaries of $h = 3/2$ and $h = 1$. Former and latter generate the $\mathbb{Z}_2^A$ and $\mathbb{Z}_2^B$ symmetries whose actions on 13 primaries are presented in table 16.

One can show that gauging the $\mathbb{Z}_2^B$ results in the non-diagonal modular invariant partition function,

$$Z_{\mathcal{B}/\mathbb{Z}_2^B}(\tau, \bar\tau) = \left|\chi_0 + \chi_2\right|^2 + \left|\chi_1 + \chi_3\right|^2 + 2\sum_{i=8}^{12}\left|\chi_i\right|^2, \tag{74}$$

which agrees with the diagonal modular invariant partition function of the $SU(12)_1$ WZW model (51). The conformal embedding [50] can explain the above coincidence.

Let us now move on the the orbifold $\tilde{\mathcal{B}} = \mathcal{B}/\mathbb{Z}_2^A$. From the $\mathbb{Z}_2^A$ action in table 16, one obtains

$$
\begin{aligned}
Z_{(1,1)}(\tau, \bar\tau) &= \sum_{i=0}^{12}\left|\chi_i(\tau)\right|^2, \quad Z_{(g,1)}(\tau, \bar\tau) = \sum_{i\in\mathcal{A}^+}\left|\chi_i(\tau)\right|^2 - \sum_{i\in\mathcal{A}^-}\left|\chi_i(\tau)\right|^2, \\
Z_{(1,g)}(\tau, \bar\tau) &= \left\{\left(\chi_0\bar\chi_1 + \chi_2\bar\chi_3 + \chi_6\bar\chi_7 + \chi_8\bar\chi_9 + \chi_{10}\bar\chi_{11}\right) + (\text{c.c})\right\} \\
&\quad + \left|\chi_4(\tau)\right|^2 + \left|\chi_5(\tau)\right|^2 + \left|\chi_{12}(\tau)\right|^2, \\
Z_{(g,g)}(\tau, \bar\tau) &= \left\{\left(-\chi_0\bar\chi_1 - \chi_2\bar\chi_3 - \chi_6\bar\chi_7 + \chi_8\bar\chi_9 - \chi_{10}\bar\chi_{11}\right) + (\text{c.c})\right\} \\
&\quad + \left|\chi_4(\tau)\right|^2 + \left|\chi_5(\tau)\right|^2 + \left|\chi_{12}(\tau)\right|^2,
\end{aligned}
\tag{75}
$$

where $\mathcal{A}^+ = \{0, 1, 2, 3, 6, 7, 10, 11\}$ and $\mathcal{A}^- = \{4, 5, 8, 9, 12\}$.

Table 17: $SO(24)_1$: primaries labeled by $i$ are characterized by weights $h$ and $SO(24)$ Dynkin labels. Their characters are denoted by $\chi_i(\tau)$. Primaries $i = 1, 2, 3$ are the generator of $\mathbb{Z}_2^A, \mathbb{Z}_2^B, \mathbb{Z}_2^C$, respectively. We highlight the primary related to the $\mathbb{Z}_2$ subgroup of the center symmetry $\mathbb{Z}_2 \times \mathbb{Z}_2$. The partition functions for $\mathcal{F}$ in the NS and R sectors are also presented.

| $i$ | $h$ | Rep | $\mathbb{Z}_2^A$ | $\mathbb{Z}_2^B$ | $\mathbb{Z}_2^C$ | $i$ | $h$ | Rep | $\mathbb{Z}_2^A$ | $\mathbb{Z}_2^B$ | $\mathbb{Z}_2^C$ |
|---|---|---|---|---|---|---|---|---|---|---|---|
| 0 | 0 | (1;0,0,0,0,0,0,0,0,0,0,0,0) | + | + | + | 2 | $\frac{1}{2}$ | (0;1,0,0,0,0,0,0,0,0,0,0,0) | - | + | - |
| 1 | $\frac{3}{2}$ | (0;0,0,0,0,0,0,0,0,0,0,0,1) | + | - | - | 3 | $\frac{3}{2}$ | (0;0,0,0,0,0,0,0,0,0,0,1,0) | - | - | + |

$$Z_{\mathcal{F}}^{\text{NS}} = \left|\chi_0 + \chi_1\right|^2, \quad Z_{\mathcal{F}}^{\text{R}} = \left|\chi_2 + \chi_3\right|^2.$$

The partition function of $\widetilde{\mathcal{B}}$ is then given by

$$Z_{\widetilde{\mathcal{B}}}(\tau, \bar{\tau}) = \sum_{i=0}^{7} \left|\chi_i(\tau)\right|^2 + \sum_{i=10}^{12} \left|\chi_i(\tau)\right|^2 + \chi_8 \bar{\chi}_9 + \chi_9 \bar{\chi}_8, \tag{76}$$

which differs from $Z_{\mathcal{B}}$ by a constant value 144. This is because $\chi_8 - \chi_9 = -12$. We thus expect that (5) with $\mathbb{Z}_2^A$ would transform the $SO(12)_2$ WZW model to a fermionic theory that preserves the supersymmetry.

Indeed, applying the generalized Jordan-Wigner transformations, one can see from the partition functions in four spin structures below

$$\begin{aligned}
Z_{\mathcal{F}}^{\text{NS}}(\tau, \bar{\tau}) &= \left|\chi_0 + \chi_1\right|^2 + \left|\chi_2 + \chi_3\right|^2 + \left|\chi_6 + \chi_7\right|^2 + \left|\chi_{10} + \chi_{11}\right|^2, \\
Z_{\mathcal{F}}^{\widetilde{\text{NS}}}(\tau, \bar{\tau}) &= \left|\chi_0 - \chi_1\right|^2 + \left|\chi_2 - \chi_3\right|^2 + \left|\chi_6 - \chi_7\right|^2 + \left|\chi_{10} - \chi_{11}\right|^2, \\
Z_{\mathcal{F}}^{\text{R}}(\tau, \bar{\tau}) &= \left|\chi_8 + \chi_9\right|^2 + \left|\sqrt{2}\chi_4\right|^2 + \left|\sqrt{2}\chi_5\right|^2 + \left|\sqrt{2}\chi_{12}\right|^2, \\
Z_{\mathcal{F}}^{\widetilde{\text{R}}}(\tau, \bar{\tau}) &= \left|\chi_8 - \chi_9\right|^2 = 12^2,
\end{aligned} \tag{77}$$

that a candidate for the supersymmetry current appears a descendant of the vacuum, and that the partition function $Z_{\tilde{R}}$ is constant. Thus we propose that the $SO(12)_2$ WZW model can fermionize to a supersymmetric theory. It is consistent with the recent result in [35] that this model has the $\mathcal{N} = 1$ SVOA.

We avoid doing the same exercise for the $SO(24)_1$ WZW models, but present useful facts in table 17 to study the fermionic models dual to them. As a final remark, it turns out that the $SO(24)_1$ WZW model can be fermionized to the well-known model exhibiting the $Co_0$ moonshine phenomenon [31, 51].

$SO(N)_3$**-type**   It is easy to show that every WZW model on the $SO(N)$ group has a primary of $h = 3/2$ when the level $k = 3$. We can further propose that the $SO(N)_3$ WZW models can be mapped to supersymmetric models via the fermionization.

Let us illustrate $SO(8)_3$ WZW model as a representative example. The model under study has 24 primaries whose $SO(8)$ representations and the conformal weights are presented in table 18. Note that the model has $\mathbb{Z}_2^A \times \mathbb{Z}_2^B \times \mathbb{Z}_2^C$ symmetry, each of which can be generated by the Verlinde lines associated with three $h = 3/2$ primaries. Gauging any of these $\mathbb{Z}_2$ symmetries would lead to essentially the same orbifold, which manifests the triality of $SO(8)$.

Table 18: $SO(8)_3$: primaries labeled by $i$ are characterized by weights $h$ and $SO(8)$ Dynkin labels. Their characters are denoted by $\chi_i(\tau)$. Primaries $i = 1, 2, 3$ are the generator of $\mathbb{Z}_2^A, \mathbb{Z}_2^B, \mathbb{Z}_2^C$, respectively. We highlight the primary related to the $\mathbb{Z}_2$ subgroup of the center symmetry $\mathbb{Z}_2 \times \mathbb{Z}_2$.

| $i$ | $h$ | Rep | $\mathbb{Z}_2^A$ | $\mathbb{Z}_2^B$ | $\mathbb{Z}_2^C$ | $i$ | $h$ | Rep | $\mathbb{Z}_2^A$ | $\mathbb{Z}_2^B$ | $\mathbb{Z}_2^C$ |
|---|---|---|---|---|---|---|---|---|---|---|---|
| 0 | 0 | (3;0,0,0,0) | + | + | + | 12 | $\frac{4}{3}$ | (0;1,0,1,1) | + | + | + |
| 1 | $\frac{3}{2}$ | (0;0,0,0,3) | + | - | - | 13 | $\frac{5}{6}$ | (1;1,0,1,0) | + | - | - |
| 2 | $\frac{3}{2}$ | (0;3,0,0,0) | - | + | - | 14 | $\frac{5}{6}$ | (1;0,0,1,1) | - | + | - |
| 3 | $\frac{3}{2}$ | (0;0,0,3,0) | - | - | + | 15 | $\frac{5}{6}$ | (1;1,0,0,1) | - | - | + |
| 4 | $\frac{8}{9}$ | (1;0,0,0,2) | + | + | + | 16 | $\frac{8}{9}$ | (1;2,0,0,0) | + | + | + |
| 5 | $\frac{7}{18}$ | (2;0,0,0,1) | + | - | - | 17 | $\frac{25}{18}$ | (0;0,0,2,1) | + | - | - |
| 6 | $\frac{25}{18}$ | (0;1,0,2,0) | - | + | - | 18 | $\frac{7}{18}$ | (2;1,0,0,0) | - | + | - |
| 7 | $\frac{25}{18}$ | (0;2,0,1,0) | - | - | + | 19 | $\frac{25}{18}$ | (0;0,0,1,2) | - | - | + |
| 8 | $\frac{8}{9}$ | (1;0,0,2,0) | + | + | + | 20 | $\frac{2}{3}$ | (1;0,1,0,0) | + | + | + |
| 9 | $\frac{25}{18}$ | (0;2,0,0,1) | + | - | - | 21 | $\frac{7}{6}$ | (0;0,1,0,1) | + | - | - |
| 10 | $\frac{25}{18}$ | (0;1,0,0,2) | - | + | - | 22 | $\frac{7}{6}$ | (0;1,1,0,0) | - | + | - |
| 11 | $\frac{7}{18}$ | (2;0,0,1,0) | - | - | + | 23 | $\frac{7}{6}$ | (0;0,1,1,0) | - | - | + |

Choosing the $\mathbb{Z}_2^B$ to construct the orbifold $\widetilde{\mathcal{B}}$. Based on the $\mathbb{Z}_2^B$ action, one can show that

$$
\begin{aligned}
Z_{(g,1)}(\tau, \bar{\tau}) &= \sum_{i \text{ even}} |\chi_i|^2 - \sum_{i \text{ odd}} |\chi_i|^2, \\
Z_{(1,g)}(\tau, \bar{\tau}) &= \sum_{\substack{i=0,4,8, \\ 12,16,20}} (\chi_i \bar{\chi}_{i+2} + \text{c.c}) + \sum_{\substack{i=1,5,9, \\ 13,17,21}} (\chi_i \bar{\chi}_{i+2} + \text{c.c}), \\
Z_{(g,g)}(\tau, \bar{\tau}) &= -\sum_{\substack{i=0,4,8, \\ 12,16,20}} (\chi_i \bar{\chi}_{i+2} + \text{c.c}) + \sum_{\substack{i=1,5,9, \\ 13,17,21}} (\chi_i \bar{\chi}_{i+2} + \text{c.c}).
\end{aligned}
\tag{78}
$$

It implies that the orbifold partition function $Z_{\widetilde{\mathcal{B}}}(\tau, \bar{\tau})$ becomes

$$
Z_{\widetilde{\mathcal{B}}}(\tau, \bar{\tau}) = \sum_{i \text{ even}} |\chi_i|^2 + \sum_{\substack{i=1,5,9, \\ 13,17,21}} (\chi_i \bar{\chi}_{i+2} + \text{c.c}),
\tag{79}
$$

which disagree with $Z_{\mathcal{B}}(\tau, \bar{\tau})$ only by a constant,

$$
Z_{\mathcal{B}}(\tau, \bar{\tau}) - Z_{\widetilde{\mathcal{B}}}(\tau, \bar{\tau}) = 128.
\tag{80}
$$

Thus, the $SO(8)_3$ WZW model satisfies the SUSY conditions.

One can verify that, in the fermionic model $\mathcal{F}$ mapped by the Jordan-Wigner transformation, the $h = 3/2$ primary behaves as the supersymmetry current, and that the partition

Table 19: Five WZW models related to spontaneously broken supersymemtric RCFTs.

| $G_k$ | $c$ | $G_k$ | $c$ | $G_k$ | $c$ | $G_k$ | $c$ | $G_k$ | $c$ |
|---|---|---|---|---|---|---|---|---|---|
| $(E_7)_2$ | 133/10 | $(E_8)_2$ | 13/2 | $SU(8)_2/\mathbb{Z}_2$ | 63/5 | $SO(16)_2/\mathbb{Z}_2$ | 15 | $SU(16)_1/\mathbb{Z}_2$ | 15 |

Table 20: $(E_7)_2$: primaries labeled by $i$ are characterized by weights $h$ and $E_7$ Dynkin labels. Their characters are denoted by $\chi_i(\tau)$. We highlight the primary related to the center symmetry $\mathbb{Z}_2$.

| $i$ | $h$ | Rep | $\mathbb{Z}_2$ | $i$ | $h$ | Rep | $\mathbb{Z}_2$ | $i$ | $h$ | Rep | $\mathbb{Z}_2$ |
|---|---|---|---|---|---|---|---|---|---|---|---|
| 0 | 0 | (2;0,0,0,0,0,0,0) | + | 2 | $\frac{21}{16}$ | (0;0,0,0,0,0,0,1) | - | 4 | $\frac{7}{5}$ | (0;0,0,0,0,1,0,0) | + |
| 1 | $\frac{3}{2}$ | (0;0,0,0,0,0,2,0) | + | 3 | $\frac{57}{80}$ | (1;0,0,0,0,0,1,0) | - | 5 | $\frac{9}{10}$ | (0;1,0,0,0,0,0,0) | + |

function in the Ramond-Ramond sector becomes constant:

$$
\begin{aligned}
Z_{\mathcal{F}}^{\text{NS}}(\tau,\bar{\tau}) ={} & \left|\chi_0+\chi_2\right|^2 + \left|\chi_4+\chi_6\right|^2 + \left|\chi_8+\chi_{10}\right|^2 + \left|\chi_{12}+\chi_{14}\right|^2 \\
& + \left|\chi_{16}+\chi_{18}\right|^2 + \left|\chi_{20}+\chi_{22}\right|^2, \\
Z_{\mathcal{F}}^{\widetilde{\text{NS}}}(\tau,\bar{\tau}) ={} & \left|\chi_0-\chi_2\right|^2 + \left|\chi_4-\chi_6\right|^2 + \left|\chi_8-\chi_{10}\right|^2 + \left|\chi_{12}-\chi_{14}\right|^2 \\
& + \left|\chi_{16}-\chi_{18}\right|^2 + \left|\chi_{20}-\chi_{22}\right|^2, \\
Z_{\mathcal{F}}^{\text{R}}(\tau,\bar{\tau}) ={} & \left|\chi_1+\chi_3\right|^2 + \left|\chi_5+\chi_7\right|^2 + \left|\chi_9+\chi_{11}\right|^2 + \left|\chi_{13}+\chi_{15}\right|^2 \\
& + \left|\chi_{17}+\chi_{19}\right|^2 + \left|\chi_{21}+\chi_{23}\right|^2, \\
Z_{\mathcal{F}}^{\widetilde{\text{R}}}(\tau,\bar{\tau}) ={} & \left|\chi_1-\chi_3\right|^2 + \left|\chi_5-\chi_7\right|^2 + \left|\chi_9-\chi_{11}\right|^2 + \left|\chi_{13}-\chi_{15}\right|^2 \\
& + \left|\chi_{17}-\chi_{19}\right|^2 + \left|\chi_{21}-\chi_{23}\right|^2 = 0^2+8^2+8^2+0^2+0^2+0^2.
\end{aligned}
\tag{81}
$$

These results strongly suggest that the fermionic model $\mathcal{F}$ corresponding to the $SO(8)_3$ WZW model preserves the supersymmetry.

Other models in this type basically share the same features without the triality, and we skip the detailed analysis for simplicity.

# 4 Models with Spontaneously Broken SUSY

The goal of this section is to present further examples of fermionic RCFTs satisfying SUSY conditions. Specifically, we propose that the five models in table 19 can satisfy the SUSY conditions after fermionization. The common feature of the five fermionized theories is $Z_{\mathcal{F}}^{\widetilde{R}}(\tau,\bar{\tau}) = 0$, unlike the models studied in the previous section. This implies that the supersymmetry is spontaneously broken in those models.

**E-type**  E-type models refer to the $(E_7)_2$ and $(E_8)_2$ WZW models. Both of them has an non-anomalous $\mathbb{Z}_2$ symmetry.

For the $(E_7)_2$ WZW models, the $\mathbb{Z}_2$ symmetry can be identified as the outer automorphism of the affine $e_7$ algebra, or equivalently the center symmetry, and is generated by the Verlinde line associated with a primary of $h = 3/2$. We can find the $\mathbb{Z}_2$ action on each primary of the model in table 20. It is straightforward to compute the $\mathbb{Z}_2$ orbifold partition function $Z_{\widetilde{\mathcal{B}}}$. It turns out that $Z_{\widetilde{\mathcal{B}}}$ agrees with $Z_{\mathcal{B}}$, namely, the orbifold $\widetilde{\mathcal{B}}$ returns back to the original model $\mathcal{B}$. This eventually results in a vanishing $Z_{\mathcal{F}}^{\widetilde{R}}$. To see this, we use (5) to compute the partition

Table 21: $(E_8)_2$: primaries labeled by $i$ are characterized by weights $h$ and $E_8$ Dynkin labels. Their characters are denoted by $\chi_i(\tau)$. We highlight the primary related to the center symmetry $\mathbb{Z}_2$. The partition functions for $\mathcal{F}$ in the NS and R sectors are also presented.

| $i$ | $h$ | Rep | $\mathbb{Z}_2$ | $i$ | $h$ | Rep | $\mathbb{Z}_2$ | $i$ | $h$ | Rep | $\mathbb{Z}_2$ |
|---|---|---|---|---|---|---|---|---|---|---|---|
| 0 | 0 | (2;0,0,0,0,0,0,0,0) | $+$ | 1 | $\frac{15}{16}$ | (0;0,0,0,0,0,0,1,0) | $-$ | 2 | $\frac{3}{2}$ | (0;1,0,0,0,0,0,0,0) | $+$ |

function for $\mathcal{F}$ in each spin structure,

$$Z_{\mathcal{F}}^{\mathrm{NS}} = \left| \chi_0 + \chi_1 \right|^2 + \left| \chi_4 + \chi_5 \right|^2, \quad Z_{\mathcal{F}}^{\widetilde{\mathrm{NS}}} = \left| \chi_0 - \chi_1 \right|^2 + \left| \chi_4 - \chi_5 \right|^2,$$
$$Z_{\mathcal{F}}^{\mathrm{R}} = \left| \sqrt{2}\chi_2 \right|^2 + \left| \sqrt{2}\chi_3 \right|^2, \quad Z_{\mathcal{F}}^{\widetilde{\mathrm{R}}} = 0. \tag{82}$$

As expected, they satisfy the SUSY conditions and $Z_{\mathcal{F}}^{\widetilde{\mathrm{R}}} = 0$.

Let us move onto the WZW model for $(E_8)_2$. The Verlinde line related to a primary of $h = \frac{3}{2}$ provides the $\mathbb{Z}_2$ action on each primary as presented in table 21. In contrast to $E_7$ case, one cannot identify the $\mathbb{Z}_2$ symmetry as the center symmetry. Nevertheless we proceed to apply the generalized Jordan-Wigner transformation with the above $\mathbb{Z}_2$ symmetry, and obtain the partition functions for $\mathcal{F}$ below,

$$Z_{\mathcal{F}}^{\mathrm{NS}} = \left| \chi_0 + \chi_2 \right|^2, \quad Z_{\mathcal{F}}^{\widetilde{\mathrm{NS}}} = \left| \chi_0 - \chi_2 \right|^2, \quad Z_{\mathcal{F}}^{\mathrm{R}} = \left| \sqrt{2}\chi_1 \right|^2, \quad Z_{\mathcal{F}}^{\widetilde{\mathrm{R}}} = 0. \tag{83}$$

We see that all SUSY conditions are satisfied. Because any primary in the R-sector cannot saturate the unitarity bound $h^{\mathrm{R}} \geq \frac{c}{24}$, the Ramond vacua break the supersymmetry spontaneously.

As a remark, the partition functions (83) can be expressed in terms of the elliptic theta functions and Dedekind eta function. More explicitly, one can show

$$\chi_0 + \chi_2 = \left( \frac{\vartheta_3}{\eta} \right)^{\frac{31}{2}} - 31 \left( \frac{\vartheta_3}{\eta} \right)^{\frac{7}{2}},$$
$$\chi_0 - \chi_2 = \left( \frac{\vartheta_4}{\eta} \right)^{\frac{31}{2}} + 31 \left( \frac{\vartheta_4}{\eta} \right)^{\frac{7}{2}}, \tag{84}$$
$$\sqrt{2}\chi_1 = \left( \frac{\vartheta_2}{\eta} \right)^{\frac{31}{2}} + 31 \left( \frac{\vartheta_2}{\eta} \right)^{\frac{7}{2}}.$$

Therefore, the NS-sector partition function (83) can be identified with the single-character solution of the fermionic modular differential equation [34].

**Orbifold type** We start from the WZW models for $SO(16)_2$, $SU(16)_1$ and $SU(8)_2$. Analogous to the $SU(4)_4$ WZW model, all of them have a primary of $h = 3/2$ for which the Verlinde line cannot generate a $\mathbb{Z}_2$ symmetry. Instead, it is a line defect associated with the $h = 2$ primary that generates a $\mathbb{Z}_2$ symmetry. One can then construct the $\mathbb{Z}_2$ orbifold theory $\tilde{\mathcal{B}}$ by employing (4). The orbifold partition functions have a form of non-diagonal modular invariant, and the chiral symmetry is enhanced by the weight-two primary.

Let us demonstrate the details of the WZW model for $SU(8)_2$. The $h = 2$ primary, labeled by $i = 4$ in table 22, is now associated with the $\mathbb{Z}_2$ generator. It is easy to show that the partition function of orbifold theory $\tilde{\mathcal{B}} = SU(8)_2/\mathbb{Z}_2$ is given by,

$$Z_{\tilde{\mathcal{B}}} = \left| \chi_0 + \chi_4 \right|^2 + \left| \chi_2 + \chi_6 \right|^2 + \left| \chi_{16} + \chi_{20} \right|^2 + \left| \chi_{18} + \chi_{22} \right|^2 + 2\left| \chi_{32} \right|^2 + 2\left| \chi_{34} \right|^2$$
$$+ \left| \chi_1 + \chi_5 \right|^2 + \left| \chi_3 + \chi_7 \right|^2 + \left| \chi_{17} + \chi_{21} \right|^2 + \left| \chi_{19} + \chi_{23} \right|^2 + 2\left| \chi_{33} \right|^2 + 2\left| \chi_{35} \right|^2, \tag{85}$$

Using the modular $S$-matrix of $\tilde{\mathcal{B}}$, which is obtained via the block-diagonalization, we can see how the quantum $\widetilde{\mathbb{Z}}_2$ symmetry of the orbifold theory $\tilde{\mathcal{B}}$ acts on each primary (see table

Table 22: $SU(8)_2$: primaries labeled by $i$ are characterized by weights $h$ and $SU(8)$ Dynkin labels. Their characters are denoted by $\chi_i(\tau)$. We highlight the primary related to the center symmetry $\mathbb{Z}_2$.

| $i$ | $h$ | Rep | $\mathbb{Z}_2$ | $i$ | $h$ | Rep | $\mathbb{Z}_2$ | $i$ | $h$ | Rep | $\mathbb{Z}_2$ |
|---|---|---|---|---|---|---|---|---|---|---|---|
| 0 | 0 | (2;0,0,0,0,0,0) | + | 12 | $\frac{303}{160}$ | (0;0,0,1,1,0,0) | − | 24 | $\frac{51}{32}$ | (0;0,1,0,0,1,0,0) | − |
| 1 | $\frac{7}{8}$ | (0;0,0,0,0,0,2) | + | 13 | $\frac{263}{160}$ | (0;0,1,1,0,0,0) | − | 25 | $\frac{43}{32}$ | (0;1,0,0,1,0,0) | − |
| 2 | $\frac{3}{2}$ | (0;0,0,0,0,2,0) | + | 14 | $\frac{183}{160}$ | (0;1,1,0,0,0,0) | − | 26 | $\frac{27}{32}$ | (1;0,0,1,0,0,0) | − |
| 3 | $\frac{15}{8}$ | (0;0,0,0,2,0,0) | + | 15 | $\frac{63}{160}$ | (1;0,0,0,0,0,1) | − | 27 | $\frac{35}{32}$ | (0;0,1,0,0,0,1) | − |
| 4 | 2 | (0;0,0,0,2,0,0,0) | + | 16 | $\frac{9}{5}$ | (0;0,0,1,0,1,0,0) | + | 28 | $\frac{35}{32}$ | (0;1,0,0,0,0,1,0) | − |
| 5 | $\frac{15}{8}$ | (0;0,0,2,0,0,0) | + | 17 | $\frac{67}{40}$ | (0;0,1,0,1,0,0) | + | 29 | $\frac{27}{32}$ | (1;0,0,0,0,1,0,0) | − |
| 6 | $\frac{3}{2}$ | (0;0,2,0,0,0,0) | + | 18 | $\frac{13}{10}$ | (0;1,0,1,0,0,0) | + | 30 | $\frac{43}{32}$ | (0;0,0,0,1,0,0,1) | − |
| 7 | $\frac{7}{8}$ | (0;2,0,0,0,0,0) | + | 19 | $\frac{27}{40}$ | (1;0,1,0,0,0,0) | + | 31 | $\frac{51}{32}$ | (0;0,1,0,0,1,0) | − |
| 8 | $\frac{63}{160}$ | (1;0,0,0,0,0,1) | − | 20 | $\frac{4}{5}$ | (0;1,0,0,0,0,1) | + | 32 | $\frac{7}{5}$ | (0;0,1,0,0,1,0,0) | + |
| 9 | $\frac{183}{160}$ | (0;0,0,0,0,1,1) | − | 21 | $\frac{27}{40}$ | (1;0,0,0,0,1,0) | + | 33 | $\frac{51}{40}$ | (0;1,0,0,1,0,0) | + |
| 10 | $\frac{263}{160}$ | (0;0,0,0,1,1,0) | − | 22 | $\frac{13}{10}$ | (0;0,0,0,1,0,1) | + | 34 | $\frac{9}{10}$ | (1;0,0,1,0,0,0) | + |
| 11 | $\frac{303}{160}$ | (0;0,0,1,1,0,0) | − | 23 | $\frac{67}{40}$ | (0;0,0,1,0,1,0) | + | 35 | $\frac{51}{40}$ | (0;0,1,0,0,0,1) | + |

Table 23: $SU(8)_2/\mathbb{Z}_2$: primaries labeled by $i$ are characterized by weights $h$. Their characters $\tilde{\chi}_i(\tau)$ are expressed in terms of the characters of $SU(8)_2$ WZW model (see table 22). We highlight the primary related to the $\widetilde{\mathbb{Z}}_2$ symmetry.

| $i$ | $h$ | $\tilde{\chi}_i$ | $\widetilde{\mathbb{Z}}_2$ | $i$ | $h$ | $\tilde{\chi}_i$ | $\widetilde{\mathbb{Z}}_2$ | $i$ | $h$ | $\tilde{\chi}_i$ | $\widetilde{\mathbb{Z}}_2$ | $i$ | $h$ | $\tilde{\chi}_i$ | $\widetilde{\mathbb{Z}}_2$ |
|---|---|---|---|---|---|---|---|---|---|---|---|---|---|---|---|
| 0 | 0 | $\chi_0 + \chi_4$ | + | 3 | $\frac{15}{8}$ | $\chi_3 + \chi_7$ | − | 6 | $\frac{13}{10}$ | $\chi_{18} + \chi_{22}$ | + | 9 | $\frac{51}{40}$ | $\chi_{33}$ | − |
| 1 | $\frac{7}{8}$ | $\chi_1 + \chi_5$ | − | 4 | $\frac{9}{5}$ | $\chi_{16} + \chi_{20}$ | + | 7 | $\frac{27}{40}$ | $\chi_{19} + \chi_{23}$ | − | 10 | $\frac{9}{10}$ | $\chi_{34}$ | + |
| 2 | $\frac{3}{2}$ | $\chi_2 + \chi_6$ | + | 5 | $\frac{67}{40}$ | $\chi_{17} + \chi_{21}$ | − | 8 | $\frac{7}{5}$ | $\chi_{32}$ | + | 11 | $\frac{51}{40}$ | $\chi_{35}$ | − |

23). Applying the generalized Jordan-Wigner transformation to (85) with the quantum $\widetilde{\mathbb{Z}}_2$ symmetry, we can analyze the fermionic partition function with each spin structure. Explicitly, we find

$$
\begin{aligned}
Z_{\mathcal{F}}^{\text{NS}} &= \left|\chi_0 + \chi_4 + \chi_2 + \chi_6\right|^2 + \left|\chi_{16} + \chi_{20} + \chi_{18} + \chi_{22}\right|^2 + 2\left|\chi_{32} + \chi_{34}\right|^2, \\
Z_{\mathcal{F}}^{\widetilde{\text{NS}}} &= \left|\chi_0 + \chi_4 - \chi_2 - \chi_6\right|^2 + \left|\chi_{16} + \chi_{20} - \chi_{18} - \chi_{22}\right|^2 + 2\left|\chi_{32} - \chi_{34}\right|^2, \\
Z_{\mathcal{F}}^{\text{R}} &= \left|\chi_1 + \chi_5 + \chi_3 + \chi_7\right|^2 + \left|\chi_{17} + \chi_{21} + \chi_{19} + \chi_{23}\right|^2 + 2\left|\chi_{33} + \chi_{35}\right|^2, \\
Z_{\mathcal{F}}^{\widetilde{\text{R}}} &= \left|\chi_1 + \chi_5 - \chi_3 - \chi_7\right|^2 + \left|\chi_{17} + \chi_{21} - \chi_{19} - \chi_{23}\right|^2 + 2\left|\chi_{33} - \chi_{35}\right|^2 = 0,
\end{aligned}
\tag{86}
$$

which indeed satisfy the SUSY conditions. Moreover, none of the primaries involved in the Ramond partition function $Z_{\mathcal{F}}^{\text{R}}$ saturate the supersymmetric unitarity bound $h^{\text{R}} \geq \frac{21}{40}$. These results strongly suggest that the orbifold $SU(8)_2/\mathbb{Z}_2$ can be fermionized to a spontaneously broken supersymmetric model.

Our next model of interest is the $SU(16)_1$ WZW model. The 16 primaries of the model are labeled by $i = 0, 1, \cdots, 15$, and their conformal weights and $SU(16)$ representations are summarized in table 24. Although the center symmetry $\mathbb{Z}_{16}$ itself is anomalous [52, 53, 54], its $\mathbb{Z}_2$ subgroup turns out to be free from the 't Hooft anomaly. The generator of $\mathbb{Z}_2$ symmetry can be identified as the Verlinde line associated with the primary of $h = 2$. The partition function of the orbifold theory $\widetilde{\mathcal{B}} = SU(16)_1/\mathbb{Z}_2$ then becomes

$$
Z_{\widetilde{\mathcal{B}}} = \left|\chi_0 + \chi_8\right|^2 + \left|\chi_4 + \chi_{12}\right|^2 + \left|\chi_2 + \chi_{10}\right|^2 + \left|\chi_6 + \chi_{14}\right|^2.
\tag{87}
$$

To search for a supersymmetric model, we start with the orbifold theory $\widetilde{\mathcal{B}}$. The theory $\widetilde{\mathcal{B}}$ has a $\widetilde{\mathbb{Z}}_2$ global symmetry. One can read off the $\widetilde{\mathbb{Z}}_2$ action from the Verlinde line associated with

Table 24: $SU(16)_1$: primaries labeled by $i$ are characterized by weights $h$ and $SU(16)$ Dynkin labels. Their characters are denoted by $\chi_i(\tau)$. We highlight the primary related to the $\mathbb{Z}_2$ subgroup of the center symmetry $\mathbb{Z}_{16}$.

| $i$ | $h$ | Rep | $\mathbb{Z}_2$ | $i$ | $h$ | Rep | $\mathbb{Z}_2$ |
|---|---|---|---|---|---|---|---|
| 0 | 0 | (1;0,0,0,0,0,0,0,0,0,0,0,0,0,0,0) | + | 8 | 2 | (0;0,0,0,0,0,0,0,1,0,0,0,0,0,0,0) | + |
| 1 | $\frac{15}{32}$ | (0;0,0,0,0,0,0,0,0,0,0,0,0,0,0,1) | - | 9 | $\frac{63}{32}$ | (0;0,0,0,0,0,0,1,0,0,0,0,0,0,0,0) | - |
| 2 | $\frac{7}{8}$ | (0;0,0,0,0,0,0,0,0,0,0,0,0,0,1,0) | + | 10 | $\frac{15}{8}$ | (0;0,0,0,0,0,1,0,0,0,0,0,0,0,0,0) | + |
| 3 | $\frac{39}{32}$ | (0;0,0,0,0,0,0,0,0,0,0,0,0,1,0,0) | - | 11 | $\frac{55}{32}$ | (0;0,0,0,0,1,0,0,0,0,0,0,0,0,0,0) | - |
| 4 | $\frac{3}{2}$ | (0;0,0,0,0,0,0,0,0,0,0,0,1,0,0,0) | + | 12 | $\frac{3}{2}$ | (0;0,0,0,1,0,0,0,0,0,0,0,0,0,0,0) | + |
| 5 | $\frac{55}{32}$ | (0;0,0,0,0,0,0,0,0,0,0,1,0,0,0,0) | - | 13 | $\frac{39}{32}$ | (0;0,0,1,0,0,0,0,0,0,0,0,0,0,0,0) | - |
| 6 | $\frac{15}{8}$ | (0;0,0,0,0,0,0,0,0,0,1,0,0,0,0,0) | + | 14 | $\frac{7}{8}$ | (0;0,1,0,0,0,0,0,0,0,0,0,0,0,0,0) | + |
| 7 | $\frac{63}{32}$ | (0;0,0,0,0,0,0,0,0,1,0,0,0,0,0,0) | - | 15 | $\frac{15}{32}$ | (0;1,0,0,0,0,0,0,0,0,0,0,0,0,0,0) | - |

Table 25: $SU(16)_1/\mathbb{Z}_2$: primaries labeled by $i$ are characterized by weights $h$. Their characters $\tilde{\chi}_i(\tau)$ are expressed in terms of the characters $\chi_i$ of the $SU(16)_1$ WZW model (see table 24). We highlight the primary related to the $\widetilde{\mathbb{Z}}_2$ symmetry.

| $i$ | $h$ | $\tilde{\chi}_i$ | $\widetilde{\mathbb{Z}}_2$ | $i$ | $h$ | $\tilde{\chi}_i$ | $\widetilde{\mathbb{Z}}_2$ | $i$ | $h$ | $\tilde{\chi}_i$ | $\widetilde{\mathbb{Z}}_2$ | $i$ | $h$ | $\tilde{\chi}_i$ | $\widetilde{\mathbb{Z}}_2$ |
|---|---|---|---|---|---|---|---|---|---|---|---|---|---|---|---|
| 0 | 0 | $\chi_0 + \chi_8$ | + | 1 | $\frac{7}{8}$ | $\chi_2 + \chi_{10}$ | - | 2 | $\frac{3}{2}$ | $\chi_4 + \chi_{12}$ | + | 3 | $\frac{7}{8}$ | $\chi_6 + \chi_{14}$ | - |

the $h = 3/2$ primary, as presented in table 25. The fermionization then gives the fermionic partition functions

$$
\begin{aligned}
Z_{\mathcal{F}}^{\text{NS}} &= \left|\chi_0 + \chi_8 + \chi_4 + \chi_{12}\right|^2, \quad Z_{\mathcal{F}}^{\widetilde{\text{NS}}} = \left|\chi_0 + \chi_8 - \chi_4 - \chi_{12}\right|^2, \\
Z_{\mathcal{F}}^{\text{R}} &= \left|\chi_2 + \chi_{10} + \chi_6 + \chi_{14}\right|^2, \quad Z_{\mathcal{F}}^{\widetilde{\text{R}}} = \left|\chi_2 + \chi_{10} - \chi_6 - \chi_{14}\right|^2 = 0.
\end{aligned}
\tag{88}
$$

It turns out that the NS vacuum contains the spin-3/2 current, and furthermore the $Z_{\mathcal{F}}^{\widetilde{\text{R}}}$ vanishes. Because the SUSY conditions are all obeyed, (88) can be considered as the partition functions of a supersymmetric RCFT. Note that the alternative expressions for (88) are given by

$$
\begin{aligned}
\chi_0 + \chi_8 + \chi_4 + \chi_{12} &= \left(\frac{\vartheta_3}{\eta}\right)^{15} - 30\left(\frac{\vartheta_3}{\eta}\right)^3, \\
\chi_0 + \chi_8 - \chi_4 - \chi_{12} &= \left(\frac{\vartheta_4}{\eta}\right)^{15} + 30\left(\frac{\vartheta_4}{\eta}\right)^3, \\
\chi_2 + \chi_{10} + \chi_6 + \chi_{14} &= \left(\frac{\vartheta_2}{\eta}\right)^{15} + 30\left(\frac{\vartheta_2}{\eta}\right)^3,
\end{aligned}
\tag{89}
$$

from which one can identify the partition functions (88) as the solution of the first order fermionic modular differential equation [34].

The WZW model for $SO(16)_2$ involves 15 primaries whose quantum numbers are summarized in table 26. There are three Verlinde lines related to the $\mathbb{Z}_2$ symmetry. We denote them as $\mathbb{Z}_2^A, \mathbb{Z}_2^B$ and $\mathbb{Z}_2^C$ which are associated with the primaries labeled by $i = 1, 2, 3$ respectively. Only two of them are independent generators, and they can be identified with the center symmetry $\mathbb{Z}_2 \times \mathbb{Z}_2$ of $SO(16)$. However, the primary of $h = 3/2$ is associated to none of non-anomalous discrete symmetries.

Let us consider an orbifold with $\mathbb{Z}_2^A$ generated by the Verlinde line $\mathcal{L}_{h=2}$. We choose the primary of $i = 1$ for an illustration, but the other choice, i.e., $i = 3$ would lead to essentially

Table 26: $SO(16)_2$: primaries labeled by $i$ are characterized by weights $h$ and $SO(16)$ Dynkin labels. Their characters are denoted by $\chi_i(\tau)$. Primaries $i=1,2,3$ are the generator of $\mathbb{Z}_2^A, \mathbb{Z}_2^B, \mathbb{Z}_2^C$, respectively. We highlight the primary related to the $\mathbb{Z}_2$ subgroup of the center symmetry $\mathbb{Z}_2 \times \mathbb{Z}_2$.

| $i$ | $h$ | Rep | $\mathbb{Z}_2^A$ | $\mathbb{Z}_2^B$ | $\mathbb{Z}_2^C$ | $i$ | $h$ | Rep | $\mathbb{Z}_2^A$ | $\mathbb{Z}_2^B$ | $\mathbb{Z}_2^C$ |
|---|---|---|---|---|---|---|---|---|---|---|---|
| 0 | 0 | (2;0,0,0,0,0,0,0) | + | + | + | 8 | $\frac{63}{32}$ | (0;0,0,0,0,0,1,1) | - | + | - |
| 1 | 2 | (0;0,0,0,0,0,0,2) | + | + | + | 9 | $\frac{15}{32}$ | (1;1,0,0,0,0,0,0) | - | + | - |
| 2 | 1 | (0;2,0,0,0,0,0,0) | + | + | + | 10 | $\frac{15}{8}$ | (0;0,0,0,0,1,0,0) | + | + | + |
| 3 | 2 | (0;0,0,0,0,0,2,0) | + | + | + | 11 | $\frac{7}{8}$ | (0;0,1,0,0,0,0,0) | + | + | + |
| 4 | $\frac{15}{16}$ | (1;0,0,0,0,0,0,1) | + | - | - | 12 | $\frac{55}{32}$ | (0;0,0,0,1,0,0,0) | - | + | - |
| 5 | $\frac{23}{16}$ | (0;1,0,0,0,0,1,0) | + | - | - | 13 | $\frac{39}{32}$ | (0;0,0,1,0,0,0,0) | - | + | - |
| 6 | $\frac{15}{16}$ | (1;0,0,0,0,0,1,0) | - | - | + | 14 | $\frac{3}{2}$ | (0;0,0,1,0,0,0,0) | + | + | + |
| 7 | $\frac{23}{16}$ | (0;1,0,0,0,0,0,1) | - | - | + | | | | | | |

the same result. A straightforward computation shows that the $\mathbb{Z}_2^A$ orbifold partition function reads

$$Z_{SO(16)_2/\mathbb{Z}_2^A} = \left|\chi_0 + \chi_1\right|^2 + \left|\chi_2 + \chi_3\right|^2 + \left|\chi_{10} + \chi_{11}\right|^2 + 2\left|\chi_{14}\right|^2 + 2\left|\chi_4\right|^2 + 2\left|\chi_5\right|^2. \tag{90}$$

We remark that the orbifold theory $\widetilde{\mathcal{B}}^A \equiv SO(16)_2/\mathbb{Z}_2^A$ has an non-anomalous $\widetilde{\mathbb{Z}}_2^A$ symmetry which is now generated by the Verlinde line $\mathcal{L}_{h=1}$ associated with the primary of $h=1$. The first four terms of (90) are even under $\widetilde{\mathbb{Z}}_2^A$ and the other terms are odd. One can further apply the transformation (4) with $\widetilde{\mathbb{Z}}^A$ to arrive at a non-diagonal partition function,

$$Z_{\widetilde{\mathcal{B}}^A/\widetilde{\mathbb{Z}}_2^A} = \left|\chi_0 + \chi_1 + \chi_2 + \chi_3\right|^2 + 2\left|\chi_{10} + \chi_{11}\right|^2 + \left|2\chi_{14}\right|^2. \tag{91}$$

We return to the diagonal partition function of the WZW model for $SO(16)_2$ and take an orbifolding with $\mathbb{Z}_2^B$ symmetry generated by the Verlinde line $\mathcal{L}_{h=1}$. As a consequence, we find a non-diagonal invariant

$$Z_{\mathcal{B}/\mathbb{Z}_2^B} = \left|\chi_0 + \chi_2\right|^2 + \left|\chi_1 + \chi_3\right|^2 + 2\sum_{i=8}^{14}\left|\chi_i\right|^2. \tag{92}$$

The orbifold theory $\widetilde{\mathcal{B}}^B \equiv \mathcal{B}/\mathbb{Z}_2^B$ possesses the non-anomalous $\widetilde{\mathbb{Z}}_2^B$ symmetry associated with the $h=2$ primary. An orbifold with $\widetilde{\mathbb{Z}}_2^B$ symmetry provides the partition function

$$Z_{\widetilde{\mathcal{B}}^B/\widetilde{\mathbb{Z}}_2^B} = \left|\chi_0 + \chi_1 + \chi_2 + \chi_3\right|^2 + 2\left|\chi_{10} + \chi_{11}\right|^2 + \left|2\chi_{14}\right|^2. \tag{93}$$

and this process reproduces the orbifold partition function (91).

We remark that the orbifold partition functions (91) and (93) exactly agree with (87). Therefore, the supersymmetric nature of $SO(16)_2$ WZW model is identical to that of the $SU(16)_1$ WZW model.

**Remarks** So far, we explore the fermionic RCFTs that satisfy the SUSY conditions. Those fermionic RCFTs arise as a consequence of the orbifold or fermionization with respect to a non-anomalous $\mathbb{Z}_2$ symmetry of the WZW model.

Here we point out that the supersymmetric RCFTs are not always obtained by orbifold or fermionization with help of an non-anomalous $\mathbb{Z}_2$ symmetry. A prominent example is the so-called non-BPS solution of the fermionic second-order modular differential equation at $c = \frac{39}{2}$ found in [34]. Two independent solutions in both NS and R-sectors at $c = \frac{39}{2}$ were shown to be

expressed in terms of the $(E_6)_4$ characters. Using those solutions, we can construct partition functions of a supersymmetric theory. However the partition function in each sector cannot be tied with any discrete $\mathbb{Z}_2$ symmetries of $(E_6)_4$ WZW model.

A similar phenomenon happens for the $Sp(14)_2$, $SU(10)_2$ and $Sp(10)_1$ WZW models. Using the block-diagonalization of the modular S-matrix, one can construct the partition functions for their corresponding fermionic models $\mathcal{F}$ satisfying the SUSY conditions. However, it is not clear if those partition functions for $\mathcal{F}$ are originated from the $\mathbb{Z}_2$ symmetry of the WZW models.

## 5 Comments on the Read-Rezayi States

In this section, we explore the possibility of emergent supersymmetry on the edges of the generalized Pfaffian Hall states proposed by Read and Rezayi [15]. It is shown in [55] that a product of the $U(1)_{k(kM+2)}$ theory and the $\mathbb{Z}_k$ parafermion CFT followed by the diagonal $\mathbb{Z}_k$ quotient,

$$\frac{U(1)_{k(kM+2)} \times \left(SU(2)_k/U(1)_{2k}\right)}{\mathbb{Z}_k}, \tag{94}$$

can describe the edge excitations of the Read-Rezayi states at filling fraction

$$\nu = \frac{k}{kM+2}, \tag{95}$$

where $(k-1)$ and $M$ are nonnegative integers. Here an odd (even) $M$ is for the fermionic (bosonic) state. We show that the generalized Jordan-Wigner transformation provides further evidence for an emergent supersymmetry on the edges of the specific Read-Rezayi states with $(k=2, M=1)$ and $(k=4, M=1)$[36, 37].

To see this, let us briefly review the $U(1)$ theory and $\mathbb{Z}_k$ parafermion CFT for the later purpose. The $U(1)$ theory with level $l$ is a free-boson theory on a circle of radius $R = \sqrt{l}$. For an even integer $l$, the $U(1)$ theory becomes rational, namely, the partition functions can be expressed in terms of a finite number of primary states. The characters of individual primaries are given by

$$\chi_\lambda^l(\tau) = \frac{1}{\eta(\tau)} \sum_{n \in \mathbb{Z}} q^{\frac{l}{2}\left(n+\frac{\lambda}{l}\right)^2}, \tag{96}$$

where $\lambda = 0, \pm 1, \pm 2, .., \pm(l/2-1), l/2$. The $S$-transformation rule of the character (96) is given by

$$\chi_\lambda^l(-1/\tau) = \frac{1}{\sqrt{l}} \sum_\mu e^{-\frac{2\pi i \lambda \mu}{l}} \chi_\mu^l(\tau). \tag{97}$$

On the one hand, the $\mathbb{Z}_k$ parafermion CFT is a coset model $SU(2)_k/U(1)_{2k}$. Thus, the central charge of the $\mathbb{Z}_k$ parafermion CFT is given by $c = \frac{3k}{k+2} - 1 = \frac{2(k-1)}{k+2}$. The $\mathbb{Z}_k$ parafermion CFT involves $\frac{k(k+1)}{2}$ primaries with conformal weights

$$h_{\ell,m}^k = \frac{\ell(\ell+2)}{4(k+2)} - \frac{m^2}{4k}, \tag{98}$$

where the range of the quantum numbers $\ell$ and $m$ is given by

$$\{(\ell,m) \mid 0 \le \ell \le k, \; -\ell+2 \le m \le \ell, \; \ell-m \in 2\mathbb{Z}\}. \tag{99}$$

Table 27: $U(1)_8 \times$ Ising model: The conformal characters $f_i$ of 24 primaries ($i = 0, 1, \cdots, 23$) can be expressed in terms of the characters of the $U(1)_8$ theory and the Ising model, denoted by $\chi_i^{l=8}$ and $\psi_{\ell,m}^{k=2}$. We highlight the primaries related to the discrete $\mathbb{Z}_2^C$ symmetry of the theory.

| $i$ | $h$ | $f_i$ | $\mathbb{Z}_2^A$ | $\mathbb{Z}_2^B$ | $\mathbb{Z}_2^C$ | $i$ | $h$ | $f_i$ | $\mathbb{Z}_2^A$ | $\mathbb{Z}_2^B$ | $\mathbb{Z}_2^C$ | $i$ | $h$ | $f_i$ | $\mathbb{Z}_2^A$ | $\mathbb{Z}_2^B$ | $\mathbb{Z}_2^C$ |
|---|---|---|---|---|---|---|---|---|---|---|---|---|---|---|---|---|---|
| 0 | 0 | $\chi_0^8 \psi_{2,2}^2$ | + | + | + | 8 | $\frac{3}{4}$ | $\chi_2^8 \psi_{2,0}^2$ | + | + | + | 16 | $\frac{5}{8}$ | $\chi_5^8 \psi_{1,1}^2$ | - | - | + |
| 1 | $\frac{1}{16}$ | $\chi_0^8 \psi_{1,1}^2$ | - | + | - | 9 | $\frac{9}{16}$ | $\chi_3^8 \psi_{2,2}^2$ | + | - | - | 17 | $\frac{17}{16}$ | $\chi_5^8 \psi_{2,0}^2$ | + | - | - |
| 2 | $\frac{1}{2}$ | $\chi_0^8 \psi_{2,0}^2$ | + | + | + | 10 | $\frac{5}{8}$ | $\chi_3^8 \psi_{1,1}^2$ | - | - | + | 18 | $\frac{1}{4}$ | $\chi_6^8 \psi_{2,2}^2$ | + | + | + |
| 3 | $\frac{1}{16}$ | $\chi_1^8 \psi_{2,2}^2$ | + | - | - | 11 | $\frac{17}{16}$ | $\chi_3^8 \psi_{2,0}^2$ | + | - | - | 19 | $\frac{5}{16}$ | $\chi_6^8 \psi_{1,1}^2$ | + | + | - |
| 4 | $\frac{1}{8}$ | $\chi_1^8 \psi_{1,1}^2$ | - | - | + | 12 | 1 | $\chi_4^8 \psi_{2,2}^2$ | + | + | + | 20 | $\frac{3}{4}$ | $\chi_6^8 \psi_{2,0}^2$ | - | + | + |
| 5 | $\frac{9}{16}$ | $\chi_1^8 \psi_{2,0}^2$ | + | - | - | 13 | $\frac{17}{16}$ | $\chi_4^8 \psi_{1,1}^2$ | - | + | - | 21 | $\frac{1}{16}$ | $\chi_7^8 \psi_{2,2}^2$ | + | - | - |
| 6 | $\frac{1}{4}$ | $\chi_2^8 \psi_{2,2}^2$ | + | + | + | 14 | $\frac{3}{2}$ | $\chi_4^8 \psi_{2,0}^2$ | + | + | + | 22 | $\frac{1}{8}$ | $\chi_7^8 \psi_{1,1}^2$ | + | - | + |
| 7 | $\frac{5}{16}$ | $\chi_2^8 \psi_{1,1}^2$ | - | + | - | 15 | $\frac{9}{16}$ | $\chi_5^8 \psi_{2,2}^2$ | + | - | - | 23 | $\frac{9}{16}$ | $\chi_7^8 \psi_{2,0}^2$ | - | - | - |

We denote the characters of primaries by $\psi_{\ell,m}^k$ in what follows. The characters of the $\mathbb{Z}_k$ parafermion CFT takes the form of

$$
\psi_{\ell,m}^k(\tau) = \frac{1}{\eta(\tau)^2}\Bigg( \Big( \sum_{i,j\leq 0} - \sum_{i,j<0} \Big)(-1)^i q^{\frac{(\ell+1+(i+2j)(k+2))^2}{4(k+2)} - \frac{(m+ik)^2}{4k}}
$$
$$
- \Big( \sum_{i\leq 0,j>0} - \sum_{i<0,j\leq 0} \Big)(-1)^i q^{\frac{(\ell+1-(i+2j)(k+2))^2}{4(k+2)} - \frac{(m+ik)^2}{4k}} \Bigg),
\tag{100}
$$

and their $S$-matrix is given by [56]

$$
S_{\ell m;\ell' m'}^k = \frac{2}{\sqrt{k(k+2)}} e^{2\pi i \frac{mm'}{2k}} \sin\left( \pi \frac{(\ell+1)(\ell'+1)}{k+2} \right).
\tag{101}
$$

**the Read-Rezayi state with** $(k = 2, M = 1)$  Armed with the characters and the modular S-matrix of the $U(1)$ theory and the $\mathbb{Z}_k$ parafermion theory, let us analyze an orbifold

$$
\widetilde{\mathcal{B}} = \frac{U(1)_8 \times (\text{Ising model})}{\mathbb{Z}_2},
\tag{102}
$$

proposed to describe the edge modes of the Read-Rezayi state at $\nu = 1/2$.

Let us first consider a product theory $\mathcal{B} = U(1)_8 \times (\text{Ising model})$ that has 24 primaries whose conformal weights and characters are given in table 27. There are three Verlinde lines $\mathcal{L}_{h=\frac{1}{2}}$, $\mathcal{L}_{h=1}$ and $\mathcal{L}_{h=\frac{3}{2}}$ that generate $\mathbb{Z}_2^A$, $\mathbb{Z}_2^B$ and $\mathbb{Z}_2^C$. The $\mathbb{Z}_2^A$ can be identified as that of the Ising model while $\mathbb{Z}_2^B$ as that of the $U(1)_8$ theory. The $\mathbb{Z}_2^C$ is the diagonal subgroup of $\mathbb{Z}_2^A \times \mathbb{Z}_2^B$.

The orbifold partition function follows from (4) with $\mathbb{Z}_2^C$ applied to diagonal modular invariant partition function $Z_{\mathcal{B}} = \sum_{i=0}^{23} |f_i|^2$ of $U(1)_8 \times (\text{Ising model})$,

$$
Z_{\widetilde{\mathcal{B}}} = \sum_{n=0}^{11} \left| f_{2n} \right|^2 + \left\{ (f_1 \bar{f}_{13} + f_3 \bar{f}_{17} + f_5 \bar{f}_{15} + f_7 \bar{f}_{19} + f_9 \bar{f}_{23} + f_{11} \bar{f}_{21}) + (\text{c.c}) \right\}.
\tag{103}
$$

We can see that $Z_{\mathcal{B}}$ and $Z_{\widetilde{\mathcal{B}}}$ differ by a constant, $Z_{\mathcal{B}} - Z_{\widetilde{\mathcal{B}}} = 3$. We therefore expect that the orbifold $\widetilde{\mathcal{B}}$ can be fermionized to a supersymmetric theory $\widetilde{\mathcal{F}}$.

Based on the web described in table 3, partition functions for $\widetilde{\mathcal{F}}$ can be determined as,

$$Z^{\mathrm{NS}}_{\widetilde{\mathcal{F}}} = \left|f_0 + f_{14}\right|^2 + \left|f_2 + f_{12}\right|^2 + \left|f_4 + f_{16}\right|^2 + \left|f_6 + f_{20}\right|^2 + \left|f_8 + f_{18}\right|^2 + \left|f_{10} + f_{22}\right|^2,$$

$$Z^{\widetilde{\mathrm{NS}}}_{\widetilde{\mathcal{F}}} = \left|f_0 - f_{14}\right|^2 + \left|f_2 - f_{12}\right|^2 + \left|f_4 - f_{16}\right|^2 + \left|f_6 - f_{20}\right|^2 + \left|f_8 - f_{18}\right|^2 + \left|f_{10} - f_{22}\right|^2,$$

$$Z^{\mathrm{R}}_{\widetilde{\mathcal{F}}} = \left|f_1 + f_{13}\right|^2 + \left|f_3 + f_{17}\right|^2 + \left|f_5 + f_{15}\right|^2 + \left|f_7 + f_{19}\right|^2 + \left|f_9 + f_{23}\right|^2 + \left|f_{11} + f_{21}\right|^2,$$

$$Z^{\widetilde{\mathrm{R}}}_{\widetilde{\mathcal{F}}} = -\left|f_1 - f_{13}\right|^2 - \left|f_3 - f_{17}\right|^2 - \left|f_5 - f_{15}\right|^2 - \left|f_7 - f_{19}\right|^2 - \left|f_9 - f_{23}\right|^2 - \left|f_{11} - f_{21}\right|^2$$

$$= -1^2 - 1^2 - 0^2 - 0^2 - 0^2 - 1^2, \tag{104}$$

which are in perfect agreement with those of [57]. Since (104) satisfies the SUSY conditions, we propose that there exists an emergent supersymmetry on the edges of the $(k = 2, M = 1)$ Read-Rezayi state. Moreover, we observe that the above partition functions (104) coincide with those of the $\mathcal{N} = 2$ unitary supersymmetric minimal model at $c = 3/2$. To find the explicit expressions of the $\mathcal{N} = 2$ super-Virasoro characters at $c = 3/2$, readers are referred to [58, 59]. It strongly suggests that the $\mathcal{N} = 2$ supersymmetry can emerge on the edges of the $(k = 2, M = 1)$ Read-Rezayi state.

**the Read-Rezayi state with $(k = 4, M = 1)$**    We next analyze the orbifold theory

$$\mathcal{B} = \frac{U(1)_{24} \times (\mathbb{Z}_4 \text{ parafermion CFT})}{\mathbb{Z}_4}, \tag{105}$$

proposed to describe the edge modes of the Read-Rezayi state with $(k = 4, M = 1)$.

$U(1)_{24} \times (\mathbb{Z}_4$ parafermion CFT$)$ has two copies of $\mathbb{Z}_4$ symmetries, one of which is originated from that of $U(1)_{24}$ and the other from that of the parafermion theory. Although the $\mathbb{Z}_4 \times \mathbb{Z}_4$ symmetries are anomalous, one can show that their diagonal subgroup $\mathbb{Z}_4$ becomes non-anomalous. It is the Verlinde line $\mathcal{L}_{h=3/2}$ associated with $\chi^{24}_{18}\psi^4_{4,2}$ that generate the non-anomalous $\mathbb{Z}_4$ symmetry. We can gauge the non-anomalous $\mathbb{Z}_4$, which results in the orbifold (105) that has 60 primaries whose conformal weights and conformal characters can be found in table 28. There are two the Verlinde lines $\tilde{\mathcal{L}}_{h=3/2}$ which generate essentially the same $\widetilde{\mathbb{Z}}_2$ symmetry.

We can fermionize the bosonic theory (105) with help of the $\widetilde{\mathbb{Z}}_2$ symmetry. The fermionic partition function in each spin structure follows from (5) applied to a diagonal partition function $Z_{\mathcal{B}} = \sum_{i=0}^{59} |f_i|^2$,

$$\begin{aligned}
Z^{\mathrm{NS}}_{\mathcal{F}} = &\left|f_0 + f_{33}\right|^2 + \left|f_2 + f_{31}\right|^2 + \left|f_5 + f_{34}\right|^2 + \left|f_6 + f_{38}\right|^2 + \left|f_9 + f_{37}\right|^2 \\
&+ \left|f_{11} + f_{42}\right|^2 + \left|f_{13} + f_{45}\right|^2 + \left|f_{14} + f_{40}\right|^2 + \left|f_{17} + f_{46}\right|^2 + \left|f_{18} + f_{49}\right|^2 \\
&+ \left|f_{20} + f_{53}\right|^2 + \left|f_{22} + f_{51}\right|^2 + \left|f_{25} + f_{54}\right|^2 + \left|f_{26} + f_{58}\right|^2 + \left|f_{29} + f_{57}\right|^2, \\
Z^{\widetilde{\mathrm{NS}}}_{\mathcal{F}} = &\left|f_0 - f_{33}\right|^2 + \left|f_2 - f_{31}\right|^2 + \left|f_5 - f_{34}\right|^2 + \left|f_6 - f_{38}\right|^2 + \left|f_9 - f_{37}\right|^2 \\
&+ \left|f_{11} - f_{42}\right|^2 + \left|f_{13} - f_{45}\right|^2 + \left|f_{14} - f_{40}\right|^2 + \left|f_{17} - f_{46}\right|^2 + \left|f_{18} - f_{49}\right|^2 \\
&+ \left|f_{20} - f_{53}\right|^2 + \left|f_{22} - f_{51}\right|^2 + \left|f_{25} - f_{54}\right|^2 + \left|f_{26} - f_{58}\right|^2 + \left|f_{29} - f_{57}\right|^2, \\
Z^{\mathrm{R}}_{\mathcal{F}} = &\left|f_1 + f_{32}\right|^2 + \left|f_3 + f_{35}\right|^2 + \left|f_4 + f_{30}\right|^2 + \left|f_7 + f_{36}\right|^2 + \left|f_8 + f_{39}\right|^2 \\
&+ \left|f_{10} + f_{43}\right|^2 + \left|f_{12} + f_{41}\right|^2 + \left|f_{15} + f_{44}\right|^2 + \left|f_{16} + f_{48}\right|^2 + \left|f_{19} + f_{47}\right|^2 \\
&+ \left|f_{21} + f_{52}\right|^2 + \left|f_{23} + f_{55}\right|^2 + \left|f_{24} + f_{50}\right|^2 + \left|f_{27} + f_{56}\right|^2 + \left|f_{28} + f_{59}\right|^2, \\
Z^{\widetilde{\mathrm{R}}}_{\mathcal{F}} = &\left|f_1 - f_{32}\right|^2 + \left|f_3 - f_{35}\right|^2 + \left|f_4 - f_{30}\right|^2 + \left|f_7 - f_{36}\right|^2 + \left|f_8 - f_{39}\right|^2 \\
&+ \left|f_{10} - f_{43}\right|^2 + \left|f_{12} - f_{41}\right|^2 + \left|f_{15} - f_{44}\right|^2 + \left|f_{16} - f_{48}\right|^2 + \left|f_{19} - f_{47}\right|^2 \\
&+ \left|f_{21} - f_{52}\right|^2 + \left|f_{23} - f_{55}\right|^2 + \left|f_{24} - f_{50}\right|^2 + \left|f_{27} - f_{56}\right|^2 + \left|f_{28} - f_{59}\right|^2.
\end{aligned} \tag{106}$$

Table 28: $\big(U(1)_{24} \times (\mathbb{Z}_4 \text{ parafermion CFT})\big)/(\mathbb{Z}_4)$: The conformal characters $f_i$ of 60 primaries ($i = 0, 1, \cdots, 59$) can be expressed in terms of the characters of the $U(1)_{24}$ theory and the $\mathbb{Z}_4$ Parafermion CFT, denoted by $\chi_i^{l=24}$ and $\psi_{\ell,m}^{k=4}$. We highlight the primaries related to the discrete $\mathbb{Z}_2$ symmetry of the theory.

| $i$ | $h$ | $f_i$ | $\mathbb{Z}_2$ | $i$ | $h$ | $f_i$ | $\mathbb{Z}_2$ | $i$ | $h$ | $f_i$ | $\mathbb{Z}_2$ |
|---|---|---|---|---|---|---|---|---|---|---|---|
| 0 | 0 | $\chi_0^{24}\psi_{4,4}^4 + \chi_{12}^{24}\psi_{4,0}^4$ | + | 20 | $\frac{1}{3}$ | $\chi_4^{24}\psi_{4,4}^4 + \chi_{16}^{24}\psi_{4,0}^4$ | + | 40 | $\frac{4}{3}$ | $\chi_8^{24}\psi_{4,4}^4 + \chi_{20}^{24}\psi_{4,0}^4$ | + |
| 1 | $\frac{13}{12}$ | $\chi_6^{24}\psi_{2,0}^4 + \chi_{18}^{24}\psi_{2,0}^4$ | - | 21 | $\frac{5}{12}$ | $\chi_{10}^{24}\psi_{2,0}^4 + \chi_{22}^{24}\psi_{2,0}^4$ | - | 41 | $\frac{5}{12}$ | $\chi_2^{24}\psi_{2,0}^4 + \chi_{14}^{24}\psi_{2,0}^4$ | - |
| 2 | $\frac{1}{3}$ | $\chi_0^{24}\psi_{2,0}^4 + \chi_{12}^{24}\psi_{2,0}^4$ | + | 22 | $\frac{2}{3}$ | $\chi_4^{24}\psi_{2,0}^4 + \chi_{16}^{24}\psi_{2,0}^4$ | + | 42 | $\frac{2}{3}$ | $\chi_8^{24}\psi_{2,0}^4 + \chi_{20}^{24}\psi_{2,0}^4$ | + |
| 3 | $\frac{3}{4}$ | $\chi_{18}^{24}\psi_{4,4}^4 + \chi_6^{24}\psi_{4,0}^4$ | - | 23 | $\frac{1}{12}$ | $\chi_{10}^{24}\psi_{4,0}^4 + \chi_{22}^{24}\psi_{4,4}^4$ | - | 43 | $\frac{1}{12}$ | $\chi_2^{24}\psi_{4,4}^4 + \chi_{14}^{24}\psi_{4,0}^4$ | - |
| 4 | $\frac{3}{4}$ | $\chi_6^{24}\psi_{4,4}^4 + \chi_{18}^{24}\psi_{4,0}^4$ | - | 24 | $\frac{13}{12}$ | $\chi_{10}^{24}\psi_{4,4}^4 + \chi_{12}^{22}\psi_{4,0}^4$ | - | 44 | $\frac{13}{12}$ | $\chi_2^{24}\psi_{4,0}^4 + \chi_{14}^{24}\psi_{4,4}^4$ | - |
| 5 | 1 | $\chi_{12}^{24}\psi_{4,4}^4 + \chi_0^{24}\psi_{4,0}^4$ | + | 25 | $\frac{4}{3}$ | $\chi_4^{24}\psi_{4,0}^4 + \chi_{16}^{24}\psi_{4,4}^4$ | + | 45 | $\frac{1}{3}$ | $\chi_8^{24}\psi_{4,0}^4 + \chi_{20}^{24}\psi_{4,4}^4$ | + |
| 6 | $\frac{1}{12}$ | $\chi_1^{24}\psi_{1,1}^4 + \chi_{13}^{24}\psi_{3,1}^4$ | + | 26 | $\frac{7}{12}$ | $\chi_5^{24}\psi_{1,1}^4 + \chi_{17}^{24}\psi_{3,1}^4$ | + | 46 | $\frac{3}{4}$ | $\chi_9^{24}\psi_{1,1}^4 + \chi_{21}^{24}\psi_{3,1}^4$ | + |
| 7 | $\frac{13}{12}$ | $\chi_7^{24}\psi_{1,1}^4 + \chi_{19}^{24}\psi_{3,1}^4$ | - | 27 | $\frac{7}{12}$ | $\chi_{11}^{24}\psi_{1,1}^4 + \chi_{23}^{24}\psi_{3,1}^4$ | - | 47 | $\frac{3}{4}$ | $\chi_3^{24}\psi_{3,1}^4 + \chi_{15}^{24}\psi_{1,1}^4$ | - |
| 8 | $\frac{7}{12}$ | $\chi_7^{24}\psi_{3,1}^4 + \chi_{19}^{24}\psi_{1,1}^4$ | - | 28 | $\frac{1}{12}$ | $\chi_{11}^{24}\psi_{3,1}^4 + \chi_{23}^{24}\psi_{1,1}^4$ | - | 48 | $\frac{1}{4}$ | $\chi_3^{24}\psi_{1,1}^4 + \chi_{15}^{24}\psi_{3,1}^4$ | - |
| 9 | $\frac{7}{12}$ | $\chi_1^{24}\psi_{3,1}^4 + \chi_{13}^{24}\psi_{1,1}^4$ | + | 29 | $\frac{13}{12}$ | $\chi_9^{24}\psi_{3,1}^4 + \chi_{21}^{24}\psi_{1,1}^4$ | + | 49 | $\frac{1}{4}$ | $\chi_0^{24}\psi_{4,4}^4 + \chi_{12}^{24}\psi_{4,0}^4$ | + |
| 10 | $\frac{13}{12}$ | $\chi_8^{24}\psi_{4,-2}^4 + \chi_{20}^{24}\psi_{4,2}^4$ | - | 30 | $\frac{3}{4}$ | $\chi_0^{24}\psi_{4,2}^4 + \chi_{12}^{24}\psi_{4,-2}^4$ | - | 50 | $\frac{13}{12}$ | $\chi_4^{24}\psi_{4,2}^4 + \chi_{16}^{24}\psi_{4,-2}^4$ | - |
| 11 | $\frac{1}{6}$ | $\chi_2^{24}\psi_{2,2}^4 + \chi_{14}^{24}\psi_{2,2}^4$ | + | 31 | $\frac{5}{6}$ | $\chi_6^{24}\psi_{2,2}^4 + \chi_{18}^{24}\psi_{2,2}^4$ | + | 51 | $\frac{1}{6}$ | $\chi_{10}^{24}\psi_{2,2}^4 + \chi_{22}^{24}\psi_{2,2}^4$ | + |
| 12 | $\frac{5}{12}$ | $\chi_8^{24}\psi_{2,2}^4 + \chi_{20}^{24}\psi_{2,2}^4$ | - | 32 | $\frac{1}{12}$ | $\chi_0^{24}\psi_{2,2}^4 + \chi_{12}^{24}\psi_{2,2}^4$ | - | 52 | $\frac{5}{12}$ | $\chi_4^{24}\psi_{2,2}^4 + \chi_{16}^{24}\psi_{2,2}^4$ | - |
| 13 | $\frac{5}{6}$ | $\chi_2^{24}\psi_{4,-2}^4 + \chi_{14}^{24}\psi_{4,2}^4$ | + | 33 | $\frac{3}{2}$ | $\chi_6^{24}\psi_{4,-2}^4 + \chi_{18}^{24}\psi_{4,2}^4$ | + | 53 | $\frac{5}{6}$ | $\chi_{10}^{24}\psi_{4,-2}^4 + \chi_{22}^{24}\psi_{4,2}^4$ | + |
| 14 | $\frac{5}{6}$ | $\chi_2^{24}\psi_{4,2}^4 + \chi_{14}^{24}\psi_{4,-2}^4$ | + | 34 | $\frac{3}{2}$ | $\chi_6^{24}\psi_{4,2}^4 + \chi_{18}^{24}\psi_{4,-2}^4$ | + | 54 | $\frac{5}{6}$ | $\chi_{10}^{24}\psi_{4,2}^4 + \chi_{22}^{24}\psi_{4,-2}^4$ | + |
| 15 | $\frac{13}{12}$ | $\chi_8^{24}\psi_{4,2}^4 + \chi_{20}^{24}\psi_{4,-2}^4$ | - | 35 | $\frac{3}{4}$ | $\chi_0^{24}\psi_{4,-2}^4 + \chi_{12}^{24}\psi_{4,2}^4$ | - | 55 | $\frac{13}{12}$ | $\chi_4^{24}\psi_{4,-2}^4 + \chi_{16}^{24}\psi_{4,2}^4$ | - |
| 16 | $\frac{1}{4}$ | $\chi_9^{24}\psi_{3,-1}^4 + \chi_{21}^{24}\psi_{3,3}^4$ | - | 36 | $\frac{1}{12}$ | $\chi_1^{24}\psi_{3,3}^4 + \chi_{13}^{24}\psi_{3,-1}^4$ | - | 56 | $\frac{7}{12}$ | $\chi_5^{24}\psi_{3,3}^4 + \chi_{17}^{24}\psi_{3,-1}^4$ | - |
| 17 | $\frac{1}{4}$ | $\chi_3^{24}\psi_{3,3}^4 + \chi_{15}^{24}\psi_{3,-1}^4$ | + | 37 | $\frac{13}{12}$ | $\chi_7^{24}\psi_{3,3}^4 + \chi_{19}^{24}\psi_{3,-1}^4$ | + | 57 | $\frac{7}{12}$ | $\chi_{11}^{24}\psi_{3,3}^4 + \chi_{23}^{24}\psi_{3,-1}^4$ | + |
| 18 | $\frac{3}{4}$ | $\chi_3^{24}\psi_{3,-1}^4 + \chi_{15}^{24}\psi_{3,3}^4$ | + | 38 | $\frac{7}{12}$ | $\chi_7^{24}\psi_{3,-1}^4 + \chi_{19}^{24}\psi_{3,3}^4$ | + | 58 | $\frac{1}{12}$ | $\chi_{11}^{24}\psi_{3,-1}^4 + \chi_{23}^{24}\psi_{3,3}^4$ | + |
| 19 | $\frac{3}{4}$ | $\chi_9^{24}\psi_{3,3}^4 + \chi_{21}^{24}\psi_{3,-1}^4$ | - | 39 | $\frac{7}{12}$ | $\chi_1^{24}\psi_{3,-1}^4 + \chi_{13}^{24}\psi_{3,3}^4$ | - | 59 | $\frac{13}{12}$ | $\chi_5^{24}\psi_{3,-1}^4 + \chi_{17}^{24}\psi_{3,3}^4$ | - |

It is clear that the SUSY conditions are all satisfied by (106); let us take a closer look at the $q$-expansion of the NS vacuum given below

$$f_0 + f_{33} = q^{-1/12}\Big[1 + q + 2q^{3/2} + 3q^2 + 4q^{5/2} + 6q^3 + \cdots\Big], \tag{107}$$

where we can see that the NS vacuum has descendants of $h = 1$ and $h = 3/2$. Those descendants are likely to play roles of the $U(1)$ R-symmetry and the supersymmetry currents of $\mathcal{N} = 2$ supersymmetry. Moreover, $Z_{\mathcal{F}}^{\widetilde{\text{R}}}$ takes a constant value and thus becomes an index,

$$Z_{\mathcal{F}}^{\widetilde{\text{R}}} = 1^2 + 0 + 0 + 1^2 + 0 + 1^2 + 0 + 0 + 0 + 0 + 0 + 1^2 + 0 + 0 + 1^2. \tag{108}$$

In fact, we can verify that the partition functions (106) agree with those of the sixth unitary $\mathcal{N} = 2$ supersymmetric minimal model at $c = 2$ [58, 59]. Especially, (107) correspond to the NS vacuum character of the $\mathcal{N} = 2$ super-Virasoro vacuum character at $c = 2$. Therefore, we propose again that there can exist an emergent $\mathcal{N} = 2$ supersymmetry on the edges of the Read-Rezayi state of $(k = 4, M = 1)$, briefly mentioned in [36].

# Acknowledgment

We thank to Gil Young Cho, Hyekyung Choi, Zhihao Duan, Dongmin Gang, Eun-Gook Moon, Kimyeong Lee, Yuji Tachikawa, David Tong, and Piljin Yi for useful discussion. The work of J.B. is supported by the European Research Council (ERC) under the European Union's Horizon

2020 research and innovation programme (Grant No. 787185). S.L. is supported by KIAS Individual Grant PG056502.

# A  Fermionization of the Parafermion CFT

The prime goal of this appendix is to apply the generalized Jordan-Wigner transformation to the $\mathbb{Z}_k$ parafermion CFT. We choose $\mathbb{Z}_2$ subgroup of $\mathbb{Z}_k$ symmetry to explore the fermionized parafermion CFT. Therefore, $k$ ought to be even integers. Furthermore, the weight formula (98) suggests that the weight $\frac{3}{2}$ primary can exist only for $k \leq 8$. To avoid the free-fermion issue, we only consider $\mathbb{Z}_6$ and $\mathbb{Z}_8$ parafermion CFT.

$\mathbb{Z}_6$ **parafermion CFT**   $\mathbb{Z}_6$ parafermion CFT has 21 primaries and the central charge is 5/4. We consider the $\mathbb{Z}_2$ subgroup of $\mathbb{Z}_6$ to apply fermionization (5). The weight 3/2 primary with $(\ell = 6, m = 0)$ generate the $\mathbb{Z}_2$ symmetry of interest and $\ell$ even (odd) primaries are even (odd) states under the $\mathbb{Z}_2$ subgroup.

It is easy to find the partition functions of $\widetilde{\mathcal{F}}$ for each spin structure. The generalized Jordan-Wigner transformation provides the following partition functions.

$$
\begin{aligned}
Z_{\widetilde{\mathcal{F}}}^{\text{NS}} = {} & \left|\psi_{6,6}^6 + \psi_{6,0}^6\right|^2 + \left|\psi_{2,2}^6 + \psi_{4,2}^6\right|^2 + \left|\psi_{4,4}^6 + \psi_{4,-2}^6\right|^2 \\
& + \left|\psi_{2,0}^6 + \psi_{4,0}^6\right|^2 + \left|\psi_{6,4}^6 + \psi_{6,2}^6\right|^2 + \left|\psi_{6,-4}^6 + \psi_{6,-2}^6\right|^2, \\
Z_{\widetilde{\mathcal{F}}}^{\widetilde{\text{NS}}} = {} & \left|\psi_{6,6}^6 - \psi_{6,0}^6\right|^2 + \left|\psi_{2,2}^6 - \psi_{4,2}^6\right|^2 + \left|\psi_{4,4}^6 - \psi_{4,-2}^6\right|^2 \\
& + \left|\psi_{2,0}^6 - \psi_{4,0}^6\right|^2 + \left|\psi_{6,4}^6 - \psi_{6,2}^6\right|^2 + \left|\psi_{6,-4}^6 - \psi_{6,-2}^6\right|^2, \\
Z_{\widetilde{\mathcal{F}}}^{\text{R}} = {} & \left|\psi_{1,1}^6 + \psi_{5,1}^6\right|^2 + \left|\psi_{5,5}^6 + \psi_{5,-1}^6\right|^2 + \left|\psi_{5,-3}^6 + \psi_{5,3}^6\right|^2 \\
& + \left|\sqrt{2}\psi_{3,3}^6\right|^2 + \left|\sqrt{2}\psi_{3,-1}^6\right|^2 + \left|\sqrt{2}\psi_{3,1}^6\right|^2, \\
Z_{\widetilde{\mathcal{F}}}^{\widetilde{\text{R}}} = {} & \left|\psi_{1,1}^6 - \psi_{5,1}^6\right|^2 + \left|\psi_{5,5}^6 - \psi_{5,-1}^6\right|^2 + \left|\psi_{5,-3}^6 - \psi_{5,3}^6\right|^2 = 1^2 + 1^2 + 0.
\end{aligned}
\tag{109}
$$

The above partition functions (109) satisfy the SUSY conditions. Especially, the primary of Ramond sector with weight $h = \frac{5}{96}$ saturate the unitarity bound. Thus, we expect that the Ramond sector ground state preserves supersymmetry.

$\mathbb{Z}_8$ **parafermion CFT**   Our next target is to fermionize the $\mathbb{Z}_8$ parafermion CFT with central charge $c = 7/5$. We first note that the weight-two primary of $(\ell = 8, m = 0)$ has a role of the generator for $\mathbb{Z}_2$ symmetry. The $\mathbb{Z}_2$ action on each primary is readily obtained from the $S$-matrix, the $\ell$ even representations are $\mathbb{Z}_2$ even while $\ell$ odd are $\mathbb{Z}_2$ odd. An orbifold partition function in turn takes the form of

$$
\begin{aligned}
Z_{\widetilde{\mathcal{B}}} = {} & \left|\psi_{8,8}^8 + \psi_{8,0}^8\right|^2 + \left|\psi_{8,-4}^8 + \psi_{8,4}^8\right|^2 + \left|\psi_{2,0}^8 + \psi_{6,0}^8\right|^2 \\
& + \left|\psi_{6,-4}^8 + \psi_{6,4}^8\right|^2 + 2\left|\psi_{4,4}^8\right|^2 + 2\left|\psi_{4,0}^8\right|^2 \\
& + \left|\psi_{2,2}^8 + \psi_{6,2}^8\right|^2 + \left|\psi_{6,6}^8 + \psi_{6,-2}^8\right|^2 + \left|\psi_{8,-6}^8 + \psi_{8,2}^8\right|^2 \\
& + \left|\psi_{8,6}^8 + \psi_{8,-2}^8\right|^2 + 2\left|\psi_{4,-2}^8\right|^2 + 2\left|\psi_{4,2}^8\right|^2.
\end{aligned}
\tag{110}
$$

In the $\mathbb{Z}_2$ orbifold theory $\widetilde{\mathcal{B}}$, the weight-3/2 primary acquire a role of the generator of new $\widetilde{\mathbb{Z}}_2$ symmetry. Under the $\widetilde{\mathbb{Z}}_2$ symmetry, the first and second lines of (110) are even while the third and fourth lines are odd. After some computation, we find that the fermionization of $Z_{\widetilde{\mathcal{B}}}$

provides the following partition functions.

$$Z^{\text{NS}}_{\widetilde{\mathcal{F}}} = \left|\psi^8_{8,8} + \psi^8_{8,0} + \psi^8_{8,-4} + \psi^8_{8,4}\right|^2 + \left|\psi^8_{2,0} + \psi^8_{6,0} + \psi^8_{6,-4} + \psi^8_{6,4}\right|^2 + \left|\psi^8_{4,4} + \psi^8_{4,0}\right|^2,$$

$$Z^{\widetilde{\text{NS}}}_{\widetilde{\mathcal{F}}} = \left|\psi^8_{8,8} + \psi^8_{8,0} - \psi^8_{8,-4} - \psi^8_{8,4}\right|^2 + \left|\psi^8_{2,0} + \psi^8_{6,0} - \psi^8_{6,-4} - \psi^8_{6,4}\right|^2 + \left|\psi^8_{4,4} - \psi^8_{4,0}\right|^2,$$

$$Z^{\text{R}}_{\widetilde{\mathcal{F}}} = \left|\psi^8_{2,2} + \psi^8_{6,2} + \psi^8_{6,6} + \psi^8_{6,-2}\right|^2 + \left|\psi^8_{8,-6} + \psi^8_{8,2} + \psi^8_{8,6} + \psi^8_{8,-2}\right|^2 + \left|\psi^8_{4,-2} + \psi^8_{4,2}\right|^2,$$

$$Z^{\widetilde{\text{R}}}_{\widetilde{\mathcal{F}}} = \left|\psi^8_{2,2} + \psi^8_{6,2} - \psi^8_{6,6} - \psi^8_{6,-2}\right|^2 + \left|\psi^8_{8,-6} + \psi^8_{8,2} - \psi^8_{8,6} - \psi^8_{8,-2}\right|^2 + \left|\psi^8_{4,-2} - \psi^8_{4,2}\right|^2.$$

$$(111)$$

Especially, the $\widetilde{\text{R}}$-sector partition function $Z^{\widetilde{\text{R}}}_{\widetilde{\mathcal{F}}}$ vanishes. Therefore the partition functions (111) satisfy the SUSY conditions, however the Ramond ground state possesses broken supersymmetry in contrast to (109).

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
