# Peer review of "Emergent Supersymmetry on the Edges"

_SciPost Physics, doi:SciPost Phys. 11, 091 (2021)_

## Round 1 · Referee Report · Anonymous (Referee 1) · 2021-7-14

Report
Recently the relation between bosonic 2d CFTs with Z2 symmetry and fermionic 2d CFTs has been greatly clarified in the literature.
In this paper the authors considered the fermonized WZW models and studied when they have supersymmetry. It is an interesting paper, and will easily pass the criteria to be published in this journal, once the issues listed below are addressed.
There are a few major comments:
In general, the title and the abstract emphasizing supersymmetry on the edges and the content which studies the fermionized non-chiral WZW models do not match very well. The referee thinks that the latter is by itself an interesting mathematical physics question worth being published here, and that it is closely related to the supersymmetry on the edges, but the authors should emphasize that they are not quite the same. The edge theory is inherently chiral; if we put the Chern-Simons theory $G_k$ on a disk with an insertion of a single anyon $\lambda$ at the center, the edge excitation has the Hilbert space whose character is $\chi_\lambda$. Although no extensive discussion is available in the literature, if we have a suitable Z2 1-form symmetry in the bulk which reduces to a suitable Z2 0-form symmetry on the edge, then we can gauge it to make it fermionic both on the edge and in the bulk at the same time. But the edge theory will not be the fermionized non-chiral WZW models as discussed in this paper, but will rather be a chiral model whose NS vacuum character would be the original vacuum character plus the character whose h=3/2.
This distinction (between the non-chiral WZW theories and the chiral edge theories) needs to be clearly made in the paper.
In Sec. 2.2, why do the authors list the conditions 1.2.3 as necessary conditions? If the NS sector partition function contains an state of dimension $(h_L,h_R)=(3/2,0)$, it becomes a spin 3/2 operator G via the state-operator correspondence, and G, T together are forced to have the OPE (2.8) simply from consistency. So the referee thinks that this condition is also sufficient.
In Sec.2, it would be nice to review the possible anomalies of Z2 symmetry in bosonic 2d theories, and how to distinguish them. It is not indispensable, but as Sec.2 is already such a nice review of various materials, the referee thinks that it would be nice to have this too.
One nice source to cite is https://arxiv.org/abs/1802.04445 , see in particular their Sec 4.4. There it is explained that the Z2 symmetry in 2d bosonic theories is either non-anomalous or anomalous, and that it can be distinguished in the case of Verlinde lines just by looking at its spin, so that it is non-anomalous when h=0 or 1/2 mod 1 and is anomalous when h=1/4 or 3/4 mod 1.
In Sec.3, it would be nice if the authors explain why $SU(N)_k$ with Nk=12 works for A-type and D-type. Is there a deep reason behind this observation?
The referee also has various minor comments:
p.1: The authors write "novel ideas such as hierarchy states [11,12]", but they are from the early eighties. I strongly doubt if the authors were even born back then, so they might want to drop the adjective "novel"
p.2: The authors refer to (R,NS) etc. without specifying which is the temporal and which is the spatial spin structure. Please do specify.
p.2: The authors refer to the nontrivial phase of the Kitaev chain simply as the Kitaev chain. This is often done in hep-th, but is frowned upon on the cond-mat side of the community. Please do add the qualification "the nontrivial (or topological) phase of" to the Kitaev chain, at least at the first mention.
p.3: consistent to -> consistent with
p.4: In (2.3) the authors refer to PP, AP etc, while they use R, $\tilde R$ etc in the other parts of the paper. Please unify the notations.
p.6: supersymemtry -> supersymmetry
p.8: in correspond with -> in correspondence with
p.9: by implementing the SageMath package -> "by writing a program in the SageMath", or if there is a particular SageMath package which does the computation of the affine character already, please mention it by name.
p.10: The logic in the last paragraph is strange. First, the sentence "The group elements of center B(G) are commutes" is grammatically incorrect. It should have been "The group elements of center B(G) all commute", but then B(G) is not defined at this point, if it is the center there is nothing to show about why its elements commute with any element of G, etc. The referee thinks that the authors should first define B(G) via (2.23) and show that it is a subgroup of the center.
p.11: It would be nicer to have a discussion of which Z2 subgroup of O(G) is (non-)anomalous.
p.13: $n^2/2$ -> $n^2/(2k)$
p.14: this originates that -> this implies that
p.14: there is one Z2 which is not mathbb
p.16: focus on discussion -> focus on the discussion
p.27: there is a comma at the start of a line
p.27: It would be nice to have a Lagrangian way to understand that the fermionized $SO(N)_3$ is always supersymmetric.
p.35: The authors say that the fermionic RCFTs are not always a $Z_2$ orbifold of WZW model. That is reasonable. But at least in the c=39/2 case mentioned here, what happens if we perform the reverse Jordan-Wigner transformation to get the bosonic theory? By construction it would contain an $(E_6)_4$ algebra, so it would presumably be a non-diagonal $(E_6)_4$ WZW invariant, is it not?
p.36: The authors say that "... provides further evidence for an emergent SUSY". It would be nice to cite [36], [37] again here.
In this paper the authors considered the fermonized WZW models and studied when they have supersymmetry. It is an interesting paper, and will easily pass the criteria to be published in this journal, once the issues listed below are addressed.
There are a few major comments:
In general, the title and the abstract emphasizing supersymmetry on the edges and the content which studies the fermionized non-chiral WZW models do not match very well. The referee thinks that the latter is by itself an interesting mathematical physics question worth being published here, and that it is closely related to the supersymmetry on the edges, but the authors should emphasize that they are not quite the same. The edge theory is inherently chiral; if we put the Chern-Simons theory $G_k$ on a disk with an insertion of a single anyon $\lambda$ at the center, the edge excitation has the Hilbert space whose character is $\chi_\lambda$. Although no extensive discussion is available in the literature, if we have a suitable Z2 1-form symmetry in the bulk which reduces to a suitable Z2 0-form symmetry on the edge, then we can gauge it to make it fermionic both on the edge and in the bulk at the same time. But the edge theory will not be the fermionized non-chiral WZW models as discussed in this paper, but will rather be a chiral model whose NS vacuum character would be the original vacuum character plus the character whose h=3/2.
This distinction (between the non-chiral WZW theories and the chiral edge theories) needs to be clearly made in the paper.
In Sec. 2.2, why do the authors list the conditions 1.2.3 as necessary conditions? If the NS sector partition function contains an state of dimension $(h_L,h_R)=(3/2,0)$, it becomes a spin 3/2 operator G via the state-operator correspondence, and G, T together are forced to have the OPE (2.8) simply from consistency. So the referee thinks that this condition is also sufficient.
In Sec.2, it would be nice to review the possible anomalies of Z2 symmetry in bosonic 2d theories, and how to distinguish them. It is not indispensable, but as Sec.2 is already such a nice review of various materials, the referee thinks that it would be nice to have this too.
One nice source to cite is https://arxiv.org/abs/1802.04445 , see in particular their Sec 4.4. There it is explained that the Z2 symmetry in 2d bosonic theories is either non-anomalous or anomalous, and that it can be distinguished in the case of Verlinde lines just by looking at its spin, so that it is non-anomalous when h=0 or 1/2 mod 1 and is anomalous when h=1/4 or 3/4 mod 1.
In Sec.3, it would be nice if the authors explain why $SU(N)_k$ with Nk=12 works for A-type and D-type. Is there a deep reason behind this observation?
The referee also has various minor comments:
p.1: The authors write "novel ideas such as hierarchy states [11,12]", but they are from the early eighties. I strongly doubt if the authors were even born back then, so they might want to drop the adjective "novel"
p.2: The authors refer to (R,NS) etc. without specifying which is the temporal and which is the spatial spin structure. Please do specify.
p.2: The authors refer to the nontrivial phase of the Kitaev chain simply as the Kitaev chain. This is often done in hep-th, but is frowned upon on the cond-mat side of the community. Please do add the qualification "the nontrivial (or topological) phase of" to the Kitaev chain, at least at the first mention.
p.3: consistent to -> consistent with
p.4: In (2.3) the authors refer to PP, AP etc, while they use R, $\tilde R$ etc in the other parts of the paper. Please unify the notations.
p.6: supersymemtry -> supersymmetry
p.8: in correspond with -> in correspondence with
p.9: by implementing the SageMath package -> "by writing a program in the SageMath", or if there is a particular SageMath package which does the computation of the affine character already, please mention it by name.
p.10: The logic in the last paragraph is strange. First, the sentence "The group elements of center B(G) are commutes" is grammatically incorrect. It should have been "The group elements of center B(G) all commute", but then B(G) is not defined at this point, if it is the center there is nothing to show about why its elements commute with any element of G, etc. The referee thinks that the authors should first define B(G) via (2.23) and show that it is a subgroup of the center.
p.11: It would be nicer to have a discussion of which Z2 subgroup of O(G) is (non-)anomalous.
p.13: $n^2/2$ -> $n^2/(2k)$
p.14: this originates that -> this implies that
p.14: there is one Z2 which is not mathbb
p.16: focus on discussion -> focus on the discussion
p.27: there is a comma at the start of a line
p.27: It would be nice to have a Lagrangian way to understand that the fermionized $SO(N)_3$ is always supersymmetric.
p.35: The authors say that the fermionic RCFTs are not always a $Z_2$ orbifold of WZW model. That is reasonable. But at least in the c=39/2 case mentioned here, what happens if we perform the reverse Jordan-Wigner transformation to get the bosonic theory? By construction it would contain an $(E_6)_4$ algebra, so it would presumably be a non-diagonal $(E_6)_4$ WZW invariant, is it not?
p.36: The authors say that "... provides further evidence for an emergent SUSY". It would be nice to cite [36], [37] again here.

---

## Round 1 · Referee Report · Anonymous (Referee 2) · 2021-8-13

Report
This is an interesting paper, studying a basic question in CFT: when do WZW models have a hidden supersymmetry? Or, more precisely, when do the fermionized versions of these models have supersymmetry?
This is an interesting question and the paper is clearly written and certainly worthy of publication in SciPost. I have only minor comments and corrections.
First, the title does not seem to fit the main result of the paper. The results are very much focussed on 2d CFTs. Their role as the edge modes of a CS theory, let alone of quantum Hall states, seems to be secondary at best. Moreover, the fermionization story involve a Z_2 quotient of the CFT, coupled with Arf invariant and I didn't understand the importance of this from the bulk or from the quantum Hall perspective.
Other comments. "Laurant modes" is misspelled, as is "supersymemtric". (This latter error is particularly egregious when discussing anyons because [e,m]\neq 0.)
Finally, there is an important Arf identity written between equations (2.6 ) and (2.7) . I wondered how it was proven but the paper cited ([21] in the bibliography) simply sends the reader to an old paper of Atiyah. It may be better to cite this directly.
This is an interesting question and the paper is clearly written and certainly worthy of publication in SciPost. I have only minor comments and corrections.
First, the title does not seem to fit the main result of the paper. The results are very much focussed on 2d CFTs. Their role as the edge modes of a CS theory, let alone of quantum Hall states, seems to be secondary at best. Moreover, the fermionization story involve a Z_2 quotient of the CFT, coupled with Arf invariant and I didn't understand the importance of this from the bulk or from the quantum Hall perspective.
Other comments. "Laurant modes" is misspelled, as is "supersymemtric". (This latter error is particularly egregious when discussing anyons because [e,m]\neq 0.)
Finally, there is an important Arf identity written between equations (2.6 ) and (2.7) . I wondered how it was proven but the paper cited ([21] in the bibliography) simply sends the reader to an old paper of Atiyah. It may be better to cite this directly.

---

## Round 2 · Referee Report · Anonymous (Referee 1) · 2021-9-24

Report
Warnings issued while processing user-supplied markup:
- Inconsistency: plain/Markdown and reStructuredText syntaxes are mixed. Markdown will be used.
Add "#coerce:reST" or "#coerce:plain" as the first line of your text to force reStructuredText or no markup.
You may also contact the helpdesk if the formatting is incorrect and you are unable to edit your text.
The authors' improvements to the v1 were satisfactory except the following two points:
============
-
The referee still thinks that the title does not appropriately convey the content of the paper; the bulk of the paper discusses the full CFT, while the title focuses on the chiral half of the CFT.
-
The referee still thinks that if we pick a single spin 3/2 primary $G$ (from $2^6$ in the GTVW model) , it is forced to satisfy the N=1 super-Virasoro OPE. The rough argument is that $G(z) G(0)$ can only have singularities of the form $\sim 1/z^3$, $J/z^2$ and $T/z$; if the coefficient of $J$ is nonzero, this means the $JGG$ 3-pt function is nonzero, meaning that $G$ is charged under the U(1) generated by $J$, which is impossible because we chose $G$ to be real.
Was it not the case?
It would also be nice if the authors explicitly describe a model which has a primary of $(h,\bar h)=(3/2,0)$ but still has $Z_{RR}\neq 0$.
============
The referee is prepared to agree to disagree with the authors on the point 1, but the referee would like the authors to take care of the point 2.

Author: Jinbeom Bae on 2021-09-28 [id 1789]
(in reply to Report 1 on 2021-09-24)We thank the referee for the comments.
As explained in the previous letter to the referee, the main purpose of the analysis done in the main context is to show the emergence of supersymmetry in the chiral part of the full CFT. More precisely, when a non-chiral CFT can be fermionized to a supersymmetric theory, we can argue that the chiral part of the given full non-chiral CFT has supersymmetry. We thus believe that the current title can convey the main idea of the paper well and wish to use the title as it is now.
We agree with the referee that an OPE of a real primary of $(h,\bar h)=(3/2,0)$ G involves only two singular terms $\frac{1}{z^3}$ and $\frac{1}{z}$. However, we are not sure that the OPE of $G(z)$ has to be identical to that of the supersymmetry current with the correct OPE coefficients.
As an illustrative example, let us consider a product of two $c=1$ CFTs, $((U(1)_4)/Z_2)^2$, as a bosonic theory $\mathcal{B}$. Performing the femionization (for details, please see a note attached to the letter), one can show that there exists a conserved current of spin-$3/2$ in the NS-sector. However, one can show that the fermion theory $\mathcal{F}$ has a non-constant Ramond-Ramond(RR) partition function and violates the supersymmetry unitarity condition $h_R \ge c/24$. In fact, the RR partition function is proportional to that of the Ising model.
Based on the above example, we believe that the mere presence of a spin-$3/2$ primary does not guarantee the existence of supersymmetry.
Attachment:
NonSUSYExample.pdf
Anonymous on 2021-09-30 [id 1792]
(in reply to Jinbeom Bae on 2021-09-28 [id 1789])I am the referee 1; I would like to thank the authors to prepare a separate note concerning the second point.
The model in the attached note can be described also as follows: you take two compact bosons, and fermionize one of them into two Majorana fermions. So you have operators $X$, $\psi_1$ and $\psi_2$. Your chosen spin-$3/2$ current is $\psi_1 \partial X$. Clearly this generates a standard N=1 super Virasoro acting on $X$ and $\psi_1$, but does not do anything on $\psi_2$. Therefore, if you take the RR partition function, you simply get the partition function of the theory of $\psi_2$, the Ising model.
Touché, indeed. You can always consider a direct sum of a susy sector and a non-susy sector, and then perform some discrete identifications.
I am still not convinced if there is a truly nontrivial example where a spin-3/2 current does not just generate a standard N=1 super Virasoro acting on an almost decoupled susy sector, but I get the main point by the authors.
I am now satisfied and the paper can be published on SciPost.

---

## Round 2 · Author Response

List of changes
To Referee 1:
Regarding major comments, please see the following:
1.) Let us denote a given bosonic theory as ${\cal B}$, its chiral part as $\chi$, and a fermion theory in correspondence to ${\cal B}$ via the generalized Jordan-Wigner transformation as ${\cal F}$. A couple of paragraphs clarifying the relation between the emergent supersymmetry of the chiral theory $\chi$ and the supersymmetry of a fermion theory ${\cal F}$ were added right below the first paragraph on page 2. We also demonstrated a well-known example, the tricritical Ising model, to understand the above relation.
2.) We believe that a primary of $(h,\bar h)=(3/2,0)$, although it is conserved, does not imply supersymmetry. (Note that Haag-Łopuszański-Sohnius theorem is not valid in two-dimension.) More precisely, it is not guaranteed that such a primary automatically satisfies the supersymmetry current OPE. For instance, let us consider a ${\cal B}=(SU(2)_1)^6$ which can be fermionized to a supersymmetric model {\cal F}, the Gaberdiel-Taormina-Volpato-Wendland model (K3 sigma model). The GTVW model has 2^6 primaries of $(h,\bar h)=(3/2,0)$. However, only four of them can satisfy N=4 supersymmetry currents, as shown in arXiv:1309.4127 and arXiv:2003.13700. We also have a few examples where primaries of $(h,\bar h)=(3/2,0)$ in the NS sector are present, but the partition functions in the Ramond-Ramond sector become nontrivial, i.e., $Z_{RR}\neq const.$. Thus, we propose that conditions 1,2, and 3 are necessary conditions.
3.) We have put the reference the referee suggested after presenting the equation (2.45).
4.) Unfortunately, we do not have an answer to the referee's question. Indeed it would be nice if there is a mathematical/physical reason behind the observation $Nk=12$.
Regarding two of the referee's minor comments, please see our responses below:
1.) We agree with the referee that it would be nice to understand why the supersymmetry emerges for the fermionized SO(N)_3 models for any N in a Lagrangian way. Due to the lack of our understanding, we however leave it as a conjecture for now in this manuscript.
- ) One can show that the fermion theory with $c=39/2$ is mapped to a bosonic CFT whose partition function is the non-diagonal modular invariant of the (E_6)_4 WZW model. For instance, the vacuum character of the bosonic CFT can be expressed as a linear combination of two characters for the vacuum and the primary of $h=2$ of (E_6)_4 WZW model.
However, the bosonic CFT of our interest cannot be obtained by Z_2 orbifolding the (E_6)_4 WZW model. This is partly understood from the fact that the Verlinde line of $h=2$ cannot be associated with a Z_2 symmetry. More precisely, the action of the Verlinde line of $h=2$ on each primary is not always given by $\pm 1$.
Following the rest of the referee's minor comments, several typos were fixed and minor corrections were made.
To Referee 2:
We agree that our results very much focused on the 2d CFTs. We added a couple of paragraphs clarifying the relation between the emergent supersymmetry of the chiral theory and the supersymmetry of a fermion theory on page 2. Also, we fixed the typos and added the reference following the referee's suggestion.

---

## Round 2 · List of Changes

To Referee 1:
Regarding major comments, please see the following:
1.) Let us denote a given bosonic theory as ${\cal B}$, its chiral part as $\chi$, and a fermion theory in correspondence to ${\cal B}$ via the generalized Jordan-Wigner transformation as ${\cal F}$. A couple of paragraphs clarifying the relation between the emergent supersymmetry of the chiral theory $\chi$ and the supersymmetry of a fermion theory ${\cal F}$ were added right below the first paragraph on page 2. We also demonstrated a well-known example, the tricritical Ising model, to understand the above relation.
2.) We believe that a primary of $(h,\bar h)=(3/2,0)$, although it is conserved, does not imply supersymmetry. (Note that Haag-Łopuszański-Sohnius theorem is not valid in two-dimension.) More precisely, it is not guaranteed that such a primary automatically satisfies the supersymmetry current OPE. For instance, let us consider a ${\cal B}=(SU(2)_1)^6$ which can be fermionized to a supersymmetric model {\cal F}, the Gaberdiel-Taormina-Volpato-Wendland model (K3 sigma model). The GTVW model has 2^6 primaries of $(h,\bar h)=(3/2,0)$. However, only four of them can satisfy N=4 supersymmetry currents, as shown in arXiv:1309.4127 and arXiv:2003.13700. We also have a few examples where primaries of $(h,\bar h)=(3/2,0)$ in the NS sector are present, but the partition functions in the Ramond-Ramond sector become nontrivial, i.e., $Z_{RR}\neq const.$. Thus, we propose that conditions 1,2, and 3 are necessary conditions.
3.) We have put the reference the referee suggested after presenting the equation (2.45).
4.) Unfortunately, we do not have an answer to the referee's question. Indeed it would be nice if there is a mathematical/physical reason behind the observation $Nk=12$.
Regarding two of the referee's minor comments, please see our responses below:
1.) We agree with the referee that it would be nice to understand why the supersymmetry emerges for the fermionized SO(N)_3 models for any N in a Lagrangian way. Due to the lack of our understanding, we however leave it as a conjecture for now in this manuscript.
- ) One can show that the fermion theory with $c=39/2$ is mapped to a bosonic CFT whose partition function is the non-diagonal modular invariant of the (E_6)_4 WZW model. For instance, the vacuum character of the bosonic CFT can be expressed as a linear combination of two characters for the vacuum and the primary of $h=2$ of (E_6)_4 WZW model.
However, the bosonic CFT of our interest cannot be obtained by Z_2 orbifolding the (E_6)_4 WZW model. This is partly understood from the fact that the Verlinde line of $h=2$ cannot be associated with a Z_2 symmetry. More precisely, the action of the Verlinde line of $h=2$ on each primary is not always given by $\pm 1$.
Following the rest of the referee's minor comments, several typos were fixed and minor corrections were made.
To Referee 2:
We agree that our results very much focused on the 2d CFTs. We added a couple of paragraphs clarifying the relation between the emergent supersymmetry of the chiral theory and the supersymmetry of a fermion theory on page 2. Also, we fixed the typos and added the reference following the referee's suggestion.

---

## Round 3 · Author Response

Dear Editor,

We are grateful for the referee's thoughtful comments on our manuscript. Regarding the referee's report, please find our response below.

  1. As mentioned in the previous letter, we can show that the chiral part of a given full non-chiral CFT has supersymmetry when the non-chiral CFT is fermionized to the supersymmetric theory. We thus wish to use the title as it is now.

  2. We agree with the referee that an OPE of a real primary of $(h,\bar h)=(3/2,0)$ G involves only two singular terms $\frac{1}{z^3}$ and $\frac{1}{z}$. However, we are not sure that the OPE of $G(z)$ has to be identical to that of the supersymmetry current with the correct OPE coefficients.

As an illustrative example, let us consider a product of two $c=1$ CFTs, $((U(1)_4)/Z_2)^2$, as a bosonic theory $\mathcal{B}$. Performing the femionization, one can show that there exists a conserved current of spin-$3/2$ in the NS-sector. However, one can show that the fermion theory $\mathcal{F}$ has a non-constant Ramond-Ramond(RR) partition function and violates the supersymmetry unitarity condition $h_R \ge c/24$. In fact, the RR partition function is proportional to that of the Ising model. As the referee pointed out, this model can be also obtained by performing certain discrete identification of the direct sum of the supersymmetric and non-supersymmetric theory.

Based on the above example, we believe that the mere presence of a spin-$3/2$ primary does not guarantee the existence of supersymmetry.

Best Regards, Jinbeom Bae, Sungjay Lee

---

## Editorial Decision

published